# Peri-mitochondrial actin filaments inhibit Parkin assembly by disrupting ER-mitochondria contacts

Tak Shun Fung [1,4], Amrapali Ghosh [2,4], Maite R Zavala [2], Zuzana Nichtova [2], Dhavalkumar Shukal [2], Marco Tigano [2], Gyorgy Csordas[2], Henry N Higgs [3] & Rajarshi Chakrabarti [2✉]

## Abstract

**Mitochondrial damage represents a dramatic change in cellular homeostasis, necessitating metabolic adaptation and clearance of the damaged organelle. One rapid response to mitochondrial damage is peri-mitochondrial actin polymerization within 2 min, which we term ADA (Acute Damage-induced Actin). ADA is vital for a metabolic shift from oxidative phosphorylation to glycolysis upon mitochondrial dysfunction. In the current study, we investigated the effect of ADA on Pink1/Parkin mediated mitochondrial quality control. We show that inhibition of proteins involved in the ADA pathway significantly accelerates Parkin recruitment onto depolarized mitochondria. Addressing the mechanism by which ADA resists Parkin recruitment onto depolarized mitochondria, we found that ADA disrupts ER–mitochondria contacts in an Arp2/3 complex-dependent manner. Interestingly, overexpression of ER–mitochondria tethers overrides the effect of ADA, allowing rapid recruitment of not only Parkin but also LC3 after mitochondrial depolarization. During chronic mitochondrial dysfunction, Parkin and LC3 recruitment are completely blocked, which is reversed rapidly by inhibiting ADA. Taken together we show that ADA acts as a protective mechanism, delaying mitophagy following acute damage, and blocking mitophagy during chronic mitochondrial damage.**

**Keywords** Actin; Arp2/3 Complex; ER; LC3; Parkin
**Subject Categories** Autophagy & Cell Death; Cell Adhesion, Polarity & Cytoskeleton; Organelles

## Introduction

Mitochondria are well known to oxidize organic molecules for ATP production, using the mitochondrial membrane potential ($\Delta\psi m$) generated by the electron transport chain (ETC) across the inner mitochondrial membrane (IMM) to drive ATP synthase (Mitchell, 2011). In addition, mitochondria are important signaling and biosynthetic hubs, participating in calcium signaling (Nicholls, 2005), lipid synthesis (Nowinski et al, 2020), amino acid synthesis (Ahn and Metallo, 2015), iron-sulfur cluster biosynthesis (Braymer and Lill, 2017), apoptosis (Green, 2022) and heat production (Rustin et al, 2025). Mitochondrial dysfunction can lead to unregulated cell death via ferroptosis (Gao et al, 2019) as well as activation of host-pathogen responses and innate immunity (Kim et al, 2023). For these reasons, several pathways exist to mitigate mitochondrial dysfunction, including mitochondrial destruction by mitophagy.

Mammalian mitophagy involves recognition of dysfunctional mitochondrion through specific receptors and/or adaptors, often in a trigger-specific manner. One such trigger is depolarization (the sustained loss of $\Delta\psi m$), a common outcome of mitochondrial dysfunction. Mitochondrial depolarization activates the Pink1/Parkin mitophagy pathway, which in turn stimulates recruitment of LC3 to feed the mitochondrion into the autophagy pathway. In a healthy mitochondrion, the serine-threonine protein kinase Pink1 is continuously degraded through a system that involves its import into the mitochondrial matrix through the Tom and Tim complexes, which requires mitochondrial membrane potential (Deas et al, 2011; Jin et al, 2010; Meissner et al, 2011; Yamano and Youle, 2013). Upon mitochondrial depolarization, Pink1 is stabilized onto the OMM, where it homodimerizes, self-activates, and phosphorylates both ubiquitin and Parkin (a cytoplasmic E3 ubiquitin ligase), leading to Parkin accumulation on OMM (Lazarou et al, 2012; Okatsu et al, 2013). Mitochondrially-recruited Parkin ubiquitinates several substrates, generating an ubiquitinated mask on the OMM which is recognized by a group of adaptors (P62, NBR1, NDP52, TAX1BP1 and Optineurin (OPTN)). These adaptor proteins subsequently recruit LC3.

A growing number of studies have documented several important roles for actin filaments in regulation of mitochondrial homeostasis and dynamics (Basu et al, 2021; Chakrabarti et al, 2018; Coscia et al, 2024; Fung et al, 2019; Korobova et al, 2013; Li et al, 2015; Moore et al, 2021). We and others have shown that acute mitochondrial depolarization using uncouplers like FCCP or CCCP generates a transient cloud of actin filaments, which we term ADA (acute damage-induced actin), around the depolarized

[1]Department of Cancer Biology and Genetics, Memorial Sloan Kettering Cancer Center, New York, NY, USA. [2]Department of Pathology and Genomic Medicine, Thomas Jefferson University, Philadelphia, PA, USA. [3]Department of Biochemistry and Cell Biology, Geisel School of Medicine at Dartmouth College, Hanover, NH, USA. [4]These authors contributed equally: Tak Shun Fung, Amrapali Ghosh. ✉E-mail: Rajarshi.Chakrabarti@jefferson.edu

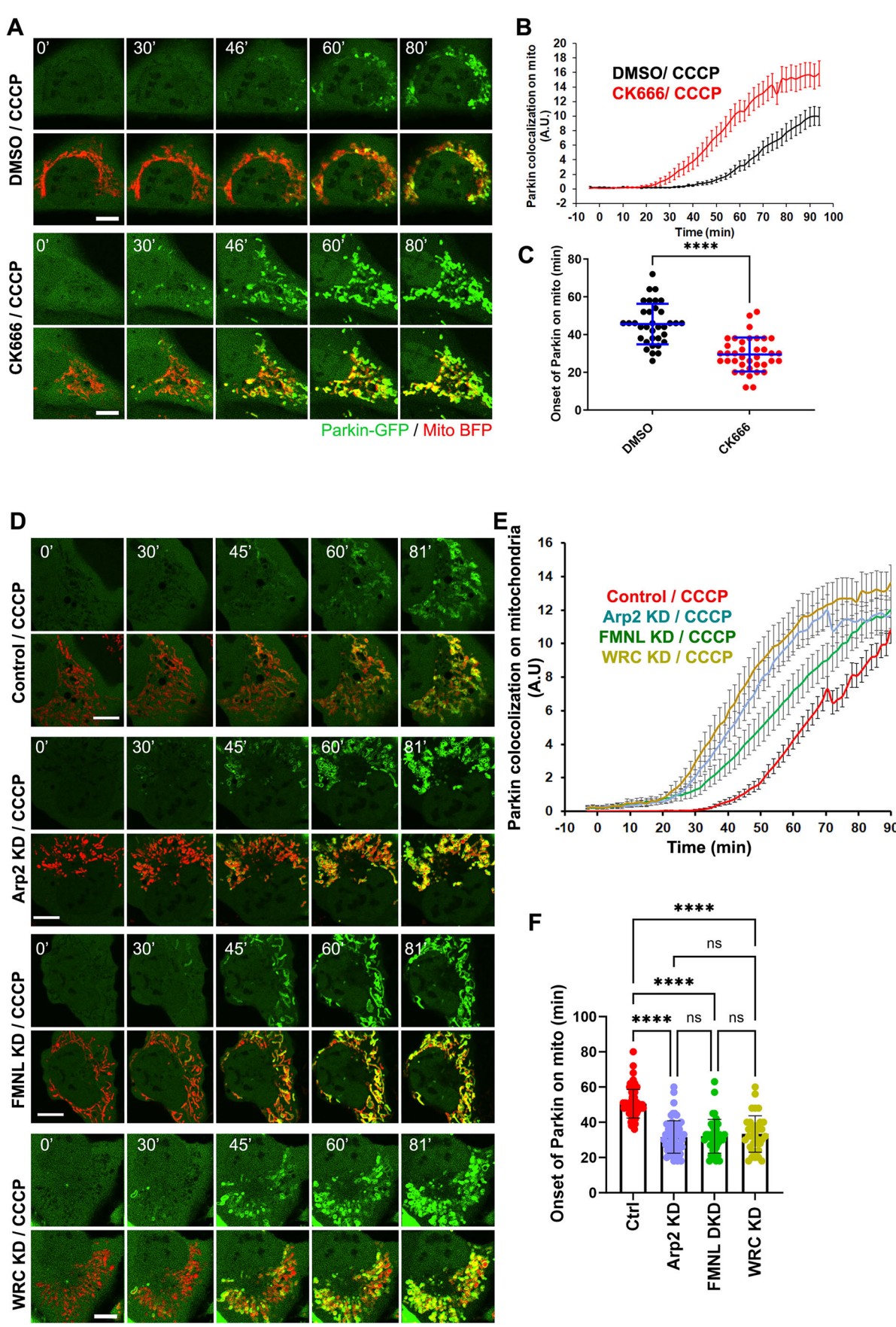

◄ **Figure 1. ADA delays Parkin recruitment to depolarized mitochondria.**

(A) Time lapse montages of WT U2OS cells transfected with GFP-Parkin (green) and Mito-BFP (red) pre-treated with 30 min 100 μM CK666 or DMSO before treatment with 20 μM CCCP treatment during live-cell imaging at time 0. Imaging conducted at the medial cell section. Scale: 10 μm. Time in minutes. (B) Quantification of colocalized Parkin signal with mitochondrial signal in live-cell imaging after 20 μM treatment of CCCP at time 0 for 30 min DMSO (black) or 100 μM CK666 (red) pre-treated U2OS cells. Data of 35 cells for DMSO/CCCP and 38 for CK666/CCCP from four independent experiments. Error ± SEM. (C) Scatter plot of Parkin signal onset on mitochondria for DMSO or 100 μM CK666 treated U2OS cells. Parkin onset time = 45.54 min ± 10.74 (mean ± s.d.) for DMSO/CCCP and 29.42 ± 10.74 for CK666/CCCP. Error: ±s.d. Same dataset as in (B) (Data of 35 cells for DMSO/CCCP and 38 for CK666/CCCP from 4 independent experiments). $P = 0.0001$ (****) using Student's unpaired $t$ test. (D) Time lapse montages of ctrl, Arp2 KD, FMNL1/3 DKD and WRC KD U2OS cells transfected with GFP-Parkin (green) and Mito-BFP (red) treated with 20 μM CCCP treatment at time 0 during live-cell imaging. Imaging conducted at the medial cell section. Scale: 10 μm. Time in minutes. (E) Quantification of colocalized Parkin signal with mitochondrial signal in live-cell imaging after 20 μM treatment of CCCP at time 0 for ctrl, FMNL1/3 DKD or Arp2 KD U2OS cells. Data of 79 cells for ctrl; 62 cells for Arp2 KD; 42 cells for FMNL1/3 DKD and 35 cells for WRC KD from 3 independent experiments. Error ± SEM. (F) Scatter plot of Parkin signal onset on mitochondria for ctrl, Arp2 KD, FMNL1/3 DKD and WRC KD in U2OS cells. Parkin onset time = 46.97 min ± 8.86 (mean ± s.d.) for ctrl; 32.06 ± 9.03 for Arp2 KD; 32.61 ± 8.75 for FMNL 1/3 DKD and 33.37 ± 10.30 for WRC KD cells. Same dataset as in (E) (Data of 79 cells for ctrl; 62 cells for Arp2 KD; 42 cells for FMNL1/3 DKD and 35 cells for WRC KD from three independent experiments). $P = 0.0001$ (****) for ctrl vs FMNL1/3 DKD and ctrl vs Arp2 KD; $P = 0.9795$ (n.s.) for FMNL1/3 DKD vs Arp2 KD, $P = 0.7808$ (n.s.) for WRC KD vs Arp2 KD. Tukey's multiple comparisons test used. Error: ± s.d.

mitochondrion within 5 min (Fung et al, 2019; Li et al, 2015) through signaling pathways requiring AMP-dependent protein kinase (AMPK) and protein kinase C (Fung et al, 2022). ADA is also activated by hypoxia, ETC inhibitors, and ATP synthase inhibition (Chakrabarti et al, 2022). Interestingly, chronic mitochondrial dysfunction (through depletion of mtDNA or the key ETC protein NDUFS4) causes ADA-like filaments around mitochondria that persist for days but are constantly turning over, with a half-life of less than 10 min (Chakrabarti et al, 2022).

Two functional consequences for ADA have been identified. ADA transiently suppresses mitochondrial morphological dynamics that are triggered by mitochondrial depolarization (Fung et al, 2022; Fung et al, 2019). These dynamics involve rearrangement of the inner mitochondrial membrane (IMM), dependent on the IMM protease Oma1, resulting in circularization of the IMM within an intact outer mitochondrial membrane (OMM) (Fung et al, 2022; Fung et al, 2019; Miyazono et al, 2018). ADA inhibits both mitochondrial circularization and proteolytic processing of the Oma1 substrate Opa1. A second ADA function is to rapidly stimulate glycolysis. This glycolytic stimulus is also a feature of the persistent ADA-like filaments that assemble following chronic mitochondrial damage (Chakrabarti et al, 2022). It is possible that ADA has additional effects.

Here, we asked whether ADA has a role in regulating mitophagy. We show that ADA significantly delays Parkin recruitment and downstream LC3 recruitment, through acutely disrupting endoplasmic reticulum (ER)-mitochondrial contacts (ERMC). Intriguingly, disassembly of the ADA-like filaments present around chronically damaged mitochondria not only allows both Parkin and LC3 recruitment but also increases mitophagy. Surprisingly, ADA inhibition also speeds up Pink1 accumulation and activity upon CCCP and Antimycin/Oligomycin treatments. Overall, our results suggest that ADA serves as an acute roadblock to the early events in Pink1-Parkin mediated mitophagy, through the regulation of ER-mitochondrial contacts.

## Results

### ADA delays Parkin recruitment onto damaged mitochondria

Damaged or depolarized mitochondria can be subject to multiple fates, including mitophagy (Boland et al, 2013; Palikaras et al, 2018). Though the final steps of mitophagy occur much later than the initial

depolarization (McWilliams and Muqit, 2017), Parkin recruitment to the OMM is an early step in the process (Harper et al, 2018; Pickles et al, 2018). To test whether induction of ADA affected recruitment of Parkin to damaged mitochondria, we blocked ADA by inhibiting the Arp2/3 complex and evaluated the kinetics of GFP-Parkin recruitment to mitochondria by live-cell microscopy and compared it with that of control cells. In control U2-OS cells, Parkin recruitment starts at 45.5 min ± 10.7 min after CCCP treatment. Both CK666 pre-treatment (Arp2/3 inhibition) (Fig. 1A–C) and Arp2 KD (Arp2/3 depletion) (Figs. 1D–F and EV1A (upper)) result in significant acceleration of Parkin recruitment (29.4 min ± 9.0 min and 31.6 min ± 8.3 min, respectively). We obtain similar results in HeLa cells, where CK666 treatment (Fig. EV1B–D) or Arp2 KD (Fig. EV1A (lower) causes significantly faster CCCP-induced Parkin recruitment (25.2 min ± 10.9 min and 29.0 min ± 6.1 min, respectively) than control cells (46.3 min ± 12.2 min). To test this effect further, we suppressed two other key components of the ADA signaling pathway: the Wave Regulatory Complex (WRC) and the FMNL formins (Fung et al, 2022). Either FMNL KD (32.6 min ± 8.7 min) or WRC KD (33.4 min ± 10.3 min) in U2OS cells cause significant acceleration of Parkin recruitment compared to control cells (Figs. 1D–F and EV1A (upper)). FMNL depletion in HeLa cells also showed a similar trend of accelerated Parkin recruitment compared to control cells (Fig. EV1A (lower), E–G). We have previously shown that a plethora of mitochondrial poisons could induce ADA in mouse embryonal fibroblasts (MEFs) in Arp2/3-dependent manner (Chakrabarti et al, 2022). We therefore tested out ADA-dependent Parkin recruitment in MEF cells. While 60 min of CCCP treatment did not show appreciable Parkin recruitment onto mitochondria in control cells, there was significantly higher recruitment of Parkin in Arp2/3 inhibited (CK666 treated) MEFs following CCCP treatment (Fig. EV1H–I). Taken together, these results suggest that ADA significantly delays Parkin recruitment onto damaged mitochondria in both U2-OS, HeLa and MEF cells.

Combined treatment of cells with Antimycin A/Oligomycin A (A/O) have been shown to recruit Parkin onto mitochondria with a similar kinetics to that of CCCP treatment (Hung et al, 2021). We therefore wanted to examine whether ADA had similar effects on Parkin recruitment in A/O stimulated cells. HeLa cells treated with A/O showed mitochondrial depolarization which was slower and lower than CCCP-treated cells (Fig. EV2A) but elicited Arp2-dependent ADA responses (Fig. EV2B,C). HeLa cells treated with CK666 (Arp2/3 inhibition) show significantly higher mitochondrial localized Parkin

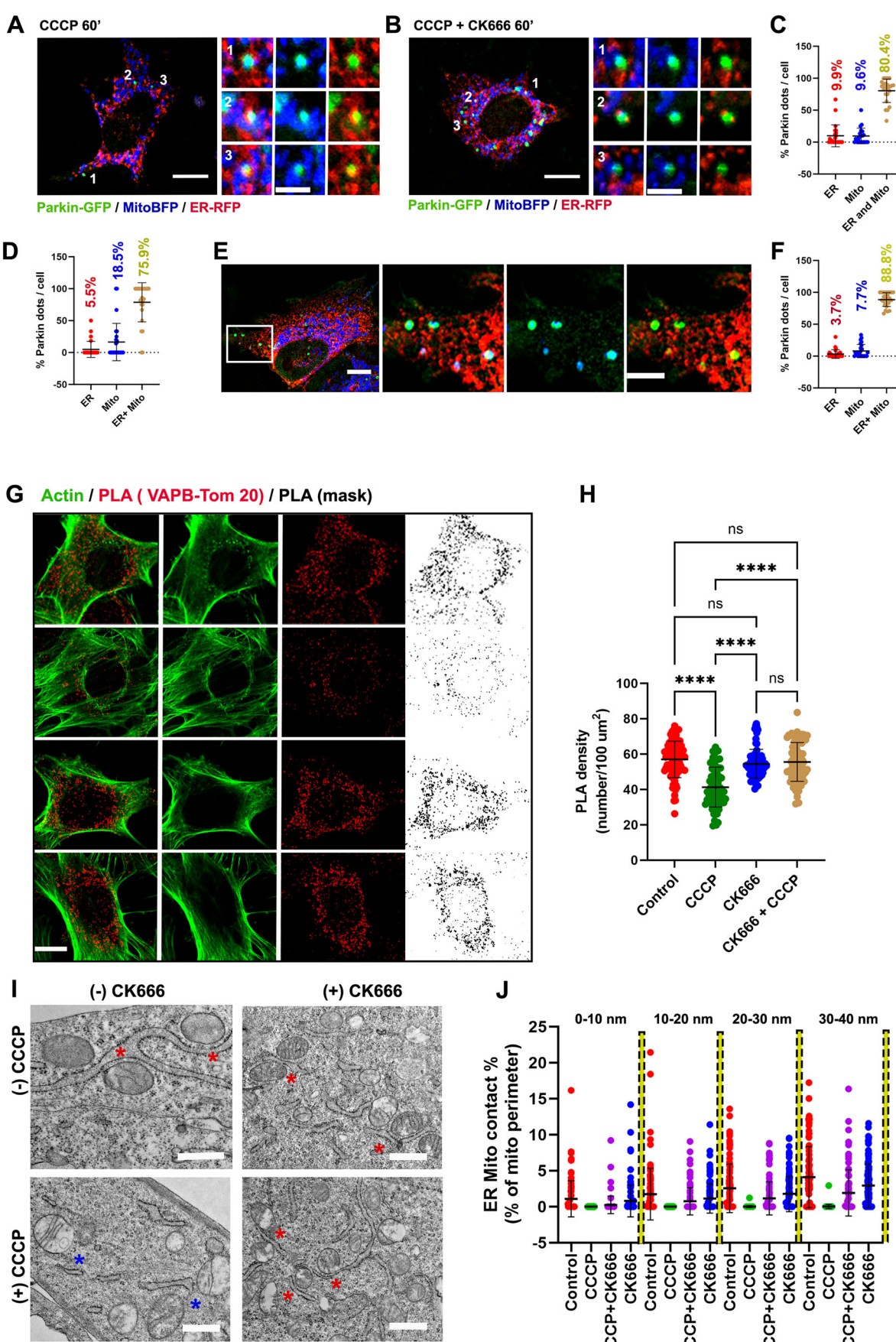

**Figure 2. ADA acutely disrupts ER–mitochondrial contacts (ERMC).**

(A) Fixed cell image of a MEF transfected with Parkin-GFP, mitoBFP and ER-RFP treated with 20 μM CCCP for 60 min. Three ROI from the image shown. Scale: 10 um (main image); 2.5 um (ROI). (B) Fixed cell image of a MEF transfected with Parkin-GFP, mitoBFP and ER-RFP treated with 20 μM CCCP + CK666 (100 μM) for 60 min. Three ROI from the image shown. Scale: 10 μm (main image); 2.5 μm (ROI). (C) Quantification showing % Parkin puncta (dots) per cell either on ER (red) or on mitochondria (blue) or on both simultaneously (gold) in MEF cells treated with 20 μM CCCP (60 min). Each dot represents one cell. Data of 23 cells for each condition from three independent experiments. Error: ±s.d. (D) Quantification showing % Parkin puncta (dots) per cell either on ER (red) or on mitochondria (blue) or on both simultaneously (gold) in MEF cells treated with 20 μM CCCP + CK666 (100 μM/60 min). Each dot represents one cell. Data from 26 cells for each condition from three independent experiments. Error: ±s.d. (E) Fixed cell image of HeLa cells transfected with Parkin-GFP, mitoBFP and ER-RFP treated with 20 μM CCCP for 20 min. White arrows indicate regions of Parkin colocalization with ER and mitochondria. Scale: 10 μm (main image); 5 μm (inset). (F) Quantification showing % Parkin puncta (dots) per cell either on ER (red) or on mitochondria (blue) or on both simultaneously (gold) in HeLa cells treated with 20 μM CCCP (20 min). Each dot represents one cell. Data of 26 cells for each condition from three independent experiments. Error: ±s.d. (G) Representative fixed-cell images from Proximity Ligation Assay (PLA) conducted between VAPB (ER) and PTPIP51 (mitochondria) to assess ERMC (red and binarized) in MEF cells treated with 20 μM CCCP for 5 min in the presence or absence of CK666 (100 μM). Cells were stained with Phalloidin-488 to label actin filaments prior to PLA staining. Arrows represent ADA induction. Scale: 10 μm. (H) Scatter plot representing density of PLA dots (no. per 100 μm$^2$) in control and CCCP-treated HeLa cells in the presence or absence of CK666. Data from 86 cells (control); 72 cells (CCCP); 75 cells (CK666) and 84 cells (CK666 + CCCP) from 20–25 imaging fields and 3 independent experiments. $P = 0.0001$ (****); for untreated vs CCCP and CCCP vs CK666/CCCP. Error: Mean ±s.d. One-way ANOVA used. (I) Representative images from thin-section TEM conducted in MEFs treated with 20 μM CCCP for 5 min in the presence or absence of CK666 (100 μM) showing ER and mitochondria in the sections. Red asterisk indicate close contacts between ER and mitochondria and blue asterisk indicate decreased contacts. Scale: 500 nm. (J) Scatter plots showing the percentage of mitochondrial perimeter within 0–40 nm distance to the ER using 10 nm bins in Control (Red; $n = 88$ mitochondria); 20 μM CCCP/ 5 min (green; $n = 94$ mitochondria); 20 μM CCCP + CK666 (100 μM/5 min) (purple; $n = 90$ mitochondria) and 100 μM CK666 treated (blue; $n = 106$ mitochondria) MEF cells. Data from three independent fixation for each condition. $P = 0.0001$(****). Unpaired $t$ test; Error: Mean ±s.d.

upon 20 and 40 min of A/O treatment as compared to the control cells (Fig. EV2D,E). The amount of mitochondrial area positive for Parkin was not significantly different after 60 min of treatment in either cases suggesting that ADA acutely delays Parkin recruitment upon A/O treatment (Fig. EV2D,E).

Next, we investigated the mechanism by which ADA delays Parkin recruitment. One possibility is that the thick network of actin filaments serves as a diffusive barrier to impede access of Parkin or its upstream factor(s) to the OMM. To test this possibility, we utilized an inducible mitochondrial recruitment system, by expressing cytosolic CFP-(FRB)5 and mitochondrially targeted AKAP1-FKBP-YFP. Upon rapamycin addition, CFP-(FRB)5 is recruited to the OMM in 15–30 s (Fig. EV3A). We tested the effect of ADA on CFP-(FRB)5 recruitment by adding rapamycin ~100 s after CCCP addition (the point of maximal ADA (Fung et al, 2022; Fung et al, 2019). The rate of CFP-(FRB)5 recruitment is not measurably influenced by ADA (Fig. EV3B,C). Since CFP-(FRB)5 is larger than Parkin (82 versus 52 kDa), these results suggest that the actin filaments polymerized during ADA do not physically block the access of Parkin to depolarized mitochondria. We therefore examined other mechanisms that could explain the delay in Parkin recruitment.

## ADA disrupts ER–mitochondrial contacts

Curiously the Parkin recruitment in MEF cells occurred as puncta as opposed to more homogenous staining as observed in HeLa or U2-OS cells. This led us to spatially characterize the Parkin puncta with respect to ER and mitochondria. Surprisingly, we noticed that about 80% and 75% of the Parkin puncta in control and Arp2/3 inhibited (CK666) CCCP-treated MEF cells respectively were positive for both ER and mitochondria (Fig. 2A–D). Similar results were also observed in HeLa cells following 20 min of CCCP treatment and selecting cells containing punctate Parkin distribution (Fig. 2E,F). Curiously, we noticed that the actin filaments assembled during ADA often occurred between mitochondria and ER (Fig. EV3D). We therefore asked whether ADA could actively displace ER from depolarized mitochondria thereby affecting the

kinetics of Parkin assembly. To test the effect of ADA on ER-Mitochondrial contacts (ERMC), we used Proximity Ligation Assay (PLA) for interacting partners that mediate ERMC: VAP-B (ER) and PTPIP51 (mitochondria) (Obara et al, 2024). MEF cells treated for 5 min with CCCP displayed ADA induction, as well as a significant decrease in PLA density (Fig. 2G,H) suggesting an acute reduction of ERMC. Consistently, simultaneous treatment of CK666 together with CCCP restores PLA density (Fig. 2G,H), while acute CK666 treatment alone does not result in significant changes in PLA density compared to the control cells (Fig. 2G,H). We repeated the PLA assay with a second combination of proteins: VAP-B (ER) and Tom20 (mitochondria). Again, CCCP causes a decrease in PLA density prevented by CK666 (Fig. EV3E,F). Overall, these results suggest that peri-mitochondrial actin assembly following acute mitochondrial depolarization dynamically disrupts ERMC. Furthermore, we measured ERMC from thin section EM of MEF cells treated with CCCP in the presence or absence of CK666. In all, 5 min of CCCP treatment showed a significant decrease in ERMC and an increase in the minimum distance between ER and mitochondria (Figs. 2I,J and EV3G). Similar to the results obtained in the PLA assay, while simultaneous treatment of CK666 together with CCCP does not disrupt these contacts (Figs. 2I,J and EV3G), cells treated with CK666 alone for 5 min do not show any disruption in ERMC compared to the control cells (Fig. 2I,J). CCCP treatment also induces a significant increase in minimum distance between ER and mitochondria which is completely blocked by Arp2/3 inhibition (Fig. EV3G) As a control, we tested whether ADA could disrupt the contact between mitochondria and another subcellular organelle, lysosomes. The extent of mitochondria and lysosome contacts was not influenced by ADA (Fig. EV3H,I). Altogether, these results suggest that ADA might specifically disrupt ERMC.

## ADA-induced ERMC disruption delays Parkin recruitment onto mitochondria

Since we showed that a substantial number of Parkin recruitment occurs close to ER and mitochondria (Fig. 2A–D) we tested whether

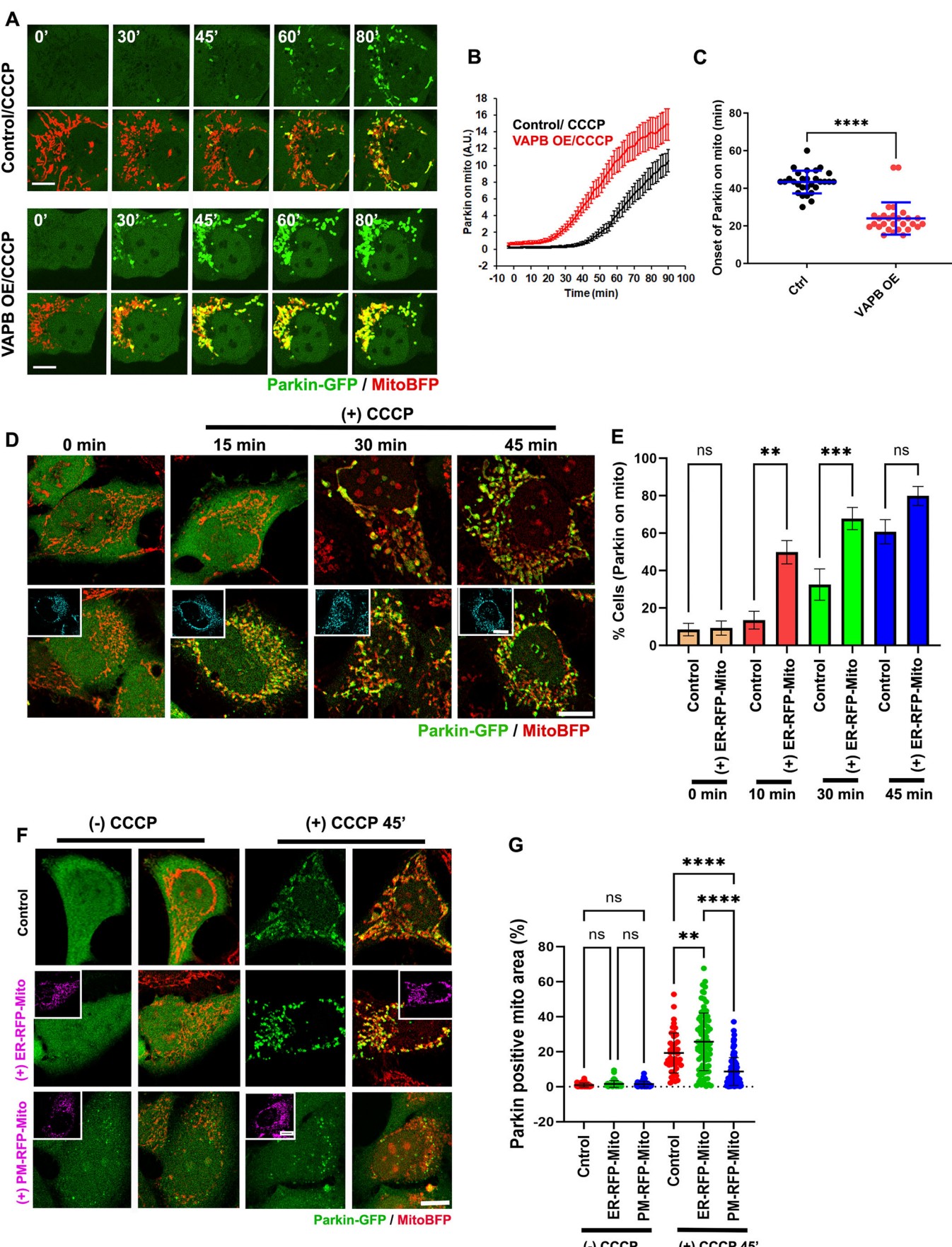

◀ **Figure 3.  Enhancing ERMC speeds up Parkin recruitment on depolarized mitochondria.**

(A) Time lapse montages of control and myc-VAP-B-overexpressing U2OS cells transfected with GFP-Parkin (green) and Mito-BFP (red) and treated with 20 μM CCCP treatment at time 0 during live-cell imaging. Imaging conducted at the medial cell section. Scale: 10 μm. Time in minutes. (B) Quantification of colocalized Parkin signal with mitochondrial signal in live-cell imaging after 20 μM treatment of CCCP at time 0 for control (black) and myc-VAP-B overexpressing (red) U2OS cells. Data of 28 cells for ctrl and 29 for VAPB overexpression from 4 independent coverslips. Error ± SEM. (C) Scatter plot of Parkin signal onset on mitochondria for control and myc-VAP-B overexpressing U2OS cells. Parkin onset time = 43.34 min ± 6.02 (mean ± s.d.) for control and 23.95 ± 8.578 for myc-VAP-B overexpression. Same dataset as in (B) (Data of 28 cells for ctrl and 29 for VAPB overexpression from four independent coverslips). $P = 0.0001$ (****). Student's unpaired t test. (D) Representative images of HeLa cells transiently expressing mitoBFP (red) and Parkin-GFP (green) either alone or in combination with ER-RFP-Mito (ER-Mito linker plasmid; cyan), treated with 20 μM CCCP at time point, fixed and imaged at various time points. Scale bar: 10 μm. (E) Quantification showing percentage of cells with Parkin positive mitochondria from images in (D). Data from three independent coverslips having 50–120 cells from 10 to 20 fields for each condition. Error ± SEM. $P = 0.002$ (**) and $P = 0.0006$ (***); Students t test used. (F) Representative images of HeLa cells transiently expressing mitoBFP (red) and Parkin-GFP (green) either alone or in combination with ER-RFP-Mito (ER-Mito linker plasmid; lower panel) or PM-RFP-Mito (PM-Mito linker plasmid; middle panel) treated with or without 20 μM CCCP. Scale bar: 10 μm. (G) Quantification showing Parkin positive mitochondria area (%) in each cell from images in (F). Data from three independent experiments where for -CCCP: Control (51 cells); ER-Mito (47 cells); PM-Mito (55 cells). + 45 min CCCP: Control (45 cells); ER-Mito (81 cells); PM-Mito (111 cells); Error: Mean ±s.d. $P = 0.0044$ (**) for Control 45 CCCP Vs ER-RFP-Mito CCCP45; $P = 0.0001$ (****) for Control 45 CCCP Vs PM-RFP-Mito CCCP45 and PM-RFP-Mito 45 CCCP vs ER-RFP-Mito CCCP45; one-way ANOVA used.

increasing ERMC could overcome the inhibitory effect of ADA on Parkin recruitment. To increase ERMC, we overexpressed VAPB (Gomez-Suaga et al, 2017). VAPB overexpression (Fig. EV4A) accelerates mitochondrial recruitment of Parkin upon CCCP treatment in U2-OS cells to an extent similar to that of Arp2- or FMNL-depletion (Fig. 3A–C), without affecting the kinetics of ADA (Fig. EV4B). To exclude the possibility of a specific effect of VAPB on stimulating Parkin recruitment by direct interaction, we also tested a previously characterized synthetic ER-mitochondrial linker (ER-RFP-Mito) that has been shown to constitutively induce ERMC (Nichtová et al, 2023). ER-RFP-Mito significantly enhances histamine-induced ER-to-mitochondria calcium transfer (Fig. EV4C,D), indicative of an increase in functional ERMC. Cells expressing ER-RFP-mito not only show a higher proportion of CCCP-treated cells displaying Parkin-positive mitochondria (49.7 ± 5.8% and 67.7 ± 5.9% in 15 and 30 min, respectively) over control cells (13.5 ± 3.6% and 32.5 ± 8.6%), (Figs. 3D,E and EV4E) but also a significant increase in Parkin-positive mitochondrial area per cell at 45 min after CCCP treatment (Fig. 3D–G). Notably, ER-RFP-mito expression does not inhibit CCCP-induced ADA (Fig. EV4E).

To corroborate the role of ERMC in Parkin recruitment, we redirected mitochondria to the plasma membrane, by overexpressing a previously characterized plasma-membrane mitochondrial linker construct (PM-RFP-mito). Expression of this construct tethers mitochondria to the plasma membrane where mitochondria still retain its membrane potential and calcium uptake ability but is significantly de-tethered from the ER. (Katona et al, 2022; Naghdi et al, 2010). Accordingly, PM-RFP-mito reduces histamine-stimulated ER-to-mitochondrial calcium transfer (Fig. EV4C,D) but do not affect CCCP-induced ADA assembly (Fig. EV4E). PM-RFP-mito expressing cells significantly lose the ability to accumulate Parkin on mitochondria, with 9.6 ± 8.6% of their mitochondrial area positive for Parkin after 45-min treatment of CCCP (Fig. 3F,G), significantly lower than control cells or ER-RFP-mito expressing cells treated with CCCP. Overall, these results suggest that (1) ER-mitochondrial proximity positively influences Parkin recruitment onto damaged mitochondria, and that (2) ADA delays Parkin recruitment by transiently disrupting ERMC.

## VDAC1 is required for CCCP-induced Parkin recruitment in HeLa cells independent of ERMC regulation

Having established that Parkin recruitment involves the presence of both ER and mitochondria in the vicinity and that modulation of

ERMC affects its kinetics following mitochondrial depolarization, we wanted to identify the molecular determinant(s) at ERMC mediating the process. MFN2 and VDAC have been independently shown to regulate both ERMC (Geisler S et al, 2010; Csordás et al, 2018) and Parkin recruitment (Chen and Dorn, 2013; Sun et al, 2012). siRNA mediated knockdown of both MFN2 and VDAC1(Fig. 4A) significantly subdued CCCP-induced Parkin recruitment in HeLa cells (Fig. 4B,C). We further wanted to test whether this was due to an alteration of ERMC or the absence of proteins per se. To evaluate this we expressed ER-RFP-Mito synthetic tethers in control, MFN2 KD and VDAC1 KD cells and treated them with CCCP for 45 min. While ER-Mito tethering increased Parkin positive mitochondrial area significantly in control and MFN2 KD cells compared to the respective untransfected cells (Fig. 4B,C), VDAC1 KD cells expressing ER-RFP-Mito lacked Parkin recruitment following CCCP treatment (Fig. 4B,C). These results suggest that while MFN2 regulates Parkin recruitment in an ERMC dependent manner, presence of VDAC1 is obligatory for CCCP-induced Parkin recruitment in HeLa cells.

## ADA delays mitochondrial LC3 recruitment downstream of Parkin accumulation in an ERMC-dependent manner

Next, we asked whether ADA influences the kinetics of LC3 recruitment downstream of Parkin accumulation. HeLa cells transfected with HA-Parkin display 9.0 ± 5.0% mitochondrial area positive for GFP-LC3 at 90 min after CCCP treatment (Fig. 5A,B). CK666 pre-treated or Arp2 KD cells show significantly higher proportions of LC3-positive mitochondrial area at a similar time point (17.0 ± 5.9% and 17.6 ± 5.4%, respectively) (Fig. 5A,B). Further, we also evaluated LC3 levels through western blots. 90 and 120 min of CCCP treatment results in significantly increased LC3B(II) levels in Arp2/3 inhibited (CK666 treated) HeLa cells compared to the control cells (Fig. 5C,D). Increasing ERMCs through overexpression of VAPB or ER-RFP-mito phenocopies Arp2/3-inhibition, with 17.7 ± 4.5% and 16.9 ± 5.3% mitochondrial area positive for LC3 after 90-minute CCCP treatment respectively (Fig. 5A,B). As a control, we confirmed that CK666 pretreatment, Arp2 KD, VAPB OE, or expression of ER-RFP-mito alone do not induce mitochondrial LC3 accumulation in the absence of CCCP stimulation (Fig. 5A,B). We also see an increased depletion of MFN2 in the Arp2 inhibited cells compared to the control cells upon 90 and 120 min of CCCP treatment (Fig. 5C,D) suggestive of higher mitophagy at these later time-points in Arp2/3 inhibited

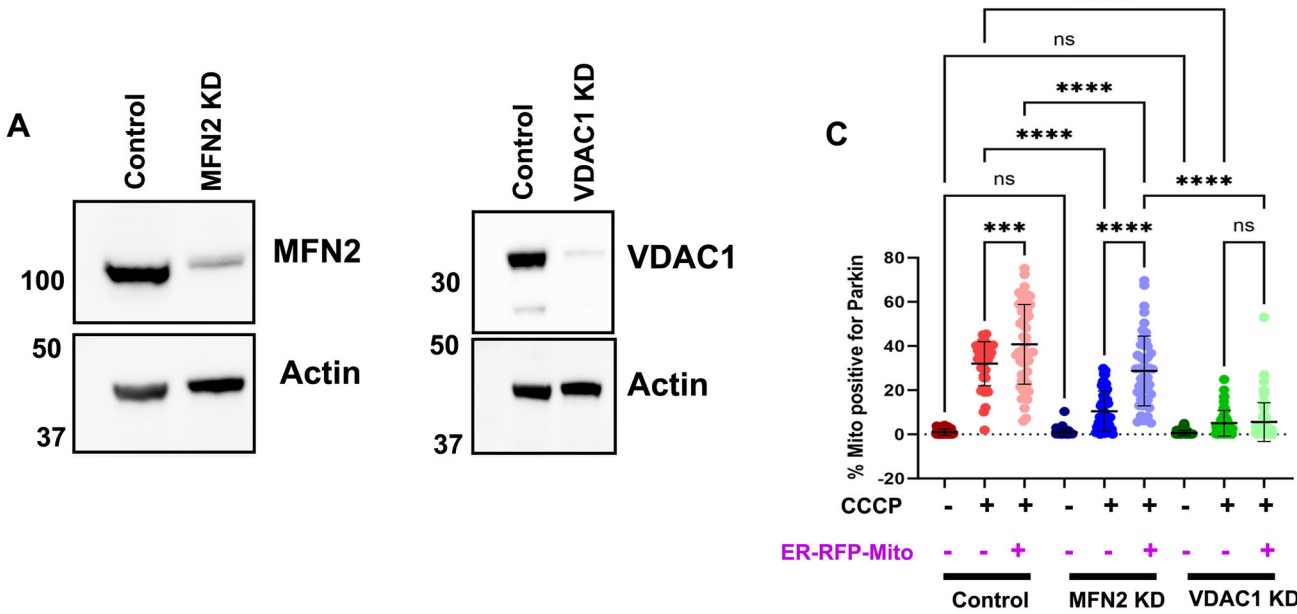

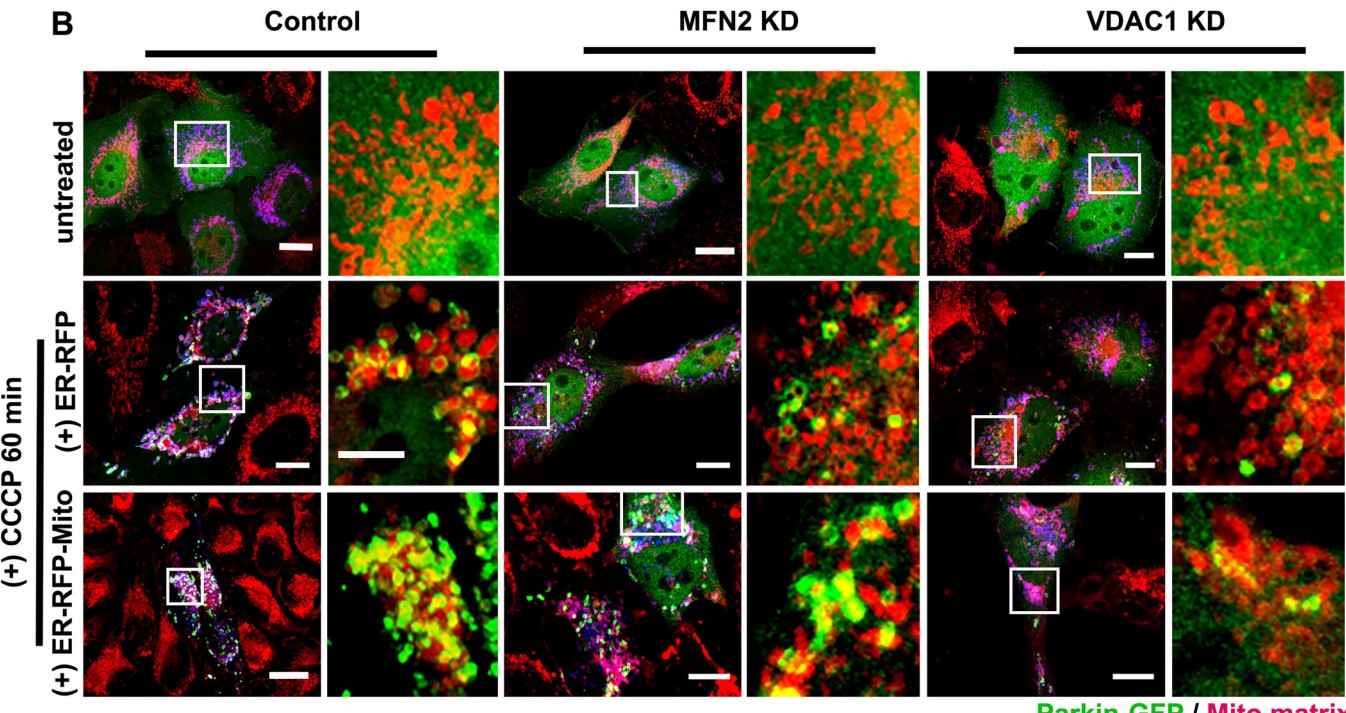

**Parkin-GFP / Mito matrix**

**Figure 4. VDAC1 regulates Parkin recruitment in an ERMC-independent manner.**

(A) Representative western blots for Mfn2 and VDAC1 knockdown in HeLa cells. Actin used as loading control. Molecular weight in kDa. (B) Representative fixed-cell from Control, Mfn2 KD and VDAC1 KD HeLa cells transfected with either ER-RFP/Parkin-GFP or ER-RFP-Mito/Parkin GFP and stained with Mito-brilliant 646 (TOCRIS). Cells were either left untreated or treated with 20 μM CCCP for either 60 min (without tether) or 45 min (with ER-RFP-mito tethers). Scale: 10 μm (main image); 2.5 μm (inset). White box denotes zoomed region. (C) Scatter plot showing percentage of mitochondrial area positive for Parkin in the images taken for (B). Data of 42 cells (Control untreated); 40 cells (Control + CCCP); 53 cells (Control + ER-RFP-Mito + CCCP); 49 cells (Mfn2 KD untreated); 54 cells (Mfn2 KD + CCCP); 47 cells (Mfn2 KD + ER-RFP-Mito + CCCP); 53 cells (VDAC1 KD untreated); 52 cells (VDAC1 KD + CCCP); 61 cells (VDAC1 KD + ER-RFP-Mito + CCCP) from three independent experiments; $P = 0.0001$ (****) one way ANOVA; Error: Mean ±s.d. Source data are available online for this figure.

**A**

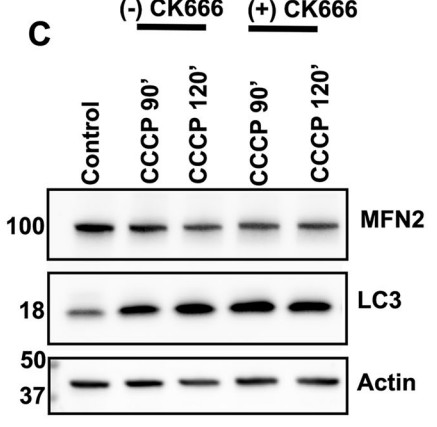

GFP-LC3 / MitoBFP

**B**

% Mito area positive for LC3

**C**

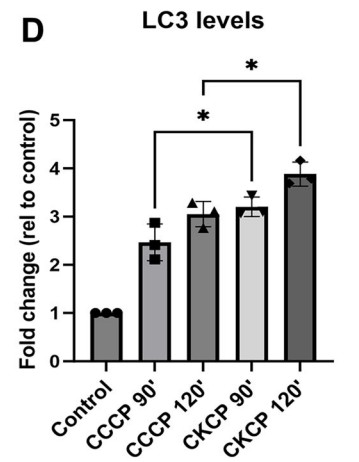

**D**

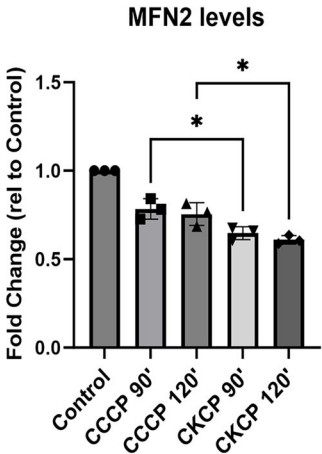

LC3 levels

MFN2 levels

◄ **Figure 5. Enhancing ERMC speeds up LC3 recruitment onto depolarized mitochondria.**

(A) Representative images of control, CK666 treated (100 μM), Arp2 KD, VAPB overexpressed (VAPB OE) and ER-RFP-Mito (blue) transfected HeLa cells transiently expressing GFP-LC3 (green), HA-Parkin and mitoBFP (red) before and after CCCP treatment (20 μM/90 min). Regions next to arrows expanded (third panel) to show LC3 recruitment around mitochondria in CCCP-treated cells from each condition. Scale: 10 μm (main panel); 2.5 μm (inset). (B) Quantification showing LC3-positive mitochondria area (%) in each cell from images in (A). Control (untreated): 28 cells; CCCP 90': 71 cells; CK666 90': 21 cells; CK666 + CCCP 90': 36 cells; Arp2 KD: 25 cells; Arp2 KD + CCCP 90': 25 cells; VAP-B OE: 21 cells; VAP-B OE + CCCP 90': 30 cells; ER-RFP-Mito: 28 cells; ER-RFP-Mito + CCCP 90': 32 cells from three independent coverslips. $P = 0.0001$ (****). Error: Mean ±s.d. One-way ANOVA used. (C) Western blot of HeLa cells overexpressing Parkin-GFP treated with either 20 μM CCCP or 20 μM CCCP + CK666 (100 μM) for 90 and 120 min and immunoblotted for Mfn2, LC3 and actin (loading control). Molecular weight in kDa. (D) Quantification showing relative levels of LC3 and Mfn2 in HeLa cells treated with either 20 μM CCCP or 20 μM CCCP + CK666 (100 μM) for 90 and 120 min. $P = 0.0323$ (CCCP 90 vs CKCP 90) (*), $P = 0.0162$ (CCCP 120 vs CKCP120 (*) for LC3 quantification and $P = 0.0206$ (CCCP 90 vs CKCP 90) (*), $P = 0.0206$ (CCCP 120 vs CKCP120 (*) for Mfn2 quantification. One-way ANOVA used. $N = 3$ independent experiments. Error: ±s.d. Source data are available online for this figure.

cells. These results suggest that by delaying Parkin accumulation, ADA delays downstream mitophagy events, through regulation of ER-mitochondrial contacts.

## ADA acutely regulates PINK1 stabilization following mitochondrial damage

Next, we asked whether ADA also regulates Pink1 stabilization following mitochondrial insults upstream of Parkin recruitment. HeLa cells transiently expressing GFP-Pink1 were treated with either CCCP or A/O in the presence or absence of CK666 (Arp2/3 inhibition). Arp2/3 inhibition showed a significantly increased levels of GFP-Pink1 at 20 min following both CCCP (Fig. 6A,B) and A/O (Fig. 6C,D) treatments. At 40 min, Arp2/3 inhibited cells continued to have increased GFP-Pink1 levels for A/O treated samples, but the levels were statistically not different from the control cells upon CCCP treatment (Fig. 6A–D). Thus it seems that Arp2/3 inhibition through CK666 significantly speeds up Pink1 stabilization upon mitochondrial damage. Pink1 upon activation phosphorylates both Parkin and Ubiquitin at Ser65 (Hung et al, 2021). Indeed Arp2/3 inhibited HeLa cells showed a higher P-ParkinS65 immunoreactive band at early time-points (10 and 20 min of treatment) compared to the control cells. These cells also showed a significant shift in Parkin bands, most likely due to phosphorylation at these early time points (Fig. 6E). Arp2/3 inhibited cells also showed significantly higher phospho-Ubiquitin bands at 20 min following CCCP treatment that continued to be significantly higher even after 60 min of CCCP treatment (Fig. 6E). Similar increased levels of phospho-Ubiquitin levels were seen in Arp2/3 inhibited HeLa cells treated with A/O for related time periods (Fig. 6F). Overall our results show that ADA not only delays Parkin recruitment through modulation of ERMC but also has a significant role in regulating the stabilization and/or activity of its upstream regulator PINK1 in HeLa cells.

## Chronically dysfunctional mitochondria display actin-dependent mitophagy inhibition

We had previously shown that chronic OxPhos reduction through mtDNA depletion in MEF cells (achieved with a low dose of ethidium bromide (EtBr)) induces ADA-like peri-mitochondrial actin filaments which are significantly stable (Chakrabarti et al, 2022). Despite persisting for days, these peri-mitochondrial actin filaments are eliminated by a 10-min treatment with CK666, showing that they are rapidly turning over and require continuous Arp2/3 complex activity (Chakrabarti et al, 2022). We therefore

tested whether these ADA-like filaments inhibit both Parkin and LC3 recruitment on chronically damaged mitochondria.

Treatment of MEFs with low doses of EtBr induces a significant reduction in mtDNA copy number (Fig. EV5A), virtual elimination of both basal and stimulated OCR (Fig. EV5B), and an increase in basal ECAR (glycolysis) (Fig. EV5B). Similar to past results (Chakrabarti et al, 2022), mtDNA-depleted MEFs display peri-mitochondrial ADA-like filaments, sensitive to CK666 treatment (Fig. EV5C). Next, we evaluated ER-mitochondrial proximity in these cells through PLA assay using antibodies against VAP-B (ER) and Tom20 (mitochondria). After 5 days of EtBr treatment, we detected a twofold reduction in the PLA density compared to that of control cells (Fig. 7A,B). This reduction is completely rescued by CK666 treatment (Fig. 7A,B). To further confirm these results we carried out ERMC evaluation through thin section EM. Similar to results obtained through PLA assay, EtBr treated MEF cells showed significant reduction in the amount of close contacts between ER and mitochondria which were completely rescued upon CK666 treatment (Figs. 7C,D and EV5D). The minimum distance between ER and mitochondria was also seen to be significantly increased in the EtBr treated cells which were rescued by CK666 treatment (Fig. EV5E). These results suggest that ADA-like peri-mitochondrial actin filaments disrupt ERMC downstream of mtDNA depletion mediated mitochondrial dysfunction.

We also transfected these cells with GFP-Parkin or GFP-LC3 to evaluate their localization during chronic mitochondrial dysfunction. Surprisingly, EtBr-treated cells show little mitochondrial accumulation of both GFP-Parkin (Fig. 7E,F) and GFP-LC3 (Fig. 7G,H), in spite of these mitochondria being chronically depolarized. Interestingly, CK666 treatment causes a significant increase in both Parkin-positive (Fig. 7E,F) and LC3-positive (Fig. 7G,H) mitochondria in EtBr treated samples but not in the control samples. The fact that depletion of ADA from these EtBr treated cells resulted in increased Parkin and LC3 recruitment at the mitochondria, we questioned whether ADA prevented mitophagy in these cells. To read out actual mitophagy signals, we transiently transduced MEF cells with a previously characterized tandem reporter construct consisting of GFP-RFP targeted to OMM using a Fis1 targeting sequence (Delgado et al, 2024) and induced chronic mitochondrial damage through EtBr treatment. While EtBr treatment showed a significant increase in the number of Red only (mitolysosome) puncta (Fig. 7I,J). CK666 treatment showed a significant increase in mitolysosome density in the EtBr treated cells but had no effect on the control cells (Fig. 7I,J). These results suggest that ADA-like filaments, assembled during chronic mitochondrial damage, disrupt ERMC and not only block

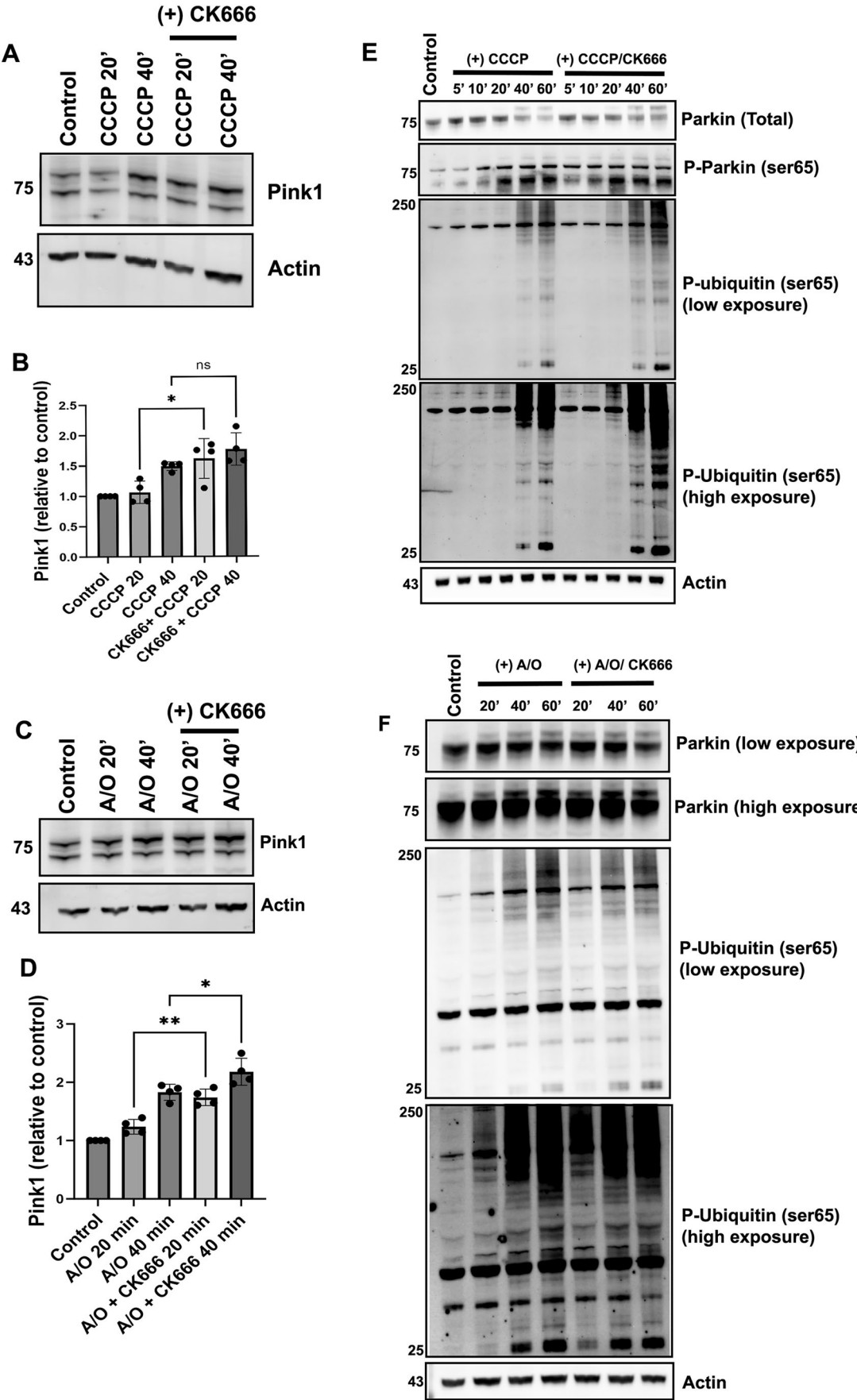

**Figure 6.  ADA regulates stabilization and activity of Pink1 following mitochondrial depolarization.**

(A) Western blot showing Pink1 levels in HeLa cells overexpressing GFP-Pink1 treated with 20 µM CCCP for 20 and 40 min in the presence or absence of 100 µM CK666. Actin used as loading control. Molecular weight in kDa. (B) Quantification showing relative Pink1 levels in HeLa cells overexpressing GFP-Pink1 following 20 and 40 min of CCCP treatment in the presence or absence of CK666 (100 µM). $N = 3$ independent experiments; Error: Mean ±s.d.; $P = 0.0145$ (CCCP 20 Vs CK666 + CCCP 20) (*), $P = 0.3731$ (CCCP 40 vs CK666 + CCCP 40) (ns), one-way ANOVA used. (C) Western blot showing Pink1 levels in HeLa cells overexpressing GFP-Pink1 treated with a combination of 15 µM Oligomycin A and 25 µM Antimycin A for 20 and 40 min in the presence or absence of 100 µM CK666. Actin used as loading control. Molecular weight in kDa. (D) Quantification showing relative Pink1 levels in HeLa cells overexpressing GFP-Pink1 following 20 and 40 min of Oligomycin A + Antimycin A treatment in the presence or absence of CK666 (100 µM). $N = 3$ independent experiments; Error: Mean ±s.d.; $P = 0.0017$ (A/O 20 Vs CK666 + A/O 20) (**), $P = 0.0291$ (A/O 40 vs CK666 + A/O 40) (*), one-way ANOVA used. (E) Western blot of HeLa cells overexpressing Parkin GFP showing Parkin, phospho-Parkin (S65) and phospho-ubiquitin (S65) upon treatment with 20 µM CCCP in the presence or absence of CK666 (100 µM) for various time points as shown. Molecular weight in kDa. (F) Western blot of HeLa cells overexpressing Parkin GFP showing Parkin, phospho-Parkin (S65) and phospho-ubiquitin (S65) upon treatment with a combination of 15 µM Oligomycin A and 25 µM Antimycin A in the presence or absence of CK666 (100 µM) for various time points as shown. Molecular weight in kDa. Source data are available online for this figure.

recruitment of Parkin and LC3 onto these damaged mitochondria but also slow down mitophagy in these cells. Removal of these ADA-like filaments restores ERMC and allows Parkin and LC3 recruitment followed by mitophagy of these damaged mitochondria.

## Increasing the importance of mitochondrially-mediated ATP production causes prolonged ADA and delayed Parkin recruitment

The studies described above were conducted using standard culture media, which is hyperglycemic (25 mM glucose) compared to physiological glucose concentrations. Thus, cells cultured in this media—similar to cancer cells—mostly rely on aerobic glycolysis to meet their ATP demands (Bensinger and Christofk, 2012), make biosynthetic precursors (Bartman et al, 2023; Ryu et al, 2024; Shen et al, 2024) and replenish $NAD^+$ (Luengo et al, 2021). Therefore, we wondered whether culturing cells under conditions requiring mitochondrial ATP production would affect ADA. We cultured HeLa cells for 10 days either in the presence of 10 mM glucose or in 10 mM galactose, the latter being known to cause a dependence on mitochondrial OxPhos for ATP production (MacVicar and Lane, 2014). Cells grown on glucose exhibit lower oligomycin-sensitive oxygen consumption rate (OCR) and higher basal extracellular acidification rate (ECAR) than galactose-grown cells (Fig. EV5F). In addition, the maximal OCR for galactose-fed cells is higher than for those grown in glucose (Fig. EV5F), suggesting that mitochondrial OxPhos is up-regulated in the absence of extracellular glucose. Mitochondrial polarization is similar in glucose- and galactose-primed HeLa cells (Fig. EV5G), suggesting that mitochondria maintain appreciable $\Delta\psi_m$ in both cases.

Next, we compared ADA kinetics in the two conditions. Fixed-cell analysis evaluating the number of HeLa cells displaying ADA at different time points following CCCP stimulation showed a significantly higher proportion of ADA-positive cells at 5 and 10 min post-CCCP treatment in galactose-primed cells compared to glucose (Fig. 8A,B).

We next compared CCCP-induced Parkin recruitment and downstream LC3 accumulation in glucose- and galactose-primed cells. Galactose-primed HeLa cells display a reduced number of cells containing mitochondrially localized GFP-Parkin following 30 min and 60 min of CCCP treatment, and this reduction is rescued by Arp2 KD. Similarly, mitochondrial levels of LC3 remain low after 90 min of CCCP in galactose primed cells (Fig. 8C,D).

Both Arp2/3 inhibition and overexpression of ER-RFP-mito significantly increased mitochondrial LC3 recruitment (Fig. 8E,F). Altogether, our data support a model whereby prolonged ADA induced by galactose-primed cells in turn results in a prolonged delay of Parkin and LC3 recruitment upon mitochondrial damage.

## Discussion

Collectively, the results presented in this study show that ADA has a significant effect on the kinetics of both Parkin recruitment and Pink1 stabilization following mitochondrial damage. Acute mitochondrial depolarization generates a transient wave of ADA, which significantly delays the recruitment of Parkin onto depolarized mitochondria. Chronic mitochondrial dysfunction results in persistent ADA-like filaments that block Parkin recruitment, with actin depolymerization allowing rapid Parkin recruitment and subsequent mitophagy of dysfunctional mitochondria. Interestingly, we see that the inhibition of ADA also stimulates faster stabilization and activity of Pink1 upon acute mitochondrial damage. Actin filaments do not merely form a barrier to prevent access but instead reduce ER-mitochondrial contacts. Overexpression of ER-mitochondrial tethers overcomes this ADA-induced block and accelerates depolarization-induced Parkin assembly in both the acute and chronic models of mitochondrial dysfunction. Our results also suggest that increasing the dependence of cells on OxPhos-derived ATP prolongs ADA, in turn causing prolonged suppression of Pink1/Parkin mediated mitophagy.

Our results show that both pharmacological intervention (through CK666) and genetic manipulation (through Arp2 KD, WRC KD and FMNL KD) significantly speed up Parkin recruitment upon acute mitochondrial damage. One likely explanation might be that these intervention through eradication of ADA induces an increased mitochondrial stress when coupled with CCCP treatment. However, the fact that modulation of ERMC either by increasing them (VAPB OE, ER-RFP-mito OE) or by depleting them (Mfn2 KD, VDAC1 KD and PM-RFP-Mito OE) significantly modifies Parkin recruitment in the presence of ADA suggests that modulation of ERMC has a profound effect on the recruitment kinetics of Parkin. This being said, modulation of ERMC either through synthetic tethers or depletion of specific molecules could induce additional mitochondrial stress that might have its own effect on the dynamics of Parkin recruitment to dysfunctional mitochondria and additional studies are specifically needed to further address this issue.

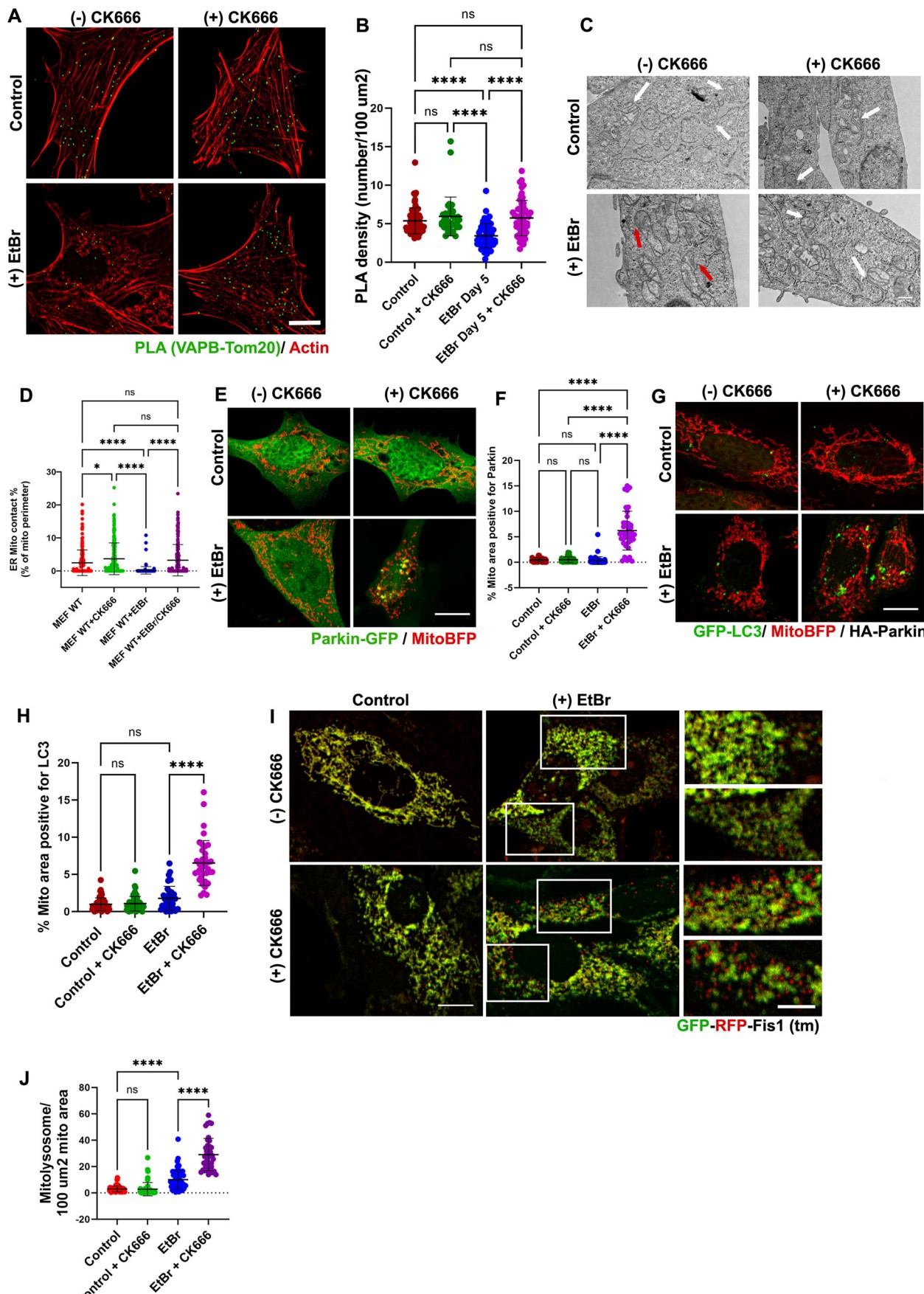

**Figure 7.  Depletion of ADA-like filaments in chronically damaged mitochondria re-establishes ERMC resulting in Parkin and LC3 recruitment followed by mitophagy.**

(A) Representative fixed-cell images from Proximity Ligation Assay (PLA; green) conducted between VAPB (ER) and Tom20 (mitochondria) to assess ERMC in control and EtBr treated (0.2 µg/ml for 5 days) MEFs treated with or without CK666 (100 µM/ 2 h). Cells were stained with Phalloidin-488 to label actin filaments prior to PLA staining. Scale: 10 µm. (B) Scatter plot representing density of PLA dots (no. per 100 µm²) in control and EtBr-treated HeLa cells with or without CK666 treatment. Same data set as (A). Control: 65 cells (without CK666); 35 cells (with CK666). +EtBr: 57 cells (without CK666) and 55 cells (with CK666). Each conditions have 20–25 fields from three independent experiments. $P = 0.0001$ (****); Error: ±s.d. One-way ANOVA used. (C) Representative images from thin-section TEM conducted in control and EtBr-treated (0.2 µg/ml for 5 days) MEFs treated with or without CK666 (100 µM/ 2 h) showing ER and mitochondria. The white arrows represent close contacts between ER and Mitochondria and the red arrows indicate decreased contacts. Scale: 500 nm. (D) Scatter plots showing the percentage of mitochondrial perimeter within 0–40 nm distance to the ER. Same dataset as (C). Controls without CK666 (Red; $n = 62$ mitochondria); with CK666 (green; $n = 63$ mitochondria). +EtBr without CK666 (purple; $n = 63$ mitochondria) and +EtBr with CK666 (blue; $n = 65$ mitochondria). Three independent experiments for each condition. $P = 0.0157$ (*); $P = 0.0001$ (****); one-way ANOVA analysis; Error: Mean ±s.d. (E) Representative fixed-cell images from control and EtBr treated (0.2 µg/ml for 5 days) MEF cells transfected with Parkin-GFP (green) and mitoBFP (red), treated with or without CK666 (100 µM/ 4 h). Scale: 10 µm. (F) Quantification showing Parkin positive mitochondria area (%) in each cell from dataset shown in (E). Controls: 42 cells (without CK666); 40 cells (with CK666); +EtBr: 58 cells (without CK666) and 40 cells (with CK666) from 25–30 fields and 3 independent coverslips/fixations. $P = 0.0001$ (****), one-way ANOVA used; Error: Mean ±s.d. (G) Representative fixed-cell images from control and EtBr treated (0.2 µg/ ml for 5 days) MEF cells transfected with LC3-GFP (green), mitoBFP (red) and HA-Parkin, treated with or without CK666 (100 µM/ 4 h). Scale: 10 µm. (H) Quantification showing LC3-positive mitochondria area (%) in each cell from images in 7 G. Control: 54 cells (without CK666); 61 cells (with CK666). + EtBr: 40 cells (without CK666) and 38 cells (with CK666) from 20–25 fields and 3 independent experiments/coverslips. $P = 0.0001$ (****), one-way ANOVA used; Error: Mean ±s.d. (I) Representative fixed-cell images from control and EtBr treated (0.2 µg/ml for 5 days) MEF cells transiently transduced with GFP-RFP-Fis1(tm) mitophagy reporter, treated with or without CK666 (100 µM/ 5 h). Scale: 10 µm. (J) Quantification showing number of RFP only puncta (mitolysosome) per cell area in each cell from images in (I). Control: 52 cells (without CK666); 64 cells (with CK666). + EtBr: 66 cells (without CK666) and 41 cells (with CK666) from three independent experiments. $P = 0.0001$ (****), one-way Anova used. Error: Mean ±s.d.

Actin filaments have diverse effects on ER-mitochondria crosstalk. Previously we have shown that linear actin filaments polymerized on the ER, generated by the formin INF2, promote ERMCs, thereby stimulating ER-to-mitochondrial calcium transfer and mitochondrial division (Chakrabarti et al, 2018). Here, we show that a different form of actin filaments, polymerized by the Arp2/3 complex around dysfunctional mitochondria, acutely disrupts ERMCs. Actin filaments, therefore, play contrasting roles in regulating ERMCs, dependent on the cellular context. The mechanism by which ADA disrupts ERMCs is unclear at present. Arp2/3-generated actin filament networks are known to generate force in many cellular contexts, such as at lamellipodia during cell migration (Krause and Gautreau, 2014) and in comet tails propelling intracellular pathogenic bacteria such as *Listeria* and *Shigella* (Rutenberg and Grant, 2001; Welch and Way, 2013). More recently, Arp2/3-mediated actin waves and comet tails around mitochondria during mitosis have been shown to regulate mitochondrial motility and distribution (Moore et al, 2021). A similar mechanism might be used by ADA to push ER and mitochondria apart. While the directionality of the growing actin filaments in ADA is still not clear, considering Arp2/3 recruitment to the mitochondrial membrane (Fung et al, 2022), the force is likely to be directed toward the mitochondrion.

It is fascinating to note that distinct actin filament populations assemble around depolarized mitochondria at different times and for different purposes (Fung et al, 2023). While acute mitochondrial depolarization results in ADA within minutes, a second wave of actin occurs 1–2 h post-depolarization, which we call PDA (prolonged depolarization-induced actin) (Kruppa et al, 2018). Both ADA and PDA are dependent on Arp2/3 but are distinct in the other proteins involved (WAVE Regulatory Complex and FMNL formins for ADA; N-WASP, myosin VI, and Parkin for PDA). Interestingly PDA appears to facilitate mitophagy in two ways: by preventing refusion of damaged mitochondria with healthy ones (Kruppa et al, 2018) and by dispersing clumps of damaged mitochondria (Hsieh and Yang, 2019). Finally, actin filaments are involved in several later steps of autophagy through a variety of Arp2/3 complex activators, including WHAMM (Kast and Dominguez, 2015), JMY (Hu and Mullins, 2019), and WASH (King et al, 2013). It is unclear whether PDA or autophagy-associated actin polymerization has effects on ERMCs.

The precise role of ERMCs in Parkin-mediated mitophagy is still unclear. Interestingly, we show that Parkin is recruited in the vicinity of ER and mitochondria and surprisingly a significant portion of Parkin puncta was exclusively on ER in both HeLa cells and MEFs. This raises the possibility that the extent of ERMC is positively correlated to Parkin recruitment. Two promising candidates that might be mediating this process are VDACs and Mfn2, both of which were shown to regulate not only ERMC (Chen et al, 2003; de Brito and Scorrano, 2008; Hu et al, 2021; Naon et al, 2016) but also Parkin recruitment (Chen and Dorn, 2013; Sun et al, 2012). In fact a progressive reduction of Mfn2, correlating with a reduction in mitophagy and accumulation of damaged mitochondria, has been linked to sarcopenia (Sebastián et al, 2016). Our results revealed that depletion of both VDAC1 and Mfn2 individually in HeLa cells downregulate Parkin recruitment following CCCP treatment. However, increasing ERMC through ER-RFP-Mito synthetic tether could significantly rescue Parkin recruitment defect in Mfn2-depleted cells but not VDAC1-depleted cells. These results demonstrate that while both Mfn2 and VDAC1 regulate Parkin recruitment, Mfn2 operates in an ERMC dependent manner while VDAC in an ERMC-independent pathway. It is still unclear whether VDAC1 regulates Parkin recruitment directly or mediates it via some other molecule. Indeed, hexokinase activity has been shown to be important for Parkin recruitment (McCoy et al, 2014) which is tethered to the OMM through its interaction with the VDACs (Bieker et al, 2025). However, the role of ERMCs in mitophagy might be more varied. While the early stages of mitophagy might require an actual loosening of ERMCs to provide space for recruitment of the phagophore membranes, some ERMCs might be retained as a source of phospholipid for the growing phagophore (Gelmetti et al, 2017; Hamasaki et al, 2013).

Our results further indicate a possible role of ADA in Pink1 stabilization and/or activity. Indeed we see that Pink1 is

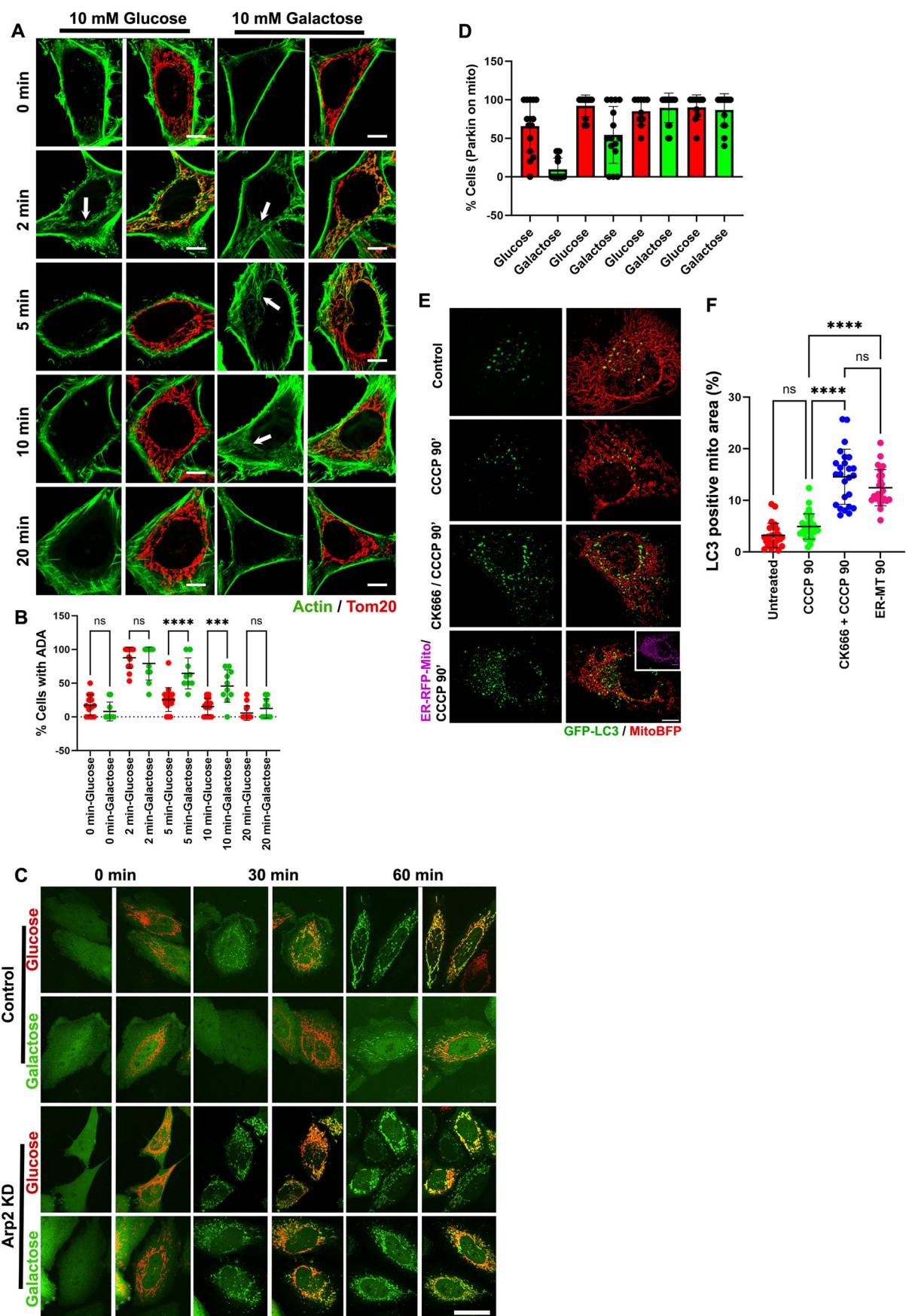

**Figure 8. OxPhos-dependent cells display sustained ADA and inhibited Parkin recruitment.**

(A) Representative images of HeLa cells cultured in either 10 mM glucose or galactose for 10 days, transfected with mitoBFP (mitochondria, red), treated with 20 μM CCCP at time 0 and fixed at various time points. Cells were stained with rhodamine-phalloidin to show actin filaments (green). Scale: 10 μm. Arrows denote actin assembly around mitochondria. (B) Quantification showing percentage of cells with ADA in HeLa cells cultured in glucose or galactose following 20 μM CCCP treatment at time 0, fixed and stained for actin (rhodamine-phalloidin, green) and mitochondria (Tom 20, red) at various time points. Same dataset as (A). Three independent experiments were performed, having 40–80 cells in each condition. Error: ±s.d.; $P = 0001$ (****) for 5 min Glucose vs 5 min Galactose and $P = 0.0009$ (***) for 10 min Glucose vs 10 min Galactose; one-way ANOVA used. (C) Representative images of WT and Arp2 KD HeLa cells cultured in 10 mM glucose or galactose showing Parkin-GFP (green) and mitoBFP (mitochondria, red) treated with 20 μM CCCP and fixed at various time points. Scale: 10 μm. (D) Quantification showing percentage of cells with Parkin positive mitochondria from images in (C). Three independent experiments were performed, having 45–60 cells in each condition. $P = 0.0443$ (*) and 0.0001 (****), unpaired $t$ test used. Error: Mean ±s.d. (E) Representative images of HeLa cells cultured in either 10 mM glucose or galactose for 10 days treated with CCCP (20 μM/ 90 min) with or without CK666 (100 μM/ 90 min) in the presence or absence of ER-RFP-mito (ER-mito linker). The cells were transiently transfected with GFP-LC3 (green), HA-Parkin and mitoBFP (red). Scale bar: 10 μm. (F) Quantification for HeLa cells showing LC3-positive mitochondria area (%) in each condition as described in 8E. Control (untreated): 22 cells; CCCP-treated: 32 cells; CK666 + CCCP: 27 cells; ER-RFP-Mito linker with CCCP: 26 cells. Data from three independent experiments. $P = 0.0001$ (****). One-way ANOVA used. Error: Mean ±s.d.

stabilized faster in the absence of ADA, but it remains to be seen whether this is ERMC dependent as well. In fact Pink1 has been shown to accumulate at ERMC following valinomycin or CCCP that allows BECN1 recruitment at the contact sites for later stages of mitophagy (Gelmetti et al, 2017). Further work is needed to elucidate whether ER localized Pink1 is transferred to mitochondrial surface upon mitochondrial damage and whether ADA selectively resists stabilization of Pink1 at these ERMC microdomains.

We show that cells cultured in galactose-containing media display prolonged ADA as well as a extended delay in Parkin recruitment upon depolarization, when compared to cells cultured in high glucose media. This result suggests that cells adapt to protect mitochondria once their ATP-producing function becomes essential. Moreover, we report that this delay in Parkin recruitment is eliminated by Arp2/3 inhibition, suggesting that the duration and persistence of peri-mitochondrial actin filaments regulates Parkin recruitment in this situation. Interestingly, studies have shown that OxPhos-dependent cells display impairment in OMA-1 dependent OPA-1 processing(MacVicar & Lane, 2014), depolarization-induced Parkin recruitment (Van Laar et al, 2011) and mitophagy (MacVicar and Lane, 2014). These impairments are reversed by culturing cells in glycolytic media conditions (Van Laar et al, 2011). Moreover, OxPhos-dependent cortical neurons do not accumulate Parkin upon acute mitochondrial damage (Van Laar et al, 2011). Coupled with our previous results showing that ADA inhibits OMA-1 dependent OPA-1 processing (Fung et al, 2022; Fung et al, 2019), our present studies suggest that peri-mitochondrial actin assembly might be a key factor regulating early steps after acute mitochondrial dysfunction.

# Methods

### Reagents and tools table

| Reagent/resource | Reference or source | Identifier or catalog number |
|---|---|---|
| **Experimental models cell lines** | | |
| Human U2-OS | ATCC | HTB-96 |
| Human HeLa | ATCC | CCL-2 |
| Mouse MEF | Losón et al, 2013 | N/A |

| Reagent/resource | Reference or source | Identifier or catalog number |
|---|---|---|
| **Recombinant DNA** | | |
| GFP-F-tractin | Johnson and Schell, 2009 | N/A |
| Mito-BFP | Korobova et al, 2013 | N/A |
| ER-tag-RFP | E. Snapp, Albert Einstein College of Medicine, NY | N/A |
| Myc-VAP-B | Gomez-Suaga et al, 2017 | N/A |
| pLAMP1-mCherry | Amy Palmer | Addgene: #45147 |
| Cyto-CFP-FRBx5 | Takanari Inoue | Addgene: #103776 |
| pEGFP-Parkin-WT | Edward Fon | Addgene: #45875 |
| pcDNA-DEST47 PINK1 C-GFP | Mark Cookson | Addgene: #13316 |
| pMXs-IP HA-Parkin | Noboru Mizushima | Addgene: # 38248 |
| EGFP-LC3 | Karla Kirkegaard | Addgene: #11546 |
| AKAP1-YFP-FKBP | Gyorgy Haznoczky, Thomas Jefferson University, Philadelphia | N/A |
| ER-RFP-Mito | Gyorgy Haznoczky, Thomas Jefferson University, Philadelphia | N/A |
| PM-RFP-Mito | Gyorgy Haznoczky, Thomas Jefferson University, Philadelphia | N/A |
| mtGCampf6 | Gyorgy Haznoczky, Thomas Jefferson University, Philadelphia | N/A |
| RFP-GFP-tf(Fis1) reporter system | Christopher Shoemaker, Geisel School of Medicine, Dartmouth, NH | N/A |
| **Antibodies** | | |
| Tom-20 | Abcam | Cat# ab78547 |
| VAP-B | Proteintech | Cat# 66191-1-Ig |
| PTPIP51 | Proteintech | Cat# 20541-1-AP |
| anti-myc-tag | Abcam | Cat# ab32 |
| INF2 | Cell Signaling Technology | Cat# 41081 |
| VAP-B | Proteintech | Cat# 14477-1-AP |
| LC3B | Cell Signaling Technology | Cat# 12741S |
| Parkin | Cell Signaling Technology | Cat# 2132S |

| Reagent/resource | Reference or source | Identifier or catalog number |
|---|---|---|
| Parkin (phosphoS65) | Abcam | Cat# 315376 |
| Pink1 | Cell Signaling Technology | Cat# 6946S |
| B-Actin | Cell Signaling Technology | Cat# 3700S |
| MFN2 | Cell Signaling Technology | Cat# 9482S |
| VDAC | Invitrogen | Cat# MA5-41088 |
| Phosphor-Ubiquitin S65 | Cell Signaling Technology | Cat# 60802S |
| HRP-conjugated anti-mouse IgG | Bio-Rad | Cat#1705047 |
| HRP-conjugated anti-rabbit IgG | Bio-Rad | Cat#1706515 |
| Alexa-Fluor coupled 488 anti-rabbit | Invitrogen | Cat# A11008 |
| Alexa-Fluor coupled 647 anti-rabbit | Invitrogen | Cat# A21245 |
| **Oligonucleotides and other sequence-based reagents** | | |
| si-FMNL1 | IDT | hs.FMNL1.13.5 |
| si-FMNL2 | IDT | hs.FMNL1.13.1 |
| si-FMNL3 | Ambion | s-40551 |
| si-Arp2 | IDT (Custom synthesized) | N/A |
| si-MFN2 | IDT | hs.Mfn2.13.1 |
| si-VDAC1 | IDT | hs.VDAC1.13.1 |
| Negative Control | IDT | 51-01-14-04 |
| RT-PCR primers | This study | Materials section |
| **Chemicals, enzymes and other reagents** | | |
| DMEM | Corning | Cat# 10013-CV |
| DMEM | Gibco | Cat# 21063-029 |
| Fetal Bovine Serum | Sigma-Aldrich | Cat# F4135 |
| OPTI-MEM | Gibco | Cat# 31985062 |
| Lipofectamine 2000 | Invitrogen | Cat# 11668 |
| Lipofectamine RNAi max | Invitrogen | Cat# 13778 |
| ECL Prime Western Blotting Detection Reagent | GE healthcare | Cat# 28980926 |
| Polyvinylidene fluoride membrane | EMD Millipore | Cat# IPFL00010 |
| Glutaraldehyde | Electron Microscopy Sciences | Cat# 16020 |
| PFA | Electron Microscopy Sciences | Cat# 15710 |
| Laemmli sample buffer | Bio-Rad | Cat#161-0747 |
| ProLong Gold antifade mounting media | Invitrogen | Cat# P36930 |
| DMSO | Invitrogen | Cat# D12345 |
| CK666 | Sigma-Aldrich | Cat# SML0006 |
| CCCP | Sigma-Aldrich | Cat# C2759 |

| Reagent/resource | Reference or source | Identifier or catalog number |
|---|---|---|
| FCCP | MedChemExpress | Cat# 100410 |
| TMRE | Sigma-Aldrich | Cat# 87917 |
| Antimycin A | Sigma-Aldrich | Cat# A8674 |
| Oligomycin A | Sigma-Aldrich | Cat# 75351 |
| Rotenone | Sigma-Aldrich | Cat# R8875 |
| Rapamycin | Sigma-Aldrich | Cat# R0395 |
| Histamine | Sigma-Aldrich | Cat# H7250 |
| Ethidium Bromide | Sigma-Aldrich | Cat# E7637 |
| Glucose | Sigma-Aldrich | Cat# 49163 |
| Galactose | Sigma-Aldrich | Cat# 1287700 |
| Duolink in situ Red Mouse/Rabbit starter kit | Sigma-Aldrich | Cat# DUO92101 |
| Phalloidin-AlexaFluor 488 | Invitrogen | Cat# A12379 |
| MycoAlert Plus Mycoplasma detection kit | ATCC | Cat# 30-1012K |
| Universal Mycoplasma detection kit | Lonza | Cat# LT07-701 |
| **Software** | | |
| Fiji Image Analysis | ImageJ | https://imagej.net/Fiji |
| Prism version 10.1.0 | GraphPad | https://www.graphpad.com/scientificsoftware/prism/ |
| Office Excel | Microsoft | https://www.microsoft.com/en-gb/ |
| Fusion | Andor | version 2.0.0.15 |
| NIS-Elements | Nikon | Version 4.0 |
| Zen Black | Zeiss | N/A |
| **Other** | | |
| Glass-bottom imaging dish | MatTek | Cat#P35G-1.5-14-C |

## Cell culture

Wild-type human osteosarcoma U2OS and human cervical cancer HeLa cells were procured from American Type Culture Collection (ATCC). Mouse embryonic fibroblasts (MEF) were a gift from David Chan and is described elsewhere (Losón et al, 2013). All cells were grown in DMEM (Corning, 10-013-CV) supplemented with 10% newborn calf serum (Hyclone, SH30118.03) for U2OS or 10% fetal bovine serum (Sigma-Aldrich F4135) at 37 °C with 5% $CO_2$. Cell lines were tested every 3 months for mycoplasma contamination using Universal Mycoplasma detection kit (ATCC, 30-1012 K) or MycoAlert Plus Mycoplasma Detection Kit (Lonza, LT07-701).

## DNA transfections, plasmids, and siRNA

For plasmid transfections, cells were seeded at $4 \times 10^5$ cells per well in a 35 mm dish at ~16 h before transfection. Transfections were performed in OPTI-MEM medium (Gibco, 31985062) using lipofectamine 2000 (Invitrogen, 11668) as per manufacturer's protocol, followed by trypsinization and re-plating onto glass bottomed dishes (MatTek Corporation, P35G-1.5–14-C) at ~$1 \times 10^5$ cells per well (for live cell imaging) or coverslips (10 mm or 25 mm) for fixed cell analysis. Cells were imaged ~16–24 h after transfection except for VAP-B over-expressing cells which were imaged within 8–12 h after transfections.

GFP-F-tractin plasmid was a gift from C. Waterman and A. Pasapera (National Institutes of Health, Bethesda, MD) and were on a GFP-N1 backbone (Clonetech), as described previously (Johnson and Schell, 2009). mito-BFP (GFP-N1 backbone) constructs were previously described (Korobova et al, 2013) and consist of amino acids 1–22 of *S. cerevisiae* COX4 N terminal to the respective fusion protein. ERtagRFP (modified GFP-N1 backbone) was a gift from E. Snapp (Albert Einstein College of Medicine, New York, NY), with prolactin signal sequence at 5′ of the fluorescent protein and KDEL sequence at 3′. Myc-VAP-B was a gift from C.C.J. Miller (King's College, London, UK) and described elsewhere (Gomez-Suaga et al, 2017). pLAMP1-mCherry was a gift from Amy Palmer (Addgene plasmid #45147). cyto-CFP-FRBx5 was a gift from Takanari Inoue (Addgene plasmid # 103776). pEGFP-parkin WT was a gift from Edward Fon (Addgene plasmid #45875). pcDNA-DEST47 PINK1 C-GFP was a gift from Mark Cookson (Addgene plasmid #13316). pMXs-IP HA-Parkin was a gift from Noboru Mizushima (Addgene plasmid # 38248). EGFP-LC3 was a gift from Karla Kirkegaard (Addgene plasmid #11546). AKAP1-YFP-FKBP, ER-RFP-Mito, PM-RFP-Mito and mtGCampf6 constructs were a kind gift from Gyorgy Haznoczky (Thomas Jefferson University, Philadelphia, USA).

The following amounts of DNA were transfected per well (individually or combined for co transfection): 500 ng for mito–BFP, GFP–F-tractin, ER-RFP-Mito, PM-RFP, Mito, mtGCampf6, pcDNA-DEST47 PINK1 C-GFP and pLAMP1-mCherry; 100 ng for pEGFP-parkin WT, pMXs-IP HA-Parkin and AKAP1-YFP-FKBP; 250 ng for GFP-LC3; 800 ng for ER-RFP; 750 ng for Cyto-CFP-FRBX5; 400 ng for myc-VAP-B.

For all siRNA transfections, $1 \times 10^5$ cells were plated onto a 35 mm dish and 2 μl RNAimax (Invitrogen, 13778) with 63 pg siRNA were used per well. Cells were analyzed 72–96 h post siRNA transfection. For Arp2 siRNA transfections, $1 \times 10^5$ cells were plated directly onto glass-bottomed dishes (MatTek Corporation, P35G-1.5-14-C) or coverslips and 2 μl RNAimax (Invitrogen, 13778) with 63 pg siRNA were used per dish. Cells were analyzed 48 h for Arp2. For live-cell imaging, plasmids containing fluorescent markers were transfected into siRNA-treated cells 18–24 h prior to imaging, as described above.

siRNAs:

human FMNL1(IDT,hs.Ri.FMNL1.13.5: 5′-GTGGTACATTCGGTGGATCATGTTCTCCACCGAAT -3′.

FMNL2 (IDT, hs.Ri.FMNL2.13.1, 5′-CATGATGCAGTTTAGTAA-3′).

FMNL3 (Ambion, s40551, 5- GCATCAAGGAGACATATGA-3′).

Arp2 (IDT, custom synthesized, HSC.RNAI.N001005386.12.6, 5′-GGAUAUAAUAUUGAGCAAGAGCAGA-3′).

Human Mfn2 (IDT, hs.Ri.Mfn2.13.3: 5′-CCCUUUCUGAAA-GAAGUAUGGCCAA-3′).

Human VDAC1 (IDT, hs.Ri.VDAC1.13.1: 5′-GUGACAACA-CUCAGAAUCUAAAUG-3′) and negative control (IDT, #51-01-14-04, 5′-CGUUAAUCGCGUAUAAUACGCGUAU-3′).

## Immunofluorescence

Cells (untransfected or transfected) were plated onto coverslips 16 h prior to fixation and staining. Following respective treatments, Cells were then fixed with either pre-warmed 4% PFA (20 min) or 1% glutaraldehyde (10 min) at room temperature, enabling optimal preservation of the actin cytoskeleton. Cells were washed with PBS and glutaraldehyde fixed cells were additionally washed with NaBH4 (1 mg/ml; 3 × 15 min each). Cells were then permeabilized with either 0.1% Triton X-100 for 1 min (PFA fixed) or 0.25% Triton X-100 for 10 min (glutaraldehyde fixed) and again washed with PBS three times. Prior to antibody staining, cells were blocked with 10% FBS in PBS for ~30 min. Primary antibody was diluted in 1% FBS/PBS and coverslips were incubated on a drop of antibody solution on parafilm in a wet chamber for 1 h. Cells were then washed six times with 1× PBS and appropriate secondary antibody was mixed with phalloidin (AF-488 or Rhodamine phalloidin) in 1% FBS/PBS was added to the coverslips and incubated for 1 h. Coverslips were again washed six times with 1× PBS and mounted on glass slides using ProLong Gold antifade mounting media (Invitrogen #P36930).

Primary antibodies: anti-Tom 20 (Abcam ab78547) used at 1:400.

Secondary antibody: AlexaFluor 488-coupled anti-rabbit (Invitrogen #A11008), AlexaFluor 647-coupled anti-rabbit (Invitrogen #A21245)used at 1:500.

Rhodamine Phalloidin (Invitrogen R415) used at 1:500; Alexa-Fluor 488-phalloidin (Invitrogen A12379) used at 1:500.

## Proximity ligation assay

Duolink™ PLA technology was used to study the ER-Mitochondrial interaction. Duolink™ In Situ Red Starter Kit Mouse/Rabbit (MilliporeSigma, cat# DUO92101) was used. In brief, cells were cultured on 10 mm coverslips for 16 h before drug treatments. Following respective drug treatments cells were fixed with prewarmed 4% PFA for 20 min at RT and washed with PBS (X three times). Cells were then permeabilized using 0.1% Triton-X-100 for 1 min, washed with PBS and incubated with AF-488-Phalloidin (diluted in 1% FBS/PBS) for 1 h in dark. Cells were then washed with PBS and incubated with Duolink blocking buffer for 1 h and incubated with primary antibodies (diluted in Duolink antibody diluent) for 2 h at RT.

Primary antibodies used: anti-VAPB (Proteintech 66191-1-Ig), anti-PTPIP51 (Proteintech 20641-1-AP) and anti-Tom20 (Abcam ab78547). Cells were then washed with 5% BSA (15 min × 3) and incubated with Duolink secondary antibodies for 1 h at 37 °C. Cells were then washed with Duolink wash buffer A (5 min × 3) and incubated with Duolink ligase for 30 min at 37 °C. Cells were further washed with Duolink wash buffer A (2 min × 3) and incubated with Duolink polymerase for 90 min at 37 °C. The cells were finally washed with Duolink wash buffer B (10 min × 3) and

mounted on glass slides using prolong gold antifade reagent. Images were acquired within 24–48 h.

## Mitochondrial staining

### TMRE

Cells were plated onto coverslips at a dilution of 200,000 cells per coverslips and incubated overnight in regular cell culture media. The following day, cells were stained with 30 nM TMRE solution prepared in regular DMEM cell culture media for 15 min at 37 °C incubator with 5% $CO_2$. Cells were then washed twice with 1× PBS and fresh DMEM cell culture media was added to them. These were then imaged on a Nikon Confocal scope to get the baseline mitochondrial fluorescence (for glucose and galactose-treated cells), or treated with CCCP or Antimycin/Oligomycin to generate TMRE traces.

### MitoBrilliant 646 staining

WT and KD cells were seeded onto coverslips at a dilution of 200,000 cells and incubated overnight in regular cell culture media. The following day cells were stained with 500 nM MitoBrilliant (TOCRIS 7700) for 30 min, washed twice with 1× PBS and resuspended in regular DMEM media for downstream experiments.

## Microscopy

Microscopy of fixed samples was performed on LSM 880 equipped with 63×/1.4 NA plan Apochromat oil objective using the Airyscan detectors (Carl Zeiss Microscopy). The Airyscan uses a 32-channel array of GaAsP detectors configured as 0.2 airy units per channel to collect the data that is subsequently processed using Zen2 software. Cells were imaged with the 405-nm laser and 450/30 filter for BFP, 488 nm and 525/30 for GFP, and 561 nm and 595/25 for RFP. Images for Figs. 2G, 3D,F, 5A, 7A,E,G,I, 8E, EV3E, and EV5C were acquired using this system and quantified using ImageJ.

Images for Figs. 2A,B,E, 4B, EV2B,D, and EV3D were acquired using a Nikon A1 on a Nikon Ti-E base and equipped with an iXon Ultra 888 EMCCD camera. A solid-state 405 smart diode 100-mW laser, solid-state 488 OPSL smart laser 50-mW laser, solid-state 560 OPSL smart laser 50-mW laser, and solid-state 637 OPSL smart laser 140-mW laser were used (objective: 60 × 1.45 NA CFI Plan Apo; Nikon). Images were acquired using Nikon Elements v5.02. Images acquired using this system and quantified using ImageJ.

Live cell imaging was conducted in DMEM (Gibco, 21063-029) with 25 mM D-glucose, 4 mM L-glutamine and 25 mM HEPES, supplemented with 10% newborn calf serum, hence referred to as "live cell imaging media". Cells ($\sim 3.5 \times 10^5$) were plated onto MatTek dishes 16 h prior to imaging. Medium was preequilibrated at 37 °C and 5% CO2 before use. Dishes were imaged using the Dragonfly 302 spinning disk confocal (Andor Technology) on a Nikon Ti-E base and equipped with an iXon Ultra 888 EMCCD camera, a Zyla 4.2 Mpixel sCMOS camera, and a Tokai Hit stage-top incubator set at 37 °C. A solid-state 405 smart diode 100 mW laser, solid state 560 OPSL smart laser 50 mW laser, and solid state 637 OPSL smart laser 140 mW laser were used. Objectives used were the CFI Plan Apochromat Lambda 100×/1.45 NA oil (Nikon, MRD01905) for all drug treatment live-cell assays; and CFI Plan Apochromat 60×/1.4 NA oil (Nikon, MRD01605) to observe transient depolarization events or Parkin recruitment during live-

cell imaging. Images were acquired using Fusion software (Andor Technology, version 2.0.0.15). Parkin recruitment was imaged at the medial section of the cell.

For Parkin recruitment assay with drug treatments, cells were pre-treated with 1 ml of live-cell medium containing 100 µM CK666 (Sigma-Aldrich, SML006) (from a 20 mM stock in DMSO) for 30 min before imaging. During imaging, cells were treated with 1 ml live-cell medium containing 40 µM CCCP at the start of the third frame (4 min, time interval set at 2 min). Imaging was continued at least 1.5 h with cells in medium containing a final concentration of 20 µM CCCP and 50 µM CK666. Control cells were pretreated with an equal volume of DMSO (replacing CK666) and stimulated with 20 µM CCCP during imaging. To visualize more cells in the field, the 60×1.4 NA objective was used. Parkin recruitment assay in KD cells was similar, except without the pre-treatment step, acquisition time interval was either 1.5 or 2 min. For cyto-CFP-FRBX5 recruitment studies, cells were transfected with respective plasmids, pre-treated with 1 ml of live cell media, and imaged for 5 frames at 12 s/frame, following which either DMSO (Invitrogen, D12345) or CCCP (final 40 µM) was added and imaged for another 100 s following which 1 ml of rapamycin (Sigma 553210) (final 10 µM)-containing live cell media was added and imaged for another 150 s.

## Transmission electron microscopy

TEM sample preparation, imaging and image analysis was performed as previously described (Katona et al, 2022). Briefly, HEK cells were fixed using 2% glutaraldehyde, then with 1% $OsO_4$ and stained with 0.5% uranyl acetate, pelleted in 2% agarose (Sigma-Aldrich, Type IX ultralow gelling temperature), dehydrated in a dilution series of acetone/water, and embedded in Spurr's resin (Electron Microscopy Sciences). The sections were examined with a JEOL JEM1010 TEM fitted with a side-mounted AMT XR- 50 5Mpx CCD camera, or an FEI Tecnai 12 TEM fitted with a bottom-mounted AMT XR-111 10.5 Mpx CCD camera. Whole-cell images were taken with ×6700–14,000 (JEOL) or ×4400–6500 (FEI) direct magnification. Mitochondria were imaged using ×30,000 and ×45,000 (JEOL, 1.5 nm/px resolution). Interfaces between ER and mitochondria were analyzed using a custom ImageJ script: https://sites.imagej.net/MitoCare/. Images were randomized by one author (RC) and quantified in a blinded manner by another author (AG).

## Image analysis and quantification

### Quantification from live-cell imaging

Unless otherwise stated, all image analysis was performed on ImageJ Fiji (version 1.51n, National Institutes of Health). Cells that shrunk during imaging or exhibited signs of phototoxicity such as blebbing or vacuolization were excluded from analysis (maximal amount 10% for any treatment).

Parkin recruitment: Parkin association with mitochondria was analyzed in Fiji using the Colocalization and Time series analyzer V3 ImageJ plugin. Firstly, GFP-Parkin (rolling ball 30.0) and mito-BFP images (rolling ball 20.0) were background-subtracted and converted into 8-bit files. Parkin-associated mitochondria were thresholded using the Colocalization ImageJ plugin with the following parameters: ratio, 25% (0–100%); threshold channel 1,

25 (0–255); threshold channel 2, 25 (0–255); and display value, 255 (0–255) for U2OS; and ratio, 50% (0–100%); threshold channel 1, 50 (0–255); threshold channel 2, 50 (0–255); and display value, 255 (0–255) for HeLa. ROIs were drawn for individual cells in the overlapped pixel stack and analyzed using Time series analyzer V3. The ROI was selected as the bulk region of the cell containing mitochondria using the mito-BFP signal. Mean colocalized signal of Parkin on mitochondria was plotted with respect to imaging duration (1.5 h) at 2 min or 1.5 min intervals. To calculate the time of initial Parkin onset, the clear deviation from the pre-treatment intensity was manually read from the individual curve of each cell. For trendline analysis, data points from the linear portion of the averaged kinetic curve were extracted and a trendline was calculated using Excel. The time for the trendline to cross the point where $y = 0$ was determined as the average time for Parkin onset. Images were randomized by one author (RC) and quantified in a blinded manner by another author (TSF).

ER-Mitochondria, lysosome-mitochondria and FRB-mitochondria overlap: U2OS cells transiently transfected with the respective markers were imaged live by spinning disc confocal fluorescence microscopy at 15-s intervals in a single focal plane at the medial section 2–4 µm from the base. ROIs in the peri-nuclear region (ER-mitochondria, lysosome-mitochondria) or for the whole mitochondrial network (FRB-mitochondria) were background-subtracted using "rolling ball radius" plugin of ImageJ with a value of 20.0. The respective channels were further processed for bleach correction in ImageJ using "simple ratio" and converted to 8-bit images. The channels were then analyzed using the colocalization plugin in ImageJ with the following parameters: ratio 50% (0–100%); threshold channel 1: 30 (0–255); threshold channel 2: 1: 30 (0–255); display value: 255 (0–255) to obtain the overlapping pixels. The overlap intensity was then normalized to the pretreatment frames (1–5) and plotted over time.

### Quantification from fixed-cell imaging

PLA assays: All images were processed using ImageJ (NIH). z-projected images were background-subtracted using "rolling ball radius" plugin of ImageJ with a value of 20.0 and phalloidin (actin) channel was used to get the cellular area. Channel containing the PLA dots were converted to 8-bit and binarized. "Analyze particles" plugin was used to record total number of dots and presented as number of dots per 100 µm$^2$ of cellular area. Images were randomized by one author (AG) and quantified in a blinded manner by another author (RC). All images collected were quantified.

Parkin and LC3-positive mitochondrial area: z-projected images were background-subtracted using "rolling ball radius" plugin of Image J with a value of 20.0. The channels were then converted to 8-bit, Individual channels were thresholded using the Colocalization ImageJ plugin with the following parameters: ratio, 25% (0–100%); threshold channel 1, 25 (0–255); threshold channel 2, 25 (0–255); and display value, 255 (0–255) and overlapping pixel generated using the same plugin. Overlapping pixels were plotted as a function of total mitochondrial pixels to evaluate % mitochondrial area positive for either Parkin or LC3. Images were randomized by one author (AG) and quantified in a blinded manner by another author (RC). All images collected were quantified.

## RT-qPCR

Total RNA was purified with NucleoSpin RNA Clean-up (Macherey-Nagel) following the manufacturer's instructions. Genomic DNA was eliminated by on-column digestion with DNase I. 500 ng of RNA were reverse transcribed using LunaScript® RT Supermix (NEB) and qPCR (45 cycles) was performed on a QuantStudio 5 (Thermo Fisher Scientific). Reactions were run in triplicates with Luna® Universal qPCR Master Mix (NEB) in a total volume of 10 µl with standard cycling conditions. Relative gene expression was calculated using the $\Delta\Delta C_T$ method and normalized using ACTB as a housekeeping gene. All calculations were performed using QuantStudio Design and Analysis Software (Thermo Fisher Scientific). The list of primers used is as follows:

FMNL1 (F): 5′-CACCTGACCATCAAGCTGACC-3′
FMNL1 (R): 5′-CGTAGGCACATAATACAGACGTG-3′
FMNL2 (F): 5′-CAGGGAGCATGGATTCGCAG-3′
FMNL2 (R): 5′-TCAGGAGGTAGGTTCATAGCATT-3′
FMNL3 (F): 5′-TGGGGCTAGAGGAGTTCCTG-3′
FMNL3 (R): 5′-CCCCGACATCAAACACGTTG-3′
NCKAP1 (F): 5′-TTGTACCCCATAGCAAGTCTCT-3′
NCKAP1 (R): 5′-GGGCATTTCTCCACTGGTCAG-3′
Arp2 (F): 5′-CACCTGTGGGACTACACATTTG-3′
Arp2 (R): 5′-TGGTTGGGTTCATAGGAGGTTC-3′
ACTB (F): 5′-ACCTTCTACAATGAGCTGCG-3′
ACTB (R): 5′-CCTGGATAGCAACGTACATGG-3′.

## Mitochondria DNA depletion and RT-PCR

mtDNA depletion was done as previously described (Chakrabarti et al, 2022). Briefly, $1 \times 10^5$ MEF cells were plated in a T-75 flask and incubated in DMEM + 10% FBS overnight. 24 h later, overnight media was replaced with either EtBr-containing media (DMEM + 10% FBS + 0.2 µg/ml EtBr + 50 µg/ml uridine) or control media (DMEM + 10% FBS + 50 µg/ml uridine). Fresh media containing the chemicals were added every 48 h. On the 5th day of treatment, cells were harvested to assess relative mtDNA copy number. Briefly, cells were trypsinized, resuspended in PBS buffer containing proteinase K (0.2 mg/ml), SDS (0.2%), EDTA (5 mM) and incubated for 6 h at 50 °C with constant shaking at 1200 rpm. After isopropanol precipitation, DNA was resuspended in TE buffer and quantified by with a UV-Vis spectrophotometer (Implen). For the determination of mitochondrial DNA copy number, the genes encoding mitochondrial 12S and nuclear actin were amplified from 25 ng of DNA by qPCR on a QuantStudio 5 (Thermo Fisher Scientific) using standard cycling conditions (35 cycles). Reactions were run in triplicates using Luna® Universal qPCR Master Mix (NEB) in a total volume of 10 µl. Mitochondrial DNA copy number is defined as the relative amount of mitochondrial 12 s normalized by a nuclear DNA normalization (ACTB), calculated with the $\Delta\Delta C_T$ method and QuantStudio Design and Analysis Software (Thermo Fisher Scientific). The list of primers used are as follows:

mt-12S (F): 5′-TAGCCCTAAACCTCAACAGT-3′
mt-12S (R): 5′-TGCGCTTACTTTGTAGCCTTCAT-3′
ACTB (F): 5′-TCACCCACACTGTGCCCATCTACGA-3′
ACTB (R): 5′-CAGCGGAACCGCTCATTGCCAATGG-3′.

## Seahorse measurements

OCR and ECAR measurements were performed using the Seahorse XFe24 analyzer. In brief, 50,000 cells were seeded into each well excluding blank wells of 24-well cell culture miniplates suitable for the XFe24 analyzer. Concentrations of Oligomycin, FCCP, and Antimycin/Rotenone used at 1.5, 2, and 2.5/1 µM, respectively. Data for OCR, basal respiration, and ATP production in samples were calculated from Wave software (Agilent Technologies) and normalized to the corresponding total protein concentration.

## Lysates and western blot

Cells from a 35 mm dish were trypsinized, pelleted by centrifugation at 300 g for 5 min and resuspended in 400 µl of 1× DB (50 mM Tris-HCl, pH 6.8, 2 mM EDTA, 20% glycerol, 0.8% SDS, 0.02% Bromophenol Blue, 1000 mM NaCl, 4 M urea). Proteins were separated by SDS-PAGE in a Bio-Rad mini-gel system (7 × 8.4 cm) and transferred onto polyvinylidene fluoride membrane (EMD Millipore, IPFL00010). The membrane was blocked with TBS-T (20 mM Tris-HCl, pH 7.6, 136 mM NaCl, 0.1% Tween32 20) containing 3% BSA (VWR Life Science, VWRV0332) for 1 h, then incubated with primary antibody solution at 4 °C overnight. After washing with TBS-T, the membrane was incubated with HRP-conjugated secondary antibody for 1 h at 23 °C. Signals were detected by chemiluminescence.

Primary antibody:
anti-myc-tag antibody (Abcam, ab32 used 1:1000)
anti-INF2 antibody (CST #41081; used at 1:1000)
anti-VAP-B antibody (Proteintech #14477-1-AP; used at 1:5000 at RT)
anti-LC3B antibody (CST #12741S; used at 1:1000)
anti-Parkin antibody (CST #2132S; used at 1:1000)
anti-Parkin(phospho S65) antibody (Abcam #315376; used at 1:1000)
anti-Pink1 antibody (CST #6946S; used at 1:1000)
anti-β-actin antibody (CST #3700S; used at 1:10,000)
anti-Mfn2 antibody (CST #9482S; used at 1:1000)
anti-VDAC antibody (Invitrogen #MA5-41088; used at 1:1000)
anti-Phospho ubiquitin (Ser65) (CST #62802S; used at 1:1000).

## Lentiviral generation and viral transduction

Lentivirus was generated using HEK293T cells with Lipofectamine 3000 kit (L3000008, Life Technologies). Cells were seeded in Opti-MEM media, containing 5% FBS and no antibiotics, overnight for ~80% confluency. Cells were transfected with packaging vectors pVSV-G and pSPAX2, along with expression construct (GFP-RFP-Fis1(tm)) at a 1:4:3 ratio, scaled accordingly. Opti-MEM media was exchanged with fresh media 6 h after transfection. The supernatant containing virus was collected at 24- and 48 h post-transfection and pooled together. The virus was cleared by centrifugation for 15 min at $1000 \times g$ and aliquoted to avoid freeze–thaw cycles.

MEF-WT cells were incubated in DMEM containing 8 µg/mL polybrene (1:1000 dilution) with the virus. Transduction were left overnight, and virus-containing media was exchanged in the morning with fresh media lacking polybrene. Transduced cells were allowed to recover in fresh media for 24 h prior to antibiotic selection and further analysis.

## Statistical analysis and graph plotting software

All statistical analyses and $P$ value determinations were conducted using GraphPad Prism QuickCalcs or GraphPad Prism 10 (version 10.1.0, GraphPad Software). To determine $P$ values, an unpaired Student's $t$ test was performed between two groups of data, comparing full datasets as stated in the figure legends. To compare across more than two conditions, multiple comparisons (unpaired) and one-way ANOVA were performed in GraphPad Prism 10. Live-cell actin burst and parkin curves, along with the standard errors of the mean (SEM) were plotted using Microsoft Excel for Office 365 (version 16.0.11231.20164, Microsoft Corporation).

# Data availability

The source data files have been uploaded at Mendeley data and can be accessed using the following link: EMBO Reports_ADA and mitophagy - Mendeley Data can be cited as *Chakrabarti, Rajarshi (2025), "EMBO Reports_ADA and mitophagy", Mendeley Data, V1, doi: 10.17632/hvyrpf7cjy.2.*

The source data of this paper are collected in the following database record: biostudies:S-SCDT-10_1038-S44319-025-00561-y.

# Peer review information

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

## Acknowledgements

We thank Dr. David Chan for kindly providing the MEF cell lines, David Weaver for his expertise in imaging, Dr. Gyorgy Hajnoczky for his critical advice and reagents, Dr. Christopher Shoemaker for the mitophagy tandem reporter construct and Pak Rin for his ubiquitous presence behind the scenes. We also thank Dr. Marilen Federico for assistance with seahorse assays and Dr. Elham Javed for assistance with confocal microscopy. We also thank the Electron Microscopy Core Facility of Thomas Jefferson University for their assistance with TEM. This work was supported by NIH R35 GM150811-1 (RC), Margaret Q. Landenberger Research Foundation grant (RC), Annesley Eye Brain Centre (RC), NIH R35 GM122545-07 (HNH) and NIH R35147191-01 (MT).

## Author contributions

**Tak Shun Fung**: Data curation; Formal analysis; Investigation. **Amrapali Ghosh**: Data curation; Formal analysis; Validation; Investigation; Writing—original draft; Writing—review and editing. **Maite R Zavala**: Data curation; Investigation. **Zuzana Nichtova**: Data curation; Formal analysis. **Dhavalkumar Shukal**: Formal analysis. **Marco Tigano**: Formal analysis; Funding acquisition. **Gyorgy Csordas**: Methodology. **Henry N Higgs**: Conceptualization; Resources; Validation; Methodology; Writing—original draft; Writing—review and editing. **Rajarshi Chakrabarti**: Conceptualization; Resources; Data curation; Formal analysis; Supervision; Funding acquisition; Validation; Investigation; Methodology; Writing—original draft; Project administration; Writing—review and editing.

Source data underlying figure panels in this paper may have individual authorship assigned. Where available, figure panel/source data authorship is listed in the following database record: biostudies:S-SCDT-10_1038-S44319-025-00561-y.

## Disclosure and competing interests statement

The authors declare no competing interests.

# Expanded View Figures

**Figure EV1.  ADA delays Parkin recruitment on depolarized mitochondria (CCCP treatment).**

(A) RT-qPCR to validate knockdown of the respective genes in U2OS (upper; green bars) and HeLa cells (lower; red bars). Normalized expression data combined from 3 biological replicates. Error: Mean ±s.d. (B) Time lapse montages of WT HeLa cells transfected with GFP-Parkin (green) and Mito-BFP (red) pre-treated with 30 min 100 µM CK666 or DMSO before treatment with 20 µM CCCP treatment during live-cell imaging at time 0. Imaging conducted at the medial cell section. Scale: 10 µm. Time in minutes. (C) Quantification of colocalized Parkin signal with mitochondrial signal in live-cell imaging after 20 µM treatment of CCCP at time 0 for 30 min DMSO (black) or 100 µM CK666 (red) pre-treated HeLa cells. $N = 26$ cells for DMSO/CCCP and 31 for CK666/CCCP. 4 independent experiments. Error ± SEM. (D) Scatter plot of Parkin signal onset on mitochondria for DMSO or 100 µM CK666 treated HeLa cells. Parkin onset time = 52.64 min ± 11.35 (mean ± s.d.) for DMSO/CCCP and 25.23 ± 10.89 for CK666/CCCP. Same dataset as in Figure EV1C ($N = 26$ cells for DMSO/CCCP and 31 for CK666/CCCP. 4 independent experiments). Error ± s.d. $P = 0.0001$ (****). Student's unpaired $t$ test used. (E) Time lapse montages of ctrl, Arp2 KD, and FMNL1/2 DKD HeLa cells transfected with GFP-Parkin (green) and Mito-BFP (red) treated with 20 µM CCCP treatment at time 0 during live-cell imaging. Imaging conducted at the medial cell section. Scale: 10 µm. Time in minutes. (F) Quantification of colocalized Parkin signal with mitochondrial signal in live-cell imaging after 20 µM treatment of CCCP at time 0 for ctrl (black), FMNL1/2 DKD (red) or Arp2 KD (gold) HeLa cells. Data of 38 cells for ctrl; 19 for FMNL1/2 DKD and 24 for Arp2 KD from 3 independent experiments. Error ± SEM. (G) Scatter plot of Parkin signal onset on mitochondria for ctrl, FMNL 1/2 DKD or Arp2 KD in HeLa cells. Parkin onset time = 46.32 min ± 12.16 (mean ± s.d.) for ctrl; 28.95 ± 6.12 for FMNL 1/2 DKD and 28.67 ± 6.01 for Arp2 KD cells. Same dataset as in Figure EV1F (Data of 38 cells for ctrl; 19 for FMNL1/2 DKD and 24 for Arp2 KD from 3 independent experiments). $P = 0.0001$ (****) for ctrl vs FMNL 1/2 DKD and ctrl vs Arp2 KD; $P = 0.9949$ (n.s.) for FMNL 1/2 DKD vs Arp2 KD. Tukey's multiple comparisons test used. (H) Representative fixed cell images of MEFs transfected with Parkin-GFP and mitoBFP treated with 20 µM CCCP in the presence or absence of CK666 (100 µM) for 60 min. Arrows show Parkin-GFP colocalized with the mitoBFP signal. Scale: 10 µm. (I) Quantification showing percentage of total mitochondrial area/cell positive for Parkin-GFP in MEFs treated with 20 µM CCCP in the presence or absence of CK666 (100 µM) for 60 min. MEFs treated with CCCP: 41 cells and MEFs treated with CCCP + CK666: 30 cells from 3 independent experiments. $P = 0.0001$ (****). Students $t$ test used. Error: Mean ±s.d.

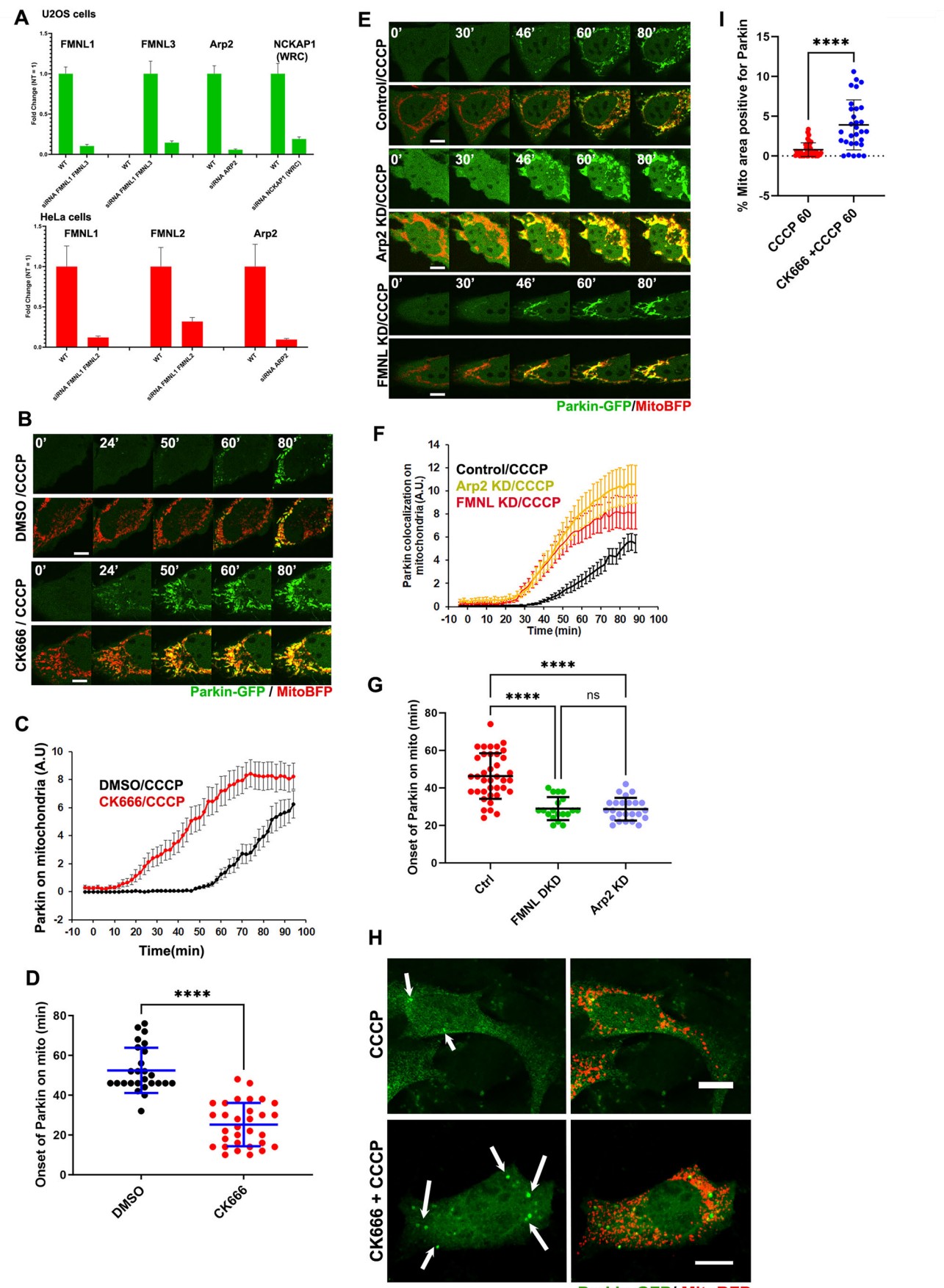

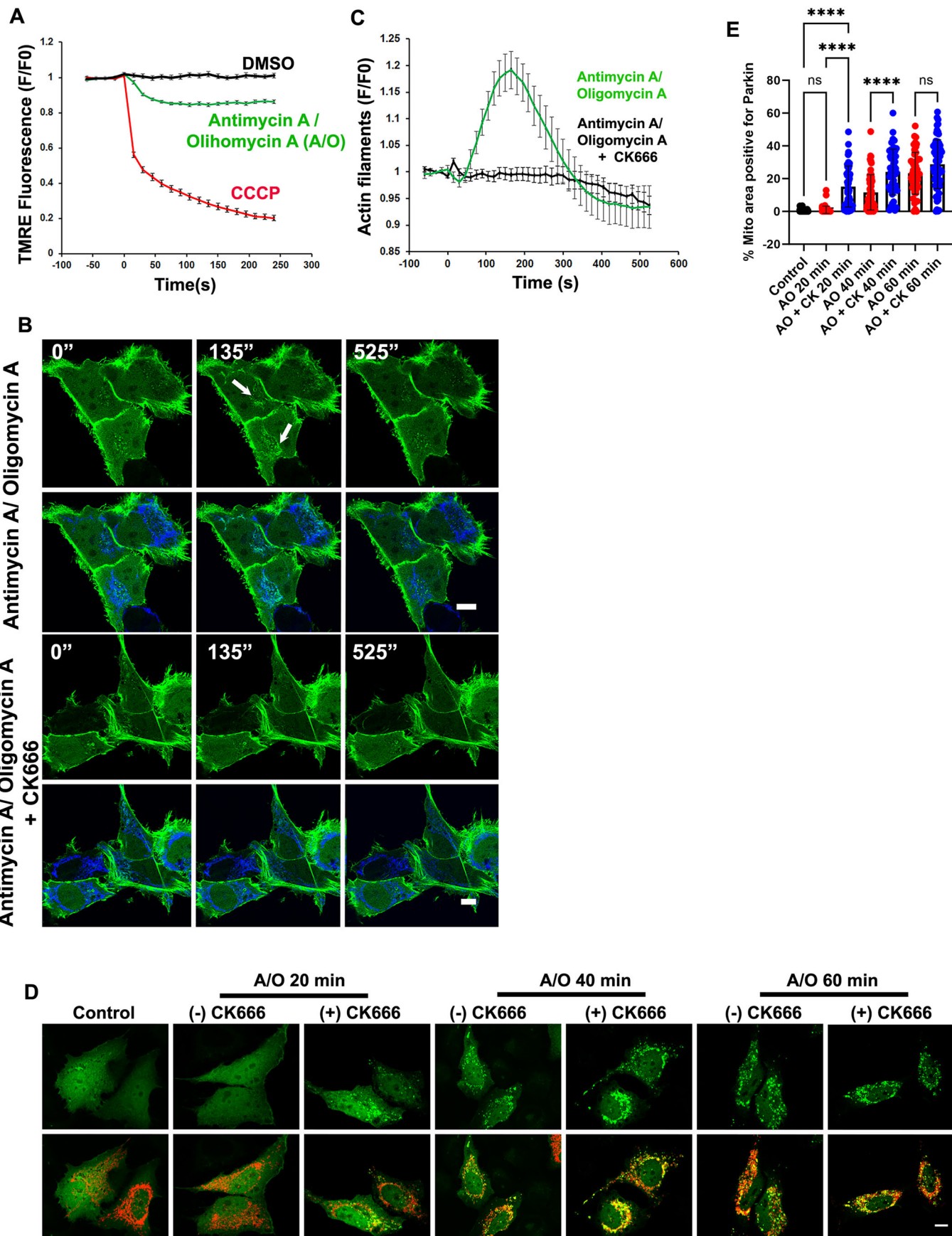

◀ **Figure EV2. Antimycin A/Oligomycin A induced ADA delays Parkin recruitment in HeLa cells.**

(A) Graph showing kinetics of TMRE fluorescence in HeLa cells treated with either DMSO (black) or 25 µM Antimycin A/ 15 µM Oligomycin A (A/O) or 20 µM CCCP. Treatment added at time 0. Data from 3 independent experiments totaling 25 cells (DMSO), 39 cells (CCCP) and 42 cells (A/O). Error: ±SEM. (B) Time-lapse montage of HeLa cells transfected with GFP-Ftractin and mitoBFP treated with 25 µM Antimycin A/ 15 µM Oligomycin A (A/O) in the presence or absence of CK666 (100 µM) at time 0 min. Arrows show the induction of mitochondrial-associated actin. Scale bar: 10 µm. (C) Graph showing actin filaments fluorescence quantified from the data set in Fig EV2B. Treatment added at time 0. Data combine 3 independent experiments having 21 cells (A/O) and 29 cells (A/O + CK666). Error: Mean ±SEM. (D) Representative fixed cell images showing HeLa cells transfected with Parkin-GFP and mitoBFP and treated with 25 µM Antimycin A/ 15 µM Oligomycin A (A/O) in the presence or absence of CK666 (100 µM) for various time points as indicated. (E) Scatter plot showing percentage of total mitochondrial area/cell positive for Parkin-GFP in HeLa cells treated with 25 µM Antimycin A/ 15 µM Oligomycin A (A/O) in the presence or absence of CK666 (100 µM) for various time points. Data from 3 independent experiments having Control: 62 cells; A/O 20 min: 53 cells; A/O + CK666 20 min: 57 cells; A/O 40 min: 64 cells; A/O + CK666 40 min: 50 cells; A/O 60 min: 47 cells; A/O + CK666 60 min: 69 cells. $P = 0.0001$ (****). Students *t* test used. Error ± s.d.

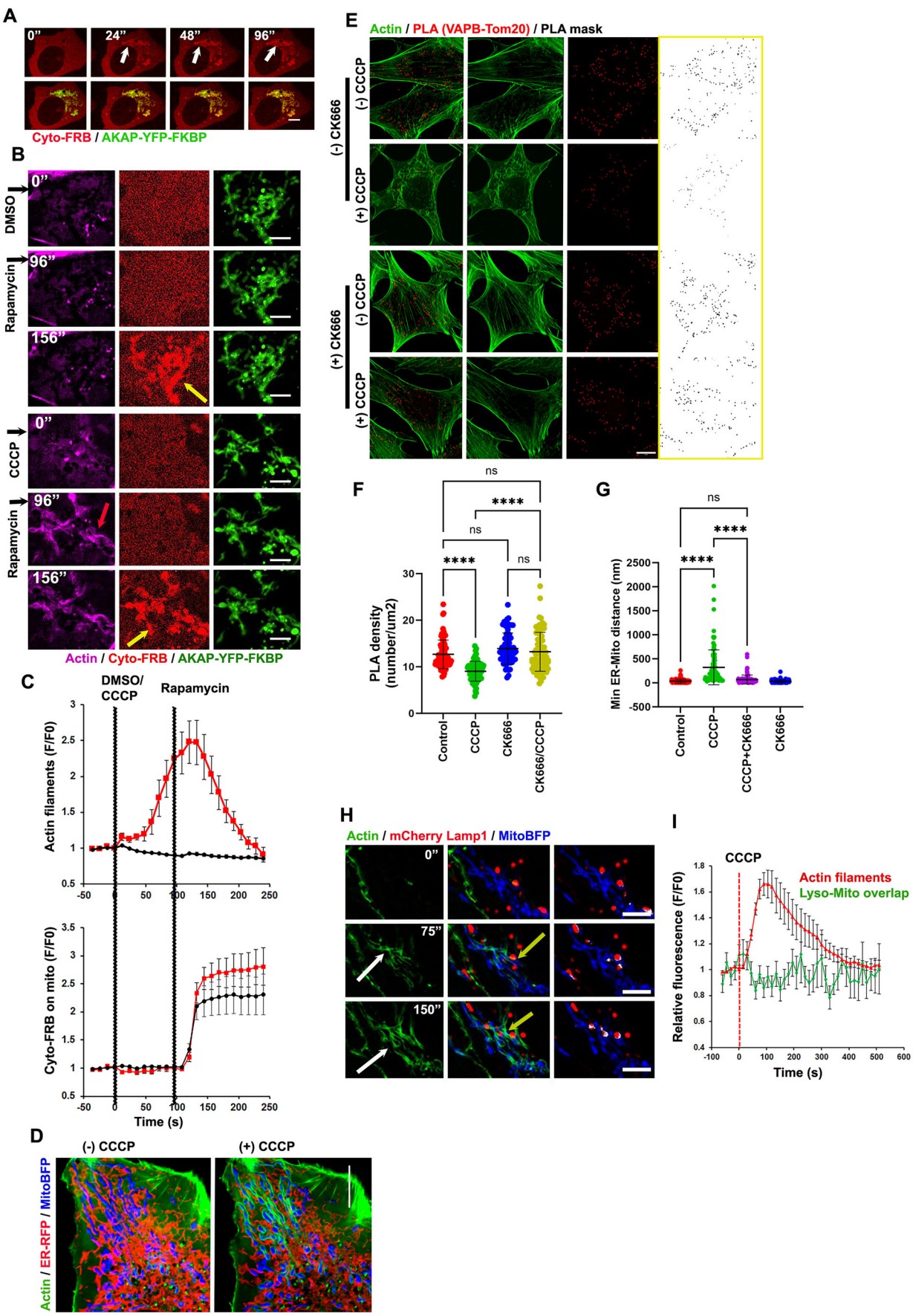

**Figure EV3.  ADA disrupts ER-mitochondrial contacts but not mitochondria-lysosome contacts.**

(A) Time lapse montage of U2OS cells transfected with cyto-(FRB)5 and AKAP1-YFP-FKBP and treated with 10 μM Rapamycin at time 0 as indicated. Scale bar: 5 μm. (B) Time lapse montage of U2OS cells transfected with cyto-(FRB)5, AKAP1-YFP-FKBP CCCP and GFP-Ftractin and treated with either DMSO or CCCP for 100 s followed by rapamycin (final 10 μM). White arrows indicate Cyto-FRB positive mitochondria. (C) Quantification of actin filaments (upper) and mitochondrially associated CFP-(FRB)5 (lower) after the following treatments: DMSO or CCCP at time 0 followed by rapamycin (final 10 μM) treatment after 100 s of initial treatment. Data from 9 cells for each condition from 3 independent runs. Error: ±SEM. (D) Micrographs from U2OS cells transfected with ER-RFP (ER; red), mito-BFP (mitochondria; blue) and GFP-F tractin (actin filaments; green) before and after 100 s of CCCP treatment. Scale bar: 5 μm. (E) Representative fixed-cell images from Proximity Ligation Assay (PLA) conducted between VAPB (ER) and Tom20 (mitochondria) to assess ERMC (red and binarized) in HeLa cells treated with 20 μM CCCP for 5 min in the presence or absence of CK666 (100 μM). Cells were stained with Phalloidin-488 to label actin filaments prior to PLA staining. Scale: 10 μm. (F) Scatter plot representing density of PLA dots (no. per 100 μm$^2$) in control and CCCP-treated HeLa cells in the presence or absence of CK666. Same data set as Fig EV3E. Data with 80 cells (control); 82 cells (CCCP); 65 cells (CK666) and 77 cells (CK666 + CCCP) from 4 independent coverslips. $P = 0.0001$ (****). Two-way ANOVA used. Error: Mean ±s.d. (G) Scatter plot showing the minimum distance between ER and mitochondria as estimated from TEM images represented in data set in Fig. 2I–J (Control $n = 88$ mitochondria; 20 μM CCCP/ 5 min: $n = 94$ mitochondria; 20 μM CCCP + CK666 (100 μM/ 5 min); $n = 90$ mitochondria and 100 μM CK666 treated; $n = 106$ mitochondria) in MEF cells. Data from three independent fixation for each condition. Error: ±s.d.; $P = 0.0001$ (****). Two-way ANOVA used. (H) Time-lapse montages of CCCP-induced actin polymerization and lysosome-mitochondria overlap in U2OS cells transfected with mCherry-Lamp1 (lysosomes; red), mito-BFP (mitochondria; green) and GFP-F tractin (actin filaments; magenta). White arrow indicates mitochondria-lysosome contact. Scale bar: 5 μm. (I) Quantification of CCCP-induced actin polymerization and lysosome-mitochondria overlap in U2OS cells transfected with mCherry-Lamp1 (lysosome; red), mito-BFP (mitochondria; green) and GFP-F tractin (actin filaments; magenta) 20 cells for each condition obtained from 2 independent experiments. Error: ±SEM.

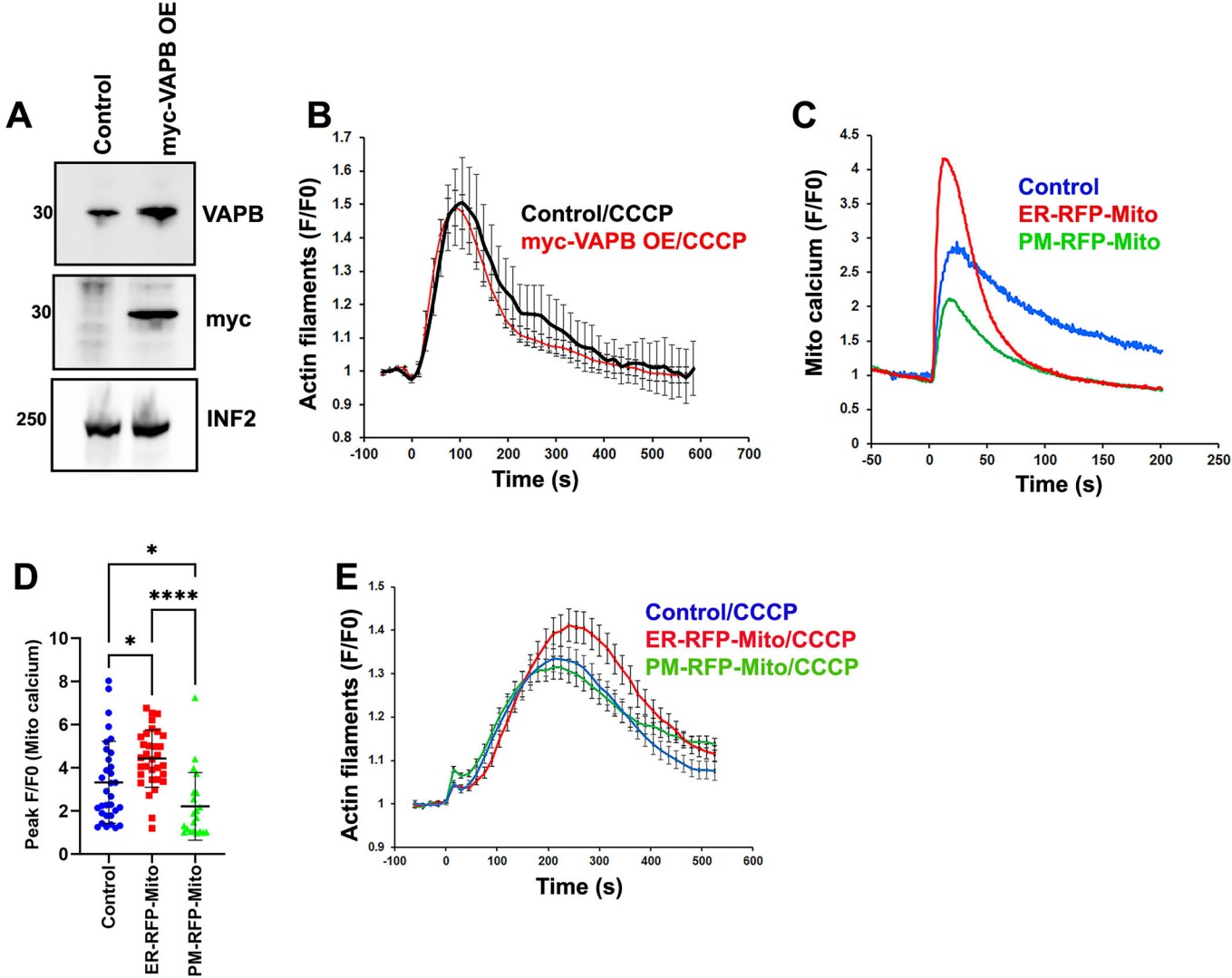

**Figure EV4. ERMC regulates CCCP-induced Parkin recruitment.**

(A) Western blot for VAP-B and myc-tag showing VAP-B expression in control and myc-VAP-B overexpressed U2OS cells. INF2 is used as loading control. Molecular weight in kDa. (B) Quantification of CCCP-induced actin polymerization in control and myc-VAP-B-overexpressing U2OS cells. Data with 65 cells (control) and 53 cells (overexpressing myc-VAP-B) from 3 independent runs. Error ± SEM. (C) Averaged traces showing mitochondrial calcium fold change (MitoGCamp6f) following histamine stimulation (at time 0) in control (ER-RFP expressing), ER-RFP-mito (ER-Mito linker) and PM-RFP-Mito (PM-Mito-linker) transfected HeLa cells. Data from 2 independent experiments comprising of 4 individual traces for each condition. Error: ±SEM. (D) Dot plot of peak fold change in mitochondrial calcium following histamine stimulation from individual traces as in Fig S4C. Data for 32 cells (Control- ER-RFP); 32 cells (ER-RFP-Mito); 28 cells (PM-RFP-Mito) for 3 independent runs. Error: ±s.d. $P = 0.0213$ (*) for Control Vs ER-RFP-Mito, $P = 0.0439$ (*) for Control Vs PM-RFP-Mito and $P = 0.0001$ (****) for ER-RFP-Mito Vs PM-RFP-Mito, One-way ANOVA used. (E) Graph showing actin filaments intensity (GFP-Ftractin) from HeLa cells transiently expressing either ER-RFP-Mito (magenta) or PM-RFP-Mito or ER-RFP, treated with 20 μM CCCP. Data from 28 cells (ER-RFP); 27 cells (ER-RFP-Mito) and 23 cells (PM-RFP-Mito) from 4 independent experiments. Error: ±SEM.

**A**

**B**

**C** (-) CK666    (+) CK666

(-) EtBr

(+) EtBr

Actin / Tom20

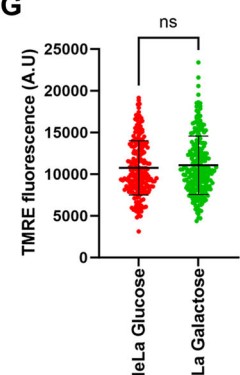

**D**

**E**

**F**

Basal    ATP-linked    Maximal

Basal

**G**

◀ **Figure EV5.  EtBr-induced mtDNA depletion & formation of ADA-like filaments and OxPhos dependency causes prolonged ADA.**

(A) RT-qPCR to validate depletion of mitochondrial DNA (mtDNA) in MEFs treated with and without EtBr (0.2 µg/ml) for 5 days. Normalized expression data derived from 3 biological replicates. $P = 0.0001$ (****). Student's $T$ test used. Error: Mean ±s.d. (B) Representative traces of OCR and ECAR readout from MEFs treated with and without EtBr (0.2 µg/ml) for 5 days. Error: ±s.d. $N=$ at least 3 independent wells. (C) MEFs treated with or without EtBr (0.2 µg/ml) for 5 days, treated with CK666 for 4 h (100 µM), fixed, and stained for actin filaments (green) and mitochondria (red). Scale bars: 10 µm (main panel); 5 µm (inset). (D) Scatter plot showing the percentage of mitochondrial perimeter within 0–40 nm distance distributed within bins of 10 nm; Same data set as in Fig. 7D. Control (Red; $n = 62$ mitochondria); Control with CK666 (green; $n = 63$ mitochondria); EtBr (purple; $n = 63$ mitochondria) EtBr with CK666 (blue; $n = 65$ mitochondria). Data from three independent fixation for each condition; Error: Mean ±s.d. (E) Scatter plot showing the minimum distance between ER and mitochondria as estimated from TEM images represented in dataset from Fig. 7C, D (Controls without CK666; $n = 62$ mitochondria; with CK666; $n = 63$ mitochondria; +EtBr without CK666: $n = 63$ mitochondria and +EtBr with CK666: $n = 65$ mitochondria). Three independent experiments for each condition $P = 0.0001$ (****). One-way ANOVA used. Error: Mean ±s.d. (F) OCR and ECAR traces of HeLa cells cultured in either 10 mM glucose or galactose for 10 days, sequentially treated with oligomycin (1.5 µM), FCCP (1 µM) and Antimycin/Rotenone (2.5 µM/ 1 µM) at designated times. Bar graph showing basal, ATP-linked, and maximal respiration calculated from traces shown. Error ± s.d. $P = 0.0041$ (**), 0.0112 (*) and 0.0056 (***), student's $t$ test used. Bar graph showing ECAR rates (glycolysis) in unstimulated HeLa cells cultured in glucose or galactose for 10 days from traces shown. $P = 0.0003$ (***), student's $t$ test used. Error: Mean ±s.d. (G) Dot plot showing TMRE fluorescence (measuring mitochondrial membrane potential) in HeLa cells cultured in either 10 mM glucose or galactose for 10 days. Data from 213 cells (HeLa glucose) and 228 cells (HeLa galactose) from 4 independent experiments. Student's $t$ test used. Error: Mean ±s.d.

