## [Peer Review File · EMBO Reports]

Peri-mitochondrial actin filaments inhibit Parkin assembly by disrupting ER-mitochondria contacts

Tak Shun Fung, Amrapali Ghosh, Maite Zavala, Zuzana Nichtova, Dhavalkumar Shukal, Marco Tigano, Gyorgy Csordas, Henry Higgs, and Rajarshi Chakrabarti

Corresponding author: Rajarshi Chakrabarti (Rajarshi.Chakrabarti@jefferson.edu)

Review Timeline:

Submission Date:	7th May 24
Editorial Decision:	23rd Jul 24
Revision Received:	7th May 25
Editorial Decision:	5th Jun 25
Revision Received:	12th Jun 25
Accepted:	6th Aug 25

Editor: Deniz Senyilmaz Tiebe

Transaction Report:

Dear Raj,

Thank you for submitting your research manuscript to our journal, which was now seen by three referees, whose reports are copied below.

Please accept my apologies for this unusual delay in getting back to you. As mentioned before, it took longer than anticipated to receive the full set of referee reports.

We concur with the referees that the proposed role of ADA in Parkin assembly through ER-mitochondria contacts is in principle very interesting. However, the referees also raise significant concerns that need to be addressed to consider publication here. In particular,

- CCCP was predominantly used in the study for Parkin induction. Alternative conditions for Parkin induction need to be applied (referee #1, standfirst, major point 2; referee #2, point 5; referee #3)
- The proposed effect of ADA on PINK1 accumulation should be tested experimentally (referee #1, major point 3; referee #2, point 1; referee #3, paragraph 4)
- Mitochondria-ER contacts and their role in Parkin recruitment need to be better characterized (referee #1, major points 5 and 7; referee #2, points 2 and 3).
- Caveats mentioned by referee #3 need to be acknowledged and discussed.

We are aware that elucidating the entire mechanism may be unfeasible, but we also think that assessing the involvement of Mfn2, VDAC1 and SYNJ2BP in the link between ADA and ER-mitochondria contacts as per referee #1 will significantly strengthen the manuscript.

I would be happy to discuss anything regarding the revision further, also by video chat, should you wish.

Given these recommendations, we would like to invite you to submit a revised manuscript. Please revise your manuscript with the understanding that the referee concerns (as in their reports) must be fully addressed and their suggestions taken on board. Please address all referee concerns in a complete point-by-point response. Acceptance of the manuscript will depend on a positive outcome of a second round of review. It is EMBO reports policy to allow a single round of major experimental revision only and acceptance or rejection of the manuscript will therefore depend on the completeness of your responses included in the next, final version of the manuscript.

We realize that it is difficult to revise to a specific deadline. In the interest of protecting the conceptual advance provided by the work, we recommend a revision within 3 months. Please discuss the revision progress ahead of this time with me if you require more time to complete the revisions, or if you have questions or comments regarding the revision (also by video chat).

1. A data availability section providing access to data deposited in public databases is missing (where applicable).
2. Your manuscript contains statistics and error bars based on $n=2$. Please use scatter plots in these cases.

You can submit the revision either as a Scientific Report or as a Research Article. For Scientific Reports, the revised manuscript can contain up to 5 main figures and 5 Expanded View figures, and it should not exceed 27000 characters. If the revision leads to a manuscript with more than 5 main figures it will be published as a Research Article. In this case the Results and Discussion section should be separate. If a Scientific Report is submitted, these sections have to be combined. This will help to shorten the manuscript text by eliminating some redundancy that is inevitable when discussing the same experiments twice. In either case, all materials and methods should be included in the main manuscript file.

4) a .docx formatted letter INCLUDING the reviewers' reports and your detailed point-by-point responses to their comments. As part of the EMBO publication's Transparent Editorial Process, EMBO reports publishes online a Review Process File (RPF) to accompany accepted manuscripts. This File will be published in conjunction with your paper and will include the referee reports, your point-by-point response and all pertinent correspondence relating to the manuscript.

<https://www.embopress.org/page/journal/14693178/authorguide#transparentprocess>

5) a complete author checklist, which you can download from our author guidelines

<https://www.embopress.org/page/journal/14693178/authorguide>. Please insert information in the checklist that is also reflected in the manuscript. The completed author checklist will also be part of the RPF.

6) Please note that all corresponding authors are required to supply an ORCID ID for their name upon submission of a revised manuscript (<<https://orcid.org/>>). Please find instructions on how to link your ORCID ID to your account in our manuscript tracking system in our Author guidelines

<<https://www.embopress.org/page/journal/14693178/authorguide#authorshipguidelines>>

7) Before submitting your revision, primary datasets produced in this study need to be deposited in an appropriate public database (see <https://www.embopress.org/page/journal/14693178/authorguide#datadeposition>). Please remember to provide a reviewer password if the datasets are not yet public. The accession numbers and database should be listed in a formal "Data Availability" section placed after Materials & Method (see also

<https://www.embopress.org/page/journal/14693178/authorguide#datadeposition>). Please note that the Data Availability Section is restricted to new primary data that are part of this study. * Note - All links should resolve to a page where the data can be accessed. *

Additional information on source data and instruction on how to label the files are available:

<https://www.embopress.org/page/journal/14693178/authorguide#sourcedata>

9) Our journal encourages inclusion of *data citations in the reference list* to directly cite datasets that were re-used and obtained from public databases. Data citations in the article text are distinct from normal bibliographical citations and should directly link to the database records from which the data can be accessed. In the main text, data citations are formatted as follows: "Data ref: Smith et al, 2001" or "Data ref: NCBI Sequence Read Archive PRJNA342805, 2017". In the Reference list, data citations must be labeled with "[DATASET]". A data reference must provide the database name, accession number/identifiers and a resolvable link to the landing page from which the data can be accessed at the end of the reference. Further instructions are available at <http://www.embopress.org/page/journal/14693178/authorguide#referencesformat>

10) Regarding data quantification (see Figure Legends:

<https://www.embopress.org/page/journal/14693178/authorguide#figureformat>)

- the name of the statistical test used to generate error bars and P values,

- the number (n) of independent experiments (please specify technical or biological replicates) underlying each data point,

- the nature of the bars and error bars (s.d., s.e.m.),

- If the data are obtained from n Program fragment delivered error ``Can't locate object method "less" via package "than" (perhaps you forgot to load "than"?) at //ejpvfs23/sites23b/embor_www/letters/embor_decision_revise_and_review.txt line 56.' 2, use scatter blots showing the individual data points.

12) Please also note our reference format:

13) All Materials and Methods need to be described in the main text using our 'Structured Methods' format, which is required for all research articles. According to this format, the Methods section includes a Reagents and Tools Table (listing key reagents, experimental models, software and relevant equipment and including their sources and relevant identifiers) followed by a Methods and Protocols section describing the methods using a step-by-step protocol format. The aim is to facilitate adoption of the methodologies across labs. More information on how to adhere to this format as well as a downloadable template (.docx) for the Reagents and Tools Table can be found in our author guidelines:

I look forward to seeing a revised version of your manuscript when it is ready. Please let me know if you have questions or comments regarding the revision.

Kind regards,

Deniz

Deniz Senyilmaz Tiebe, PhD
Scientific Editor
EMBO Reports

Referee #1:

In this manuscript, Fung and colleagues investigate the contribution of a transient cloud of actin filaments, termed ADA, at mitochondria during PARKIN-dependent mitophagy. This study is a significant follow-up, as this group was the first to identify and characterize ADA and its role in various mitochondrial functions. Using microscopy, the authors demonstrate that inhibition of Arp2/3-complex activity, or other actin-related proteins involved in ADA formation, enhances the recruitment of Parkin and LC3 to mitochondria upon CCCP/FCCP-mediated mitochondrial damage. They report that CCCP treatment disrupts mitochondrial-ER contact sites (MERCs), coinciding with the accumulation of peri-mitochondrial actin. Interestingly, this disruption is rescued upon inhibition of Arp2/3-complex activity or ADA, suggesting a direct role of ADA in MERC biology. Similar to Arp2/3 inhibition, the stimulation of functional MERCs restores the kinetics of Parkin and LC3 recruitment upon both acute and chronic mitochondrial damage in CCCP- and EtBr-treated cells, respectively. Finally, the authors show that an increased dependence on mitochondrial metabolism, through growth in galactose media, prolongs damage-induced mitochondrial actin and impairs mitochondrial Parkin recruitment.

Overall, this study provides an intriguing link between critical mitochondrial functions, including Parkin-dependent mitophagy and mitochondria-ER contacts, adding new complexity to these mechanisms. Identifying and characterizing the role of ADA in Parkin-dependent mitophagy is novel and interesting, and the data support the idea that ADA controls mito-ER contacts formation delaying Parkin recruitment to mitochondria, and potentially the whole mitophagy process.

However, further experiments are needed to strengthen the manuscript. In particular, a further characterization of mitochondrial-ER contact site and mitophagy analysis, and additional image quantification (ADA formation...).

The authors only use systems where they overexpress PARKIN and treat cell with FCCCP which limits the study. The use of other stimuli for Parkin-dependent activation are necessary. The authors should also investigate if these findings also apply to other types of mitophagy. Moreover, the authors should propose mechanisms by which ADA controls MERCs. Finally, particular attention should be given to the statistical analysis and the number of experiments presented.

Major points

- While elucidating the full mechanism can be challenging, presenting some evidence, even if incomplete, on how ADA controls mitochondria-ER contacts would be beneficial. For example, McLelland et al. (PMID: 29676259) proposed that Mfn2 is ubiquitinated early on during the PINK1-PARKIN dependent mitophagy to decrease MERCs and allow mitophagy. The authors should discuss this paper and check the levels of Mfn2 or Ub-Mfn2 in their different systems, as this could be useful for understanding the mechanism. Other proteins involved in this process, such as VDAC1 and SYNJ2BP, should also be checked.
- Can the authors reproduce their data under other conditions that stimulate Parkin-dependent mitophagy, aside from CCCP? Additionally, since endogenous Parkin is expressed in some cell lines, it would be interesting to analyze the recruitment of endogenous Parkin rather than overexpressed Parkin. Moreover, is this mechanism conserved across other types of mitophagy induction? These points are crucial for broadening the applicability of the study's results.
- The authors currently analyze the speed of overexpressed Parkin and LC3 recruitment to mitochondria upon CCCP treatment. It would be insightful to investigate upstream steps of Parkin recruitment at endogenous levels, such as PINK1 accumulation and phospho-ubiquitin.
- Does Parkin is recruited at mito-ER contacts upon stimulation and how it is affected by Arp2/3 and CCCP or increased MERCs?
- The analysis of mitochondria-ER interaction needs to be strengthened. While the authors used PLA, this system is highly dependent on the pair of antibodies used, and it seems evident now that tethering complexes can behave differently depending on the cellular context. For example, TOMM20 should not be used in PLA as it is one of the first targets of PARKIN during CCCP treatment. The authors should use transmission electron microscopy to validate their PLA results in Figures 2 and 5. Additionally, the authors should analyze mitochondria-ER contacts under their experimental conditions with galactose and EtBr treatment.
- In EV3X, while mitochondrial calcium analysis is appreciated, it does not provide evidence that the contacts are increased or decreased upon tethering mitochondria-ER or mitochondria-PM, respectively. A microscopy analysis of these contacts should be performed. Additionally, mitochondrial localization, morphology, and membrane potential in PM-Mito tether conditions should be assessed to demonstrate that mitochondria are indeed localized at the PM and are functional.
- The authors should decrease mitochondria-ER contacts, and investigate what happens to ADA, Parkin, and LC3 recruitment under basal and CCCP conditions. For now, all their rescue experiments are based on overexpressing systems (tether, VAPB...)
- The authors changed cell lines without explanation in the manuscript, which is confusing. While analyzing U2OS and HeLa cells is acceptable since they have been characterized for ADA and Parkin recruitment, the use of MEFs is difficult to understand. In Figure 2C-F, PLA in MEFs is confusing as previous experiments have not been conducted in these cell lines to show ADA or Parkin recruitment. If MEFs are to be used, ADA, delay in Parkin recruitment, and LC3 should be demonstrated in these cell lines. Additionally, in EV4, the characterization of EtBr is done in MEFs while the functional assay is done in HeLa cells, which is inconsistent. Similarly, in Figure 6, the functional assay is done in U2OS while the characterization of galactose was performed in HeLa cell. The characterization should be done in the corresponding cell line where the functional assay is performed.

Minor Points

- A significant number of quantifications present statistical analysis while the experiments have only been performed twice (e.g., EC1F, G, H, I; 2F, EV2F, EV3, EV4, EV5, Fig4, Fig5, and Fig6). For all these panels, three independent experiments should be conducted to ensure the robustness of the statistical analysis. Additionally, it would be beneficial if the figure legends included the total number of cells analyzed and the statistical tests used (even if these tests are described in the Materials and Methods section).
- The authors need to provide clearer explanations of how certain quantifications were performed and what they represent. For example, what does "Parkin on mitochondria (a.u.)" or "Onset of Parkin at mitochondria (min)" mean? How are these values calculated? What does a.u. stand for? What is the threshold for considering a cell to have Parkin at mitochondria? Is it based on

individual mitochondria? Should the entire mitochondrion be Parkin-positive or just a part of it? Additionally, in Figure 3E and G, the method for quantifying Parkin at mitochondria appears to change. Can the authors explain why and clarify what is represented by "% cells with Parkin on mitochondria" (e.g., all mitochondria, the majority of mitochondria)? The concept of "Parkin-positive mitochondrial area" also needs explanation. These quantifications are critical for the conclusions of the paper and need to be significantly better explained.

- During galactose treatment and in EtBr-treated cells, the authors should investigate if Parkin and LC3 recruitment can be reversed by overexpressing their artificial mitochondria-ER tether or VAPB overexpression as shown in Figure 6.
- Quantification of ADA needs to be performed in EV3E, EV4C, 6A
- EV1 H, I: I don't see Parkin recruitment in the images shown. Why the authors choose 30 min of FCCCP treatment for this condition?
- The presentation of images should be optimized. Some images are missing scale bars, zoom areas are not consistently represented, and some zoomed areas are missing.
- The reason for the change in quantification methods from Figure 3E to 3G should be explained.
- Mitochondria-lysosome interactions could be strengthened.
- Autophagy fluxes (LC3-I vs LC3-II, p62) could be analyzed by immunoblots to complement the immunofluorescence of LC3 overexpression.
- The authors should discuss previous literature related to mitochondria-ER contacts during mitophagy that is currently missing from their discussion and introduction.

Referee #2:

Comments to the authors

In a previous study, the authors identified that mitochondrial depolarization generates a transient cloud of actin filaments around depolarized mitochondria (ADA), and this phenomenon is critical for a metabolic shift from oxidative phosphorylation to glycolysis upon mitochondrial dysfunction. Here, they found another possible function of ADA formation. They found that ADA prevents the efficient Parkin translocation to dysfunctional mitochondria. Their series of data indicated that the disruption of mitochondria-ER contact sites by ADA may explain the prevention of the efficient Parkin translocation by ADA. Each experiment was carefully performed and the overall data are quite convincing. Therefore, I fully agree that ADA formation does indeed prevent the mitochondria-ER contact site formation and the efficient Parkin translocation to dysfunctional mitochondria. These are very interesting findings. However, the actual delay of Parkin recruitment is only about 15 minutes, and the actual effects on the efficiency of mitochondrial clearance are not shown. Therefore, the physiological significance of the slight delay in Parkin recruitment remained unclear. Please see specific comments below.

Major points

1. The time scale of Fig. 2A and 2B should be seconds, as mitochondria do not appear to be fragmented yet (also given the time scale of similar experiments conducted in extended Fig. 2D-F). If this is the case, the disruption of the ADA-mediated mitochondria-ER contact sites is a transient event that may recover within 10 min (Fig. 2B). How does this transient event affect the Parkin translocation at 30-60 min? The authors mentioned the possibility that ADA may affect the PINK1 accumulation, but this can be tested experimentally.
2. In Fig. 3, artificial tethering of mitochondria and ER clearly accelerate the Parkin recruitment. What is the actual role of mitochondria-ER contact sites for Parkin recruitment? Is the Ca²⁺ ion somehow involved in this?
3. There are several mitochondria-ER tethering factors reported. Does ADA specifically disrupt the VAPB/PTPIP51-mediated mitochondria-ER contact sites formation? For example, the efficient Parkin recruitment under ADA inhibition is cancelled by VAPB or PTPIP51 knockdown?
4. In Fig. 4, the authors showed that ADA delays the GFP-LC3 accumulation on mitochondria along with the inhibition of efficient Parkin recruitment. Does this slight delay of LC3 accumulation affect the overall efficiency of mitophagy? In contrast, the data showing accelerated Parkin recruitment by ADA inhibition in EtBr-induced chronic mitochondrial dysfunction (Fig. 5) is more robust. Under these conditions, does ADA inhibition enhance the clearance of mitochondria at the endpoint?

5. In Fig. 6, the authors showed that ADA formation is prolonged and Parkin recruitment is delayed in galactose media. The authors reasoned that this phenomenon may protect mitochondria from mitophagy when cells rely on mitochondria for their energy source. However, CCCP disables the OXPHOS-mediated ATP production anyway, so I feel that still physiological meaning is not clearly demonstrated. It might be interesting to see whether galactose media delays Parkin recruitment in other mitochondria stressed conditions such as Actinonin or dOTC expression (mitochondrial proteotoxicity).

6. In extended Fig. 5D, it is better to test whether CCCP treatment can depolarize mitochondria in galactose media in a similar manner as in glucose media.

Referee #3:

The manuscript by Fung et al examines the relationship between mitochondrial actin accumulation (ADA: acute damage-induced actin) and mitophagy and ER-mitochondria contacts. They find that inhibition of actin accumulation results in a speeding up of Parkin recruitment. They attribute this phenotype to the observation that ADA disrupts ER-mitochondrial contacts. Treatments that increase ER-mitochondrial contacts can mitigate the delay in Parkin recruitment that is caused by inhibition of ADA. They interpret these results as ADA being a protective mechanism that delays mitophagy following acute and chronic damage.

This manuscript contains a number of interesting observations, but the authors should consider alternative explanations. They interpret the data as showing that ADA's normal function is to delay Parkin recruitment and mitophagy during acute mitochondrial damage. Alternative models are plausible and should be discussed. The key observation is that Arp2/3 inhibition by CK666 results in faster CCCP-induced Parkin recruitment. This observation could simply suggest that CCCP + CK666 results in a higher level of cellular/mitochondrial stress, and results in faster Parkin recruitment. That explanation seems more intuitive than proposing that the role of ADA is to delay mitophagy.

It also seems counter-intuitive to suggest that during chronic mitochondrial damage (ethidium bromide treatment), actin filaments are assembled to prevent Parkin recruitment and mitophagy. Ethidium bromide treatment causes damage to mtDNA, and it is unclear whether keeping mitochondria around under such stress conditions is beneficial.

In the Discussion, the role of ER-mitochondrial contacts in Pink1 stabilization is suggested. It would be informative to directly examine this issue.

Another caveat with the study is that the experimental approach is highly dependent on drugs and is quite blunt.

The study would also benefit by a clearer and more detailed discussion of cellular processes. Why would ADA inhibit ER-mitochondrial contacts, given that previous papers suggested that contacts are facilitated by the cytoskeleton? Why would increasing contacts increase Parkin recruitment, given that Parkin recruitment depends on stabilization of Pink1 and there is little reason to think actin would be important?

Minor points:

- 1) In abstract, the following typo should be corrected: Acute Damaged-induced Actin.
- 2) In Figure 2A, it is difficult to appreciate any differences in ER-mitochondrial overlap in the images.
- 3) Specificity controls for the PLA assay should be shown.

- CCCP was predominantly used in the study for Parkin induction. Alternative conditions for Parkin induction need to be applied (referee #1, standfirst, major point 2; referee #2, point 5; referee #3):

We thank all the reviewers for this suggestion. In the modified manuscript we include a new Figure (Fig EV2) outlining the role of ADA in Parkin recruitment upon Antimycin/Oligomycin (A/O). We show that:

- A/O induces Arp2 dependent ADA in HeLa albeit with a significantly milder depolarization (Fig EV2).
- A/O induced Parkin recruitment is significantly faster in Arp2 inhibited HeLa cells (Fig EV2).
- A/O induced Pink1 stabilization and activity is also higher in Arp2 inhibited HeLa cells (Fig 6)

With these results we show that ADA influences Parkin recruitment upon acute mitochondrial depolarization (CCCP and Antimycin/Oligomycin).

Additionally in Figure 7 we show that more stable ADA-like filaments induced during chronic mitochondrial depolarization blocks Parkin recruitment and mitophagy which are reversed upon Arp2/3 inhibition using CK666.

- The proposed effect of ADA on PINK1 accumulation should be tested experimentally (referee #1, major point 3; referee #2, point 1; referee #3, paragraph 4):

This was an excellent suggestion, and the results turned out to be very interesting. We show that Arp2/3 inhibition using CK666 significantly increases the levels of GFP-Pink1 in HeLa cells upon CCCP and A/O treatments at 20 minutes time-points (Fig 6). Further we also show an increase in levels of Pink1-induced phospho-ParkinS65 and phospho-UbiquitinS65 at 20 and 40 minutes of mitochondrial insults (using CCCP and A/O) in Arp2/3 inhibited HeLa cells compared to the control cells (Fig 6). These results point to the fact that ADA might have additional acute effects on Pink1 stabilization and/or activity upstream of its modulation of Parkin recruitment. Additional experiments will be needed to validate whether the ADA-mediated regulation of Pink1 stabilization/activity is also through modulation of ER-Mitochondrial contacts (ERMC).

- Mitochondria-ER contacts and their role in Parkin recruitment need to be better characterized (referee #1, major points 5 and 7; referee #2, points 2 and 3)

We thank the reviewers for bringing up this point.

-In a new set of experiments we show that both in MEF and HeLa cells Parkin recruitment occurs primarily close to regions that are positive for both ER and Mitochondria (Fig 2). Interestingly the evaluation also shows that a significant percentages of the Parkin puncta in both cell types are positive for only ER enhancing the role of ER in Parkin recruitment.

- Overexpression of VAP-B (Fig 3) and ER-Mito synthetic tether (Fig 3) induced faster and higher Parkin recruitment on mitochondria.

- Overexpression of PM-Mito synthetic tether which reduces availability of ER for mitochondria diminished Parkin recruitment on mitochondria (Fig 3).

- In a new data set (Fig 4) we evaluate the role of MFN2 and VDAC1 on Parkin recruitment. Interestingly both MFN2 and VDAC1 have been shown to control ERMC

and mitochondrial Parkin recruitment. However we additionally show that inducing ERMC could significantly rescue Parkin recruitment defect in MFN2 KD HeLa cells but not in VDAC1 KD cells. Thus our new data shows that Parkin recruitment occurs in regions positive for ER and Mitochondria and that VDAC1 on mitochondria (most probably at ERMC) is required for Parkin recruitment upon CCCP treatment in HeLa cells.

- Caveats mentioned by referee #3 need to be acknowledged and discussed.
We have added this to the new discussion section.

We are aware that elucidating the entire mechanism may be unfeasible, but we also think that assessing the involvement of Mfn2, VDAC1 and SYNJ2BP in the link between ADA and ER-mitochondria contacts as per referee #1 will significantly strengthen the manuscript.

In a new data set (Fig 4) we evaluate the role of MFN2 and VDAC1 on Parkin recruitment. Interestingly both MFN2 and VDAC1 have been shown to control both ERMC and mitochondrial Parkin recruitment. However we additionally show that inducing ERMC could significantly rescue Parkin recruitment in MFN2 KD HeLa cells but not in VDAC1 KD cells. Thus our new data shows that Parkin recruitment occurs in regions positive for ER and Mitochondria and that VDAC1 on mitochondria is required for Parkin recruitment upon CCCP treatment in HeLa cells.

Referee #1:

In this manuscript, Fung and colleagues investigate the contribution of a transient cloud of actin filaments, termed ADA, at mitochondria during PARKIN-dependent mitophagy. This study is a significant follow-up, as this group was the first to identify and characterize ADA and its role in various mitochondrial functions. Using microscopy, the authors demonstrate that inhibition of Arp2/3-complex activity, or other actin-related proteins involved in ADA formation, enhances the recruitment of Parkin and LC3 to mitochondria upon CCCP/FCCP-mediated mitochondrial damage. They report that CCCP treatment disrupts mitochondrial-ER contact sites (MERCs), coinciding with the accumulation of peri-mitochondrial actin. Interestingly, this disruption is rescued upon inhibition of Arp2/3-complex activity or ADA, suggesting a direct role of ADA in MERC biology. Similar to Arp2/3 inhibition, the stimulation of functional MERCs restores the kinetics of Parkin and LC3 recruitment upon both acute and chronic mitochondrial damage in CCCP- and EtBr-treated cells, respectively. Finally, the authors show that an increased dependence on mitochondrial metabolism, through growth in galactose media, prolongs damage-induced mitochondrial actin and impairs mitochondrial Parkin recruitment.

Overall, this study provides an intriguing link between critical mitochondrial functions, including Parkin-dependent mitophagy and mitochondria-ER contacts, adding new complexity to these mechanisms. Identifying and characterizing the role of ADA in Parkin-dependent mitophagy is novel and interesting, and the data support the idea that

ADA controls mito-ER contacts formation delaying Parkin recruitment to mitochondria, and potentially the whole mitophagy process.

However, further experiments are needed to strengthen the manuscript. In particular, a further characterization of mitochondrial-ER contact site and mitophagy analysis, and additional image quantification (ADA formation...).

The authors only use systems where they overexpress PARKIN and treat cell with FCCCP which limits the study. The use of other stimuli for Parkin-dependent activation are necessary. The authors should also investigate if these findings also apply to other types of mitophagy. Moreover, the authors should propose mechanisms by which ADA controls MERCs. Finally, particular attention should be given to the statistical analysis and the number of experiments presented.

Major points

- While elucidating the full mechanism can be challenging, presenting some evidence, even if incomplete, on how ADA controls mitochondria-ER contacts would be beneficial. For example, Mclelland et al. (PMID: 29676259) proposed that Mfn2 is ubiquitinated early on during the PINK1-PARKIN dependent mitophagy to decrease MERCs and allow mitophagy. The authors should discuss this paper and check the levels of Mfn2 or Ub-Mfn2 in their different systems, as this could be useful for understanding the mechanism. Other proteins involved in this process, such as VDAC1 and SYNJ2BP, should also be checked.

We thank the reviewers for citing this published article and we agree it is intriguing as to how ADA control ER-Mitochondrial contacts. Our PLA and EM analysis reveals that within 5 minutes of its induction, ADA significantly disrupts close contacts (0-40 nm) between the two organelles. This time frame is well before Pink1 stabilization (about 20 min: **Fig 6**) and Parkin recruitment (20-40 minutes; **Fig 1**). Since Arp2 mediated branched actin assembly have been shown to induce force especially well characterized during cell migration, we draw a parallel here and hypothesize that something similar is being done here. Additional experiments are needed to evaluate and measure the force generated at these organelle membrane by ADA, but it might be unlikely that ADA targets any specific tether for this purpose.

Mfn2 has been an interesting molecule in the context of Pink1/Parkin mitophagy. Indeed Chen et al., Science 2013 have shown that Mfn2 KD blocks Parkin recruitment and Mclelland et al., eLife, 2018 have shown that Mfn2 is rapidly degraded for mitophagy to proceed. Since Mfn2 degradation in the latter manuscript mainly happens post 60 minutes (in many cases 2-4 hours) of CCCP

treatment, we think that ADA mediated ERMC modulation is a much acute response to this (within 5 minutes). In fact we see Mfn2 depletion at 90 and 120 minutes of CCCP treatment (**Fig 5C, D**) which is significantly increased upon Arp2/3 inhibition. In fact in earlier time points (upto 60 minutes) we do not see significant depletion of Mfn2 (**Fig R1 in previous page**).

Additionally we show that while both Mfn2 and VDAC1 depletion can dampen Parkin recruitment after CCCP stimulation (**Fig 4**), inducing ER-mito contacts through overexpression of the constitutive tether can speed up Parkin recruitment in Mfn2 KD scenario but not in VDAC1 KD cells (**Fig 4**). This further shows that Mfn2 regulation of Parkin recruitment is through modulation of ERMC while VDAC1 is required for efficient Parkin recruitment downstream of ERMC.

- Can the authors reproduce their data under other conditions that stimulate Parkin-dependent mitophagy, aside from CCCP? Additionally, since endogenous Parkin is expressed in some cell lines, it would be interesting to analyze the recruitment of endogenous Parkin rather than overexpressed Parkin. Moreover, is this mechanism conserved across other types of mitophagy induction? These points are crucial for broadening the applicability of the study's results.

We thank the reviewer for the suggestions. We have previously shown that a multitude of mitochondrial insults induce ADA in Arp2/3 dependent manner (Chakrabarti R et al., 2022) and it has been shown that combined treatment with Antimycin A/Oligomycin (A/O) induces Parkin recruitment with a similar kinetics to that of CCCP (Hung et al., Sci Adv. 2021). In the modified manuscript we present data (**Fig EV2**) that A/O treatment induces Arp2/3 dependent ADA albeit with a significantly slower and lower mitochondrial depolarization. Additionally we show that inhibition of ADA through CK666 significantly speeds up Parkin recruitment in A/O treated HeLa cells (**Fig EV2**).

Fig R2: Immunofluorescence showing Parkin (green) and Tom20 (plum) in untreated and CCCP treated SH-Sy5Y cells

We agree that the entire study was conducted with overexpressed Parkin however we tried using documented cell lines to immuno-stain for endogenous Parkin. While the Parkin antibody gave reproducible results for western blot assays, Immunofluorescence in SH-SY5Y cells showed irregular staining of Parkin on mitochondria in control cells and no detectable change upon CCCP treatment (**Fig R2**). We believe that we have to try out a few other Parkin antibodies tested to validate the results in an endogenous system which is beyond the scope of this present study.

On the question regarding other types of mitophagy induction, we show that inducing

acute mitochondrial damage through uncoupler (**CCCP; Fig 1**), ETC poisons (**Antimycin A/Oligomycin; Fig EV2**) or chronic mitochondrial dysfunction through mtDNA depletion (**Fig 7**), ADA inhibits Parkin recruitment and downstream events of mitophagy. Additionally in the revised manuscript we provide evidence (**Fig 7**) that removal of ADA like filaments promotes mitophagy downstream of Parkin and LC3 recruitment. While in this manuscript we wanted to restrict our study towards Parkin mediated mitophagy, future experiments are needed to extend this mechanism to other non-ubiquitin mediated mitophagy pathways.

- The authors currently analyze the speed of overexpressed Parkin and LC3 recruitment to mitochondria upon CCCP treatment. It would be insightful to investigate upstream steps of Parkin recruitment at endogenous levels, such as PINK1 accumulation and phospho-ubiquitin.

This was an excellent suggestion by the reviewers in general and we have some insightful results on ADA mediated PINK1 regulation that is presented in the modified **Fig 6**. We show that Arp2/3 inhibition using CK666 significantly increases the levels of GFP-PINK1 in HeLa cells upon CCCP and A/O treatments at 20 minutes time-points (**Fig 6**). Further we also show an increase in levels of PINK1-induced phospho-ParkinS65 and phospho-UbiquitinS65 at 20 and 40 minutes of mitochondrial insults (using CCCP and A/O) in Arp2/3 inhibited HeLa cells compared to the control cells. These results point to the fact that ADA might have additional acute effects on Pink1 stabilization and/or activity upstream of its modulation of Parkin recruitment. These results open up the possibility of exploring the possible roles of ADA and/or ERMC in Pink1 dynamics. Additional experiments will be needed to validate whether the ADA-mediated regulation of Pink1 stabilization/activity is also through modulation of ERMC.

- Does Parkin is recruited at mito-ER contacts upon stimulation and how it is affected by Arp2/3 and CCCP or increased MERCs?

We carried out a spatial characterization of Parkin puncta in MEF and HeLa cells and the data is presented in modified **Fig 2A-F**. Our investigation reveals that 75-80% of Parkin puncta across the two cell lines tested were positive for the presence of both ER and mitochondria. Interestingly 5-10% of these puncta co-localized with only ER devoid of mitochondria raising the importance of ER in the whole process. So in conclusion we do see Parkin recruitment at ER-Mito contact sites in both MEF and HeLa cells.

In **Fig 2G-J**, through PLA assay and EM analysis we show that within 5 minutes of CCCP treatment there appears to be a significant decrease in close contacts between ER and mitochondria in an ADA dependent (Arp2/3 dependent) manner. Consequently we see that Arp2/3 inhibition causes faster Parkin recruitment (**Fig 1**). Additionally increasing ERMC through VAP-B overexpression or ER-RFP-Mito overexpression speeds up Parkin recruitment without affecting ADA (**Fig 3**). On the other hand reducing

ERMC though expression of PM-RFP-Mito or knocking down Mfn2 and VDAC1 significantly reduces Parkin recruitment upon CCCP treatment in HeLa cells (**Fig 4**). Our data points to the fact Parkin is recruited at ERMC and that ADA disrupts ERMC to slow down Parkin recruitment

- The analysis of mitochondria-ER interaction needs to be strengthened. While the authors used PLA, this system is highly dependent on the pair of antibodies used, and it seems evident now that tethering complexes can behave differently depending on the cellular context. For example, TOMM20 should not be used in PLA as it is one of the first targets of PARKIN during CCCP treatment. The authors should use transmission electron microscopy to validate their PLA results in Figures 2 and 5. Additionally, the authors should analyze mitochondria-ER contacts under their experimental conditions with galactose and EtBr treatment.

We thank the reviewer for the suggestion, and we have now carried out a detailed EM analysis for ADA mediated ERMC disruption in the context of CCCP (modified **Fig 2I-J; EV3G**) and EtBr treatments (**Fig 7C-D; EV5D, E**). Our results from the EM analysis aligns with that shown in the PLA assay showing that 5 min of CCCP treatment disrupts ERMC (contacts < 30 nm) in an Arp2/3 dependent manner. Similarly ADA like filaments upon EtBr treatment cause significant reduction in ERMC which are established upon removal of these filaments with CK666.

While we acknowledge that PLA assay system is highly dependent on the choice of antibody pairs being used, we performed the assay with two independent pairs of proteins. While VAP-B and PTP51 represent bonafide ERMC pair, the choice of VAP-B and Tom20 was to characterize ERMC in an unbiased manner. While we agree that Tom20 is one of the targets of Parkin, we perform the PLA assay within 5 minutes of CCCP treatment much before Parkin or Pink1 is recruited to mitochondria. We believe that the results from the PLA assay coupled with the ERMC analysis from TEM robustly show that ADA acutely causes a disruption of ERMC.

- In EV3X, while mitochondrial calcium analysis is appreciated, it does not provide evidence that the contacts are increased or decreased upon tethering mitochondria-ER or mitochondria-PM, respectively. A microscopy analysis of these contacts should be performed. Additionally, mitochondrial localization, morphology, and membrane potential in PM-Mito tether conditions should be assessed to demonstrate that mitochondria are indeed localized at PM and are functional.

The artificial synthetic Mitochondrial-ER tether (ER-RFP-mito) was developed by Gyorgy Hajnoczky's group and characterized in the following publication (Csordas et al., JCB 2006; PMID: 16982799). The construct have been used by several groups including a study by CC Miller's group (Gomez-Suaga et al., Curr. Biol., 2017; PMID:

28132811) and more recently by Gyorgy Csordas' group (Nichtova S et al., Circ. Res. 2023; PMID 37057625) to induce ER-Mito and ER-SR artificial tethering respectively.

The OMM-PM constitutive linker (PM-RFP-mito) was also developed by Gyorgy Hajnoczky's group and was used in the study by Wolfgang F. Graier and Roland Malli's group (Naghdi S et al., JCS 2010; PMID: 20587595) where they characterized that expression of the construct induced clustering of mitochondria at the plasma membrane (Fig 6 of their manuscript PMID: 20587595) and restricted the overall motility of mitochondria (Fig 7 of their manuscript PMID: 20587595). They further showed that ER to mitochondria calcium transfer is reduced but the mitochondria retain the ability to uptake calcium from extra-cellular source (Fig 8 of their manuscript PMID: 20587595). The construct was more recently used by the Hajnoczky group to restrict ER-Mito overlap (Katona M et al., Nat. Commun., 2022; PMID: 36351901). We have added all these references in the modified manuscript.

- The authors should decrease mitochondria-ER contacts, and investigate what happens to ADA, Parkin, and LC3 recruitment under basal and CCCP conditions. For now, all their rescue experiments are based on overexpressing systems (tether, VAPB...)

We agree that initially most of our data was to increase ER-Mito tethering (VAP-B and ER-RFP-Mito overexpression) and evaluate the kinetics of Parkin and LC3 recruitment following CCCP treatments. We now have at least three ways in which we reduce ERM and report ADA and Parkin kinetics.

1. We use Mito-RFP-PM construct to tether mitochondria to plasma membrane. This does not hamper the kinetics of ADA but significantly reduces Parkin and LC3 recruitment to depolarized mitochondria (**Fig 3F,G**)
2. We knock down MFN2 and VDAC1 which have been shown to reduce ERM. The knock down of both these factors reduced Parkin recruitment on depolarized mitochondria. Further use of ER-RFP-Mito synthetic tether could overcome the effect only in MFN2 KD cells but not in VDAC1 KD cells (**Fig 4**).

Thus these new data show that while MFN2 affects Parkin recruitment in an ERM dependent way, the physical presence of VDAC1 is required to recruit Parkin on depolarized mitochondria in HeLa cells upon CCCP stimulation (**Fig 4**)

- The authors changed cell lines without explanation in the manuscript, which is confusing. While analyzing U2OS and HeLa cells is acceptable since they have been characterized for ADA and Parkin recruitment, the use of MEFs is difficult to understand. In Figure 2C-F, PLA in MEFs is confusing as previous experiments have not been conducted on these cell lines to show ADA or Parkin recruitment. If MEFs are to be used, ADA, delay in Parkin recruitment, and LC3 should be demonstrated in these cell lines. Additionally, in EV4, the characterization of EtBr

is done in MEFs while the functional assay is done in HeLa cells, which is inconsistent. Similarly, in Figure 6, the functional assay is done in U2OS while the characterization of galactose was performed in HeLa cell. The characterization should be done in the corresponding cell line where the functional assay is performed.

We apologize for this confusion and thank the reviewer for pointing this out. We have previously shown that ADA can be induced in MEF cells following CCCP and/or ETC poisons (Chakrabarti et al., JCB 2022) in an Arp2/3 dependent manner. In the modified manuscript we image and quantify Parkin recruitment in control and Arp2/3 inhibited MEF cells (**Fig EV1H, I**). While Parkin recruitment in MEF cells is not as homogenous and rapid as seen in HeLa or U2OS cells, Arp2/3 inhibition indeed increases Parkin

positive mitochondria in MEF cells as well. This shows that ADA is important in modulating Parkin recruitment in MEF cells (**Fig EV1H, I**). We therefore present the PLA assays and EM analysis for ERMC in MEF cells (**Fig 2G-J**)

Regarding the EtBr studies, we apologize for the mislabeling of the figure legends. All analysis in that Figure (**currently Fig 7**) was conducted in MEF cells including the initial analysis of OCR and mtDNA depletion. We additionally present that EtBr treated MEF cells show increased mitophagy upon Arp2/3 inhibition (**Fig 7I-J**).

Regarding the galactose assay, we now have removed the U2OS data and kept the HeLa cell data as modified **Figure 8**. We had previously done the OCR analysis in the U2OS cells as well (**Fig R3**) but since we did the functional analysis of Parkin

recruitment in the HeLa cells we chose to keep the **Figure 8** with analysis from HeLa cells only.

Minor Points

- A significant number of quantifications present statistical analysis while the experiments have only been performed twice (e.g., EC1F, G, H, I; 2F, EV2F, EV3, EV4, EV5, Fig4, Fig5, and Fig6). For all these panels, three independent experiments should be conducted to ensure the robustness of the statistical analysis. Additionally, it would be beneficial if the figure legends included the total number of cells analyzed and the statistical tests used (even if these tests are described in the Materials and Methods section).

We apologize for this and have updated all the experiments to have at least three independent repeats and/or coverslips imaged. Additionally we have added the number of cells/experiments performed in the figure legends.

- The authors need to provide clearer explanations of how certain quantifications were performed and what they represent. For example, what does "Parkin on mitochondria (a.u.)" or "Onset of Parkin at mitochondria (min)" mean? How are these values calculated? What does a.u. stand for? What is the threshold for considering a cell to have Parkin at mitochondria? Is it based on individual mitochondria? Should the entire mitochondrion be Parkin-positive or just a part of it? Additionally, in Figure 3E and G, the method for quantifying Parkin at mitochondria appears to change. Can the authors explain why and clarify what is represented by "% cells with Parkin on mitochondria" (e.g., all mitochondria, the majority of mitochondria)? The concept of "Parkin-positive mitochondrial area" also needs explanation. These quantifications are critical for the conclusions of the paper and need to be significantly better explained.

The quantifications stating "Parkin on mitochondria" and "onset of Parkin at mitochondria" have been defined in the methods section.

When calculating % cells having Parkin on mitochondria we consider cells as positive even if a part of the mitochondrial network is positive for Parkin. In Fig 3E we counted cells that showed Parkin on mitochondria and since we were doing time kinetics we could clearly see that % cells having Parkin positive mitochondria was significantly higher in the early time points for cells overexpressing ER-Mito tether compared to control cells, but at 45 minutes were almost equal. In Fig 3G we carried out an end point assay where % cells having Parkin positive mitochondria would be similar across all conditions. In this case we now evaluated the extent of mitochondrial area that was positive for Parkin. As the results indicate though all cells showed Parkin positive mitochondria, the amount of mitochondrial area positive for Parkin was significantly higher in ER-RFP-mito overexpressing cells and significantly lower in PM-RFP-mito overexpressing cells compared to control cells. Therefore there was this shift in quantification method from Fig 3E to 3G.

- During galactose treatment and in EtBr-treated cells, the authors should investigate if Parkin and LC3 recruitment can be reversed by overexpressing their artificial mitochondria-ER tether or VAPB overexpression as shown in Figure 6.

This is an excellent suggestion. The idea of this figure was to examine whether the persistence of ADA had an effect on the overall kinetics of Parkin recruitment. Since galactose treated cells had a significantly persistent ADA, we tried to quantify Parkin recruitment in these cells compared to glucose treated cells. Indeed we show that a slightly prolonged ADA in galactose treated cells further delays Parkin recruitment in Arp2/3 dependent manner. Further we also show that persistent ADA in EtBr treated

cells completely block Parkin recruitment which could be reversed with Arp2/3 inhibition. Overall our results show that persistence of ADA could significantly delay (transient ADA) or block (persistent ADA) Parkin recruitment in an ERM1 dependent manner.

- Quantification of ADA needs to be performed in EV3E, EV4C, 6A

We have quantified ADA upon expression of ER-Mito and PM-Mito tethers and have presented it in modified **Figure EV4E**

- EV1 H, I: I don't see Parkin recruitment in the images shown. Why do the authors choose 30 min of FCCCP treatment for this condition?

In the light of new experiments we have removed this figure panel.

- The presentation of images should be optimized. Some images are missing scale bars, zoom areas are not consistently represented, and some zoomed areas are missing.

We have tried to rectify this in the modified manuscript

- The reason for the change in quantification methods from Figure 3E to 3G should be explained.

Initially we imaged Parkin recruitment live on a spinning disc confocal which allowed imaging for long time without losing focus. However due to a change of research place and equipment subsequent Parkin recruitment images had to be done using fixed cells on a Zeiss Airyscan that is not made for long-term focus steady imaging.

In Figure 3E, we quantify the percentage of cells having parkin on mitochondria. We consider cells as "positive" even if a part of mitochondrial network or single mitochondria have Parkin accumulated on it. We saw a significant difference in the % cells with this categorization in control and ER-RFP-mito treated cells. However at 45 minutes the % cells with Parkin positive mitochondria is indistinguishable between the two groups.

In Fig 3G, we did an end point assay instead of time-course. Now at this 45 minutes when we assessed the percentage of mitochondrial area positive for Parkin in the different treatment conditions, we found that ER-RFP-mito had significantly "more" Parkin positive mitochondrial area compared to the control cells (even when % cells in both cases were same) and PM-RFP-mito expressing cells. Thus in Fig 3G we wanted to show that though the cells in all the three groups have similar percentages of cells having Parkin on mitochondria, the actual amount of mitochondrial area positive for Parkin was still significantly different.

- Mitochondria-lysosome interactions could be strengthened.

Since the initial characterization did not show significant difference between mitochondria-lysosome contacts following ADA stimulation, we invested more effort to characterizing the ER-Mitochondrial contacts upon ADA stimulation.

- Autophagy fluxes (LC3-I vs LC3-II, p62) could be analyzed by immunoblots to complement the immunofluorescence of LC3 overexpression.

We have now added LC3 blots and quantified them presented as part of Figure 5

- The authors should discuss previous literature related to mitochondria-ER contacts during mitophagy that is currently missing from their discussion and introduction.

We have now added this in the modified discussion

Referee #2:

Comments to the authors

In a previous study, the authors identified that mitochondrial depolarization generates a transient cloud of actin filaments around depolarized mitochondria (ADA), and this phenomenon is critical for a metabolic shift from oxidative phosphorylation to glycolysis upon mitochondrial dysfunction. Here, they found another possible function of ADA formation. They found that ADA prevents the efficient Parkin translocation to dysfunctional mitochondria. Their series of data indicated that the disruption of mitochondria-ER contact sites by ADA may explain the prevention of the efficient Parkin translocation by ADA. Each experiment was carefully performed and the overall data are quite convincing. Therefore, I fully agree that ADA formation does indeed prevent the mitochondria-ER contact site formation and the efficient Parkin translocation to dysfunctional mitochondria. These are very interesting findings. However, the actual delay of Parkin recruitment is only about 15 minutes, and the actual effects on the efficiency of mitochondrial clearance are not shown. Therefore, the physiological significance of the slight delay in Parkin recruitment remained unclear. Please see specific comments below.

Major points

1. The time scale of Fig. 2A and 2B should be seconds, as mitochondria do not appear to be fragmented yet (also given the time scale of similar experiments conducted in extended Fig. 2D-F). If this is the case, the disruption of the ADA-mediated mitochondria-ER contact sites is a transient event that may recover within 10 min (Fig. 2B). How does this transient event affect the Parkin

translocation at 30-60 min? The authors mentioned the possibility that ADA may affect the PINK1 accumulation, but this can be tested experimentally.

We thank the reviewer for raising this issue and we agree that Fig 2A, B (old manuscript) was indeed seconds instead of minutes. However from these Airyscan images we cannot confirm if the re-establishment of overlap that we are seeing are indeed recovery of close contacts. In the light of the analysis from the EM images conducted for ERMIC, we see that there is a drastic disruption in close ERMIC upon 5 minutes of CCCP treatment in an ADA dependent way (**Fig 2I-J**). However further studies are required to carefully map the recovery of these close contacts if any in the later time-points either through PLA studies and/or EM analysis. We plan to carry them out in future studies.

As per the suggestion of the reviewer we evaluated the role of ADA on Pink1 accumulation upon mitochondrial damage. Indeed there appears to be a higher Pink1 accumulation acutely in the absence of ADA (CK666 treatment) both with CCCP and Antimycin/Oligomycin treatment. We also show that there is higher Pink1 activity characterized by an increase in phospho-ParkinS65 and phospho-ubiquitinS65 at these time points for both treatment conditions. These data are now presented as **Figure 6** in the modified manuscript. These data shows that ADA acutely affects Pink1 stabilization and/or activity, and further studies are required to ascertain if this is through modulation of ERMIC.

2. In Fig. 3, artificial tethering of mitochondria and ER clearly accelerate the Parkin recruitment. What is the actual role of mitochondria-ER contact sites for Parkin recruitment? Is the Ca²⁺ ion somehow involved in this?

We thank the reviewer for raising this point.

1. Careful imaging in MEF cells and HeLa cells show that Parkin recruitment indeed proceeds at ER-Mito contact regions (**Fig 2A-F**).
2. Further we show that while increasing ERMIC (through VAPB and ER-RFP-Mito overexpression) accelerated Parkin recruitment, decreasing ERMIC (through Mito-RFP-PM overexpression, MFN2 KD, VDAC1 KD) reduced Parkin recruitment following CCCP treatment (**Fig 3, Fig 4**)
3. Additionally we show that VDAC1(possibly at ERMIC) is required for successful Parkin recruitment at ERMIC (**Fig 4**)

While we showed that increase in cytosolic Ca²⁺ is required for generation of ADA through activation of Rac1, WAVE and Arp2/3 (*Fung et al., Curr. Biol.*), further studies are required to identify whether cytosolic or mitochondrial calcium is required for Parkin recruitment beyond its role in ADA generation.

3. There are several mitochondria-ER tethering factors reported. Does ADA specifically disrupt the VAPB/PTPIP51-mediated mitochondria-ER contact

sites formation? For example, the efficient Parkin recruitment under ADA inhibition is cancelled by VAPB or PTPIP51 knockdown?

This is an interesting thought raised by the reviewer. From the PLA assay we see a reduction in VAP-B-PTPIP51 (**Fig 2G-H**) interaction, but this might simply be due to an overall decrease in ER-Mitochondrial proximity as evaluated by PLA assays through VAP-B-Tom20 (**Fig EV3E, F**). Further our EM analysis reveals that there is a drastic disruption in the density of close contacts between ER and mitochondria (**Fig 2I-J**), further pointing out to the fact that ADA might be causing a forceful disruption of ER-mitochondrial contacts rather than specifically favoring any particular tether.

However our analysis through MFN2 KD and VDAC1 KD cells interestingly show that while restoring close contacts in MFN2 KD can rescue defects in Parkin recruitment, absence of VDAC1 precludes Parkin recruitment even upon ER-Mitochondrial tethering (**Fig 4**). These new results point to the fact that VDAC1 is physically required at ERMC to recruit Parkin following CCCP treatment in HeLa cells.

4. In Fig. 4, the authors showed that ADA delays the GFP-LC3 accumulation on mitochondria along with the inhibition of efficient Parkin recruitment. Does this slight delay of LC3 accumulation affect the overall efficiency of mitophagy? In contrast, the data showing accelerated Parkin recruitment by ADA inhibition in EtBr-induced chronic mitochondrial dysfunction (Fig. 5) is more robust. Under these conditions, does ADA inhibition enhance the clearance of mitochondria at the endpoint?

We thank the reviewer for this query, and we now show new evidence to answer this. We transiently infected MEF cells with a mitophagy reporter construct previously characterized and used by Chris Shoemake's group (Delgado et al., EMBO J, 2024; PMID: 38177312). These cells were treated with EtBr for 5 days to chronically induce ADA. While EtBr treatment showed the generation of some mitolysosome (Red only puncta), removal of ADA like filaments using CK666 treatment significantly increased the density of mitolysosome. CK666 treatment in the control cells did not show generation of any mitolysosome. These data presented in **Figure 7I-J** confirms that during chronic mitochondrial damage, persistent ADA like filaments significantly restrict mitophagy and removal of dysfunctional mitochondria which can be reversed by removal of ADA through inhibition of Arp2/3.

5. In Fig. 6, the authors showed that ADA formation is prolonged and Parkin recruitment is delayed in galactose media. The authors reasoned that this phenomenon may protect mitochondria from mitophagy when cells rely on mitochondria for their energy source. However, CCCP disables the OXPHOS-mediated ATP production anyway, so I feel that still physiological meaning is not clearly demonstrated. It might be interesting to see whether galactose media

delays Parkin recruitment in other mitochondria stressed conditions such as Actinonin or dOTC expression (mitochondrial proteotoxicity).

While this is an excellent suggestion by the reviewer, we feel that we might have to standardize conditions with these new treatments (Actinonin, dOTC) to first check whether ADA is generated in the same time frame as CCCP and/or Antimycin/Oligomycin to further characterize their role in Parkin recruitment kinetics in galactose media. In the light of the new data presented, we feel that this is beyond the scope of this manuscript

6. In extended Fig. 5D, it is better to test whether CCCP treatment can depolarize mitochondria in galactose media in a similar manner as in glucose media.

While we have not measured TMRE kinetics following CCCP treatment in galactose cells but considering induction of ADA and its prolonged kinetics following CCCP treatment, we think that the mitochondria depolarizes to a similar extent in galactose treated cells compared to glucose-treated cells. Moreover, Arp2 KD in galactose treated cells significantly speeds up parkin recruitment in galactose-treated cells (Fig 8)

Referee #3:

The manuscript by Fung et al examines the relationship between mitochondrial actin accumulation (ADA: acute damage-induced actin) and mitophagy and ER-mitochondria contacts. They find that inhibition of actin accumulation results in a speeding up of Parkin recruitment. They attribute this phenotype to the observation that ADA disrupts ER-mitochondrial contacts. Treatments that increase ER-mitochondrial contacts can mitigate the delay in Parkin recruitment that is caused by inhibition of ADA. They interpret these results as ADA being a protective mechanism that delays mitophagy following acute and chronic damage.

This manuscript contains a number of interesting observations, but the authors should consider alternative explanations. They interpret the data as showing that ADA's normal function is to delay Parkin recruitment and mitophagy during acute mitochondrial damage. Alternative models are plausible and should be discussed. The key observation is that Arp2/3 inhibition by CK666 results in faster CCCP-induced Parkin recruitment. This observation could simply suggest that CCCP + CK666 results in a higher level of cellular/mitochondrial stress, and results in faster Parkin recruitment. That explanation seems more intuitive than proposing that the role of ADA is to delay mitophagy.

Fig R4: TMRE time course in CCCP and CK666/CCCP treated HeLa cells

We thank the reviewer for bringing up this point and we added it to the discussion section. We also checked for TMRE kinetics in cells treated with CCCP and CK666 + CCCP. The kinetics of TMRE loss is identical (**Fig R4**) in the wee hours of the treatment but we agree that the cells could experience additional stress due to the combination of CCCP and CK666 at subsequent time points which are not adequately reflected in the time course of TMRE measurements. We also feel that since ERMC enhancement could over-ride the effects of ADA and speed up Parkin and LC3

recruitment, points to the fact that ERMC plays an integral role in modulating Parkin mediated mitophagy which is regulated by ADA. Additional studies measuring other markers of cellular and mitochondrial stress in control and Arp2/3 inhibited cells would clarify this.

It also seems counter-intuitive to suggest that during chronic mitochondrial damage (ethidium bromide treatment), actin filaments are assembled to prevent Parkin recruitment and mitophagy. Ethidium bromide treatment causes damage to mtDNA, and it is unclear whether keeping mitochondria around under such stress conditions is beneficial.

This is an excellent point raised by the reviewer and we were equally surprised to see the preservation of mitochondrial mass upon EtBr treatment not only for the days presented in the manuscript but for months of EtBr treatment under completely non-existent mitochondrial membrane potential or OxPhos. These cells showed unchanged staining for Tom20 suggesting that mitochondria was indeed preserved. Interestingly Rho0 cells have been established permanently in several cell lines where these compromised and damaged mitochondria are preserved by the cells eternally. Till data we have not come across any literature citing functional relevance for the preservation of these defective mitochondria both in the short term (EtBr treatment) or long term (Rho0 cells). Further studies will clarify whether these persistent ADA-like filaments play a role in preservation of mitochondrial mass in these Rho0 cells and the functional outcome of such act.

However at least in the short-term condition (EtBr treatment) we find that removal of ADA like filaments not only re-enforces ERMC but also promotes Parkin and LC3 accumulation leading to a significant increase in mitophagy (**Fig 7**). Further experiments will be required to see whether this finally causes a significant loss of mitochondrial mass from these cells.

In the Discussion, the role of ER-mitochondrial contacts in Pink1 stabilization is suggested. It would be informative to directly examine this issue.

This was an excellent suggestion by the reviewers in general and we have some insightful results on ADA mediated Pink1 regulation that is presented in the modified **Fig 6**. We show that Arp2/3 inhibition using CK666 significantly increases the levels of GFP-Pink1 in HeLa cells upon CCCP and A/O treatments at 20 minutes time-points (**Fig 6**). Further we also show an increase in levels of Pink1-induced phospho-ParkinS65 and phospho-UbiquitinS65 at 20 and 40 minutes of mitochondrial insults (using CCCP and A/O) in Arp2/3 inhibited HeLa cells compared to the control cells. These results point to the fact that ADA might have additional acute effects on Pink1 stabilization and/or activity upstream of its modulation of Parkin recruitment. These results open up the possibility of exploring the potential role(s) of ADA in Pink1 dynamics. Additional experiments will be needed to validate whether the ADA-mediated regulation of Pink1 stabilization/activity is also through modulation of ERM.

Another caveat with the study is that the experimental approach is highly dependent on drugs and is quite blunt.

We agree that a lot of results were shown through the use of CK666 which is highly specific for Arp2/3 complex inhibition and can be implemented acutely as opposed to days of Arp2/3 KD. In several figures we have tried to re-validate the results with siRNA mediated knockdown and/or using over-expression systems.

1. The accelerated Parkin recruitment seen with the use of CK666 + CCCP has been validated using siRNA mediated knockdown of Arp2 (Fig 1D-F; EV1C-D) in two different cell lines (U2OS and HeLa cells). Knockdown of other key components of the ADA machinery (WAVE complex and FMNL formins) were also tested to show the effect which was similar to CK666 + CCCP condition.
2. The modulation of Parkin recruitment was tested modifying ERM either by increasing them (through VAPB or ER-RFP-Mito overexpression) or by reducing them (through PM-RFP-Mito overexpression and siRNA mediated knockdown of MFN2 and VDAC1)

The study would also benefit by a clearer and more detailed discussion of cellular processes. Why would ADA inhibit ER-mitochondrial contacts, given that previous papers suggested that contacts are facilitated by the cytoskeleton? Why would increasing contacts increase Parkin recruitment, given that Parkin recruitment depends on stabilization of Pink1 and there is little reason to think actin would be important?

We think that dynamic forms of actin filaments have divergent roles in mediating ER-mitochondrial contacts. We agree with the reviewers that we have previously shown that actin filaments generated by an ER bound formin protein INF2 regulates and promotes close contact between ER and mitochondria (Chakrabarti R et al., JCB 2018). However Arp2/3 fundamentally generates distinctly different form of actin filaments. While Formins generates linear actin filaments, Arp2/3 complex generates branched actin filaments that have been shown to produce force specially

characterized and implicated in lamellipodia based cell migration. In the case of ADA, we think that the Arp2/3 mediated branched actin assembly produces a force to separate the two organelles while in case of INF2 generated actin filaments from the ER promotes contacts through Myosin II mediated contraction of the two organelles (Chakrabarti R et al., JCB, 2022; Kage et al., MboC, 2022).

We show that Parkin recruitment in MEF and HeLa cells preferentially occurs at ERMC with a significant population of Parkin puncta present exclusively at ER (**Fig 2A-F**). This raises the importance of ER in recruitment of Parkin. Further we show that enhancing and reducing ERMC increases or decreases Parkin recruitment respectively indicating that abundance of ERMC is important for Parkin recruitment (**Fig 3**) We also show that VDAC1 (a known ERMC tether protein) is required downstream of ERMC enhancement to facilitate Parkin recruitment (**Fig 4**). Though we show that ADA acutely regulates Pink1 accumulation and/or activity (**Fig 6**), further studies are required to ascertain whether this is through modulation of ERMC. In the present study we show that ADA acutely not only alters ERMC, which is important for regulating kinetics of Parkin recruitment but also affects Pink1 stabilization which would regulate Parkin recruitment as well.

Minor points:

1) In abstract, the following typo should be corrected: Acute Damaged-induced Actin.

Corrected

2) In Figure 2A, it is difficult to appreciate any differences in ER-mitochondrial overlap in the images.

In the light of the new data we have omitted this data

3) Specificity controls for the PLA assay should be shown.

We performed negative controls of PLA assay either with VAP-B antibody alone or with PTPIP51 antibody alone (**Fig R5**).

The antibody concentration and incubation

times were standardized to significantly reduce non-specific signals

Dear Dr. Chakrabarti,

Thank you for submitting your revised manuscript. It has now been seen by all original referees. My apologies for the delay in getting back to you, which was due to the delay in receiving complete set of referee reports and conference travel.

As you will see, referees find that the study is significantly improved during revision and recommend publication. However, referees #2 and #3 have remaining outstanding concerns. Please address them by making textual alterations. Please acknowledge the limitations pointed out by referee #3 regarding the manipulation of ERM by expression of tethers in the Discussion section. Please provide a point-by-point response outlining the revisions made in response to each comment.

Moreover, the editorial points below need to be addressed before I can accept the manuscript.

- Please rename Data availability section as Data Availability and place it before the Acknowledgements.
- Please remove the Author Contribution section from the manuscript text, in the relevant sections.
- Please fill out and include an author checklist as listed in our online guidelines (<https://www.embopress.org/page/journal/14693178/authorguide>)
- Please address the following regarding the funding information: the two funders provided in the Comments box need to be removed from the box and entered as separate funders (More Funders option); it is perfectly fine if our system does not recognize a funder and a yellow attention sign appears - we just need the separate entries to match the funders listed in the Acknowledgments.
- We note that the figures are provided in .ppt format, which is not allowed by EMBO Press policy. For publication, we require TIFF, PDF or EPS files in PC or Macintosh format, preferably from PhotoShop or Illustrator software. We cannot accept Freehand, Canvas, CorelDRAW or MacDrawPro files. These files must be converted to postscript (eps) format. For any figures submitted in Photoshop or TIF(F) format we require layered files to be sent whereby all text, arrows or additional attributes are placed on individual layers within the file. For line art/charts/graphs we prefer to work with Adobe Illustrator AI, EPS, or high-resolution PDF files.
- Along similar lines, during our routine image analysis, we've noticed a compression issue affecting all figures in the manuscript. It appears the images were imported into PowerPoint, which has introduced visible pixelation and reduced resolution. To ensure the best possible reproduction quality, could you please provide the original, high-resolution versions of all figures (preferably in TIFF or high-quality PNG format). Avoiding intermediary formats like PowerPoint will help preserve image clarity and detail."
- We note that Figure 2H is currently not called out in the text.
- All research articles submitted as revised versions must include a structured methods section that includes a Reagents and Tools Table followed by a Methods and Protocols section. Please see <https://www.embopress.org/page/journal/14693178/authorguide#structuredmethods> for further information.
- Figure legends should be placed at the end of the manuscript, after the References.
- Our production/data editors have asked you to clarify several points in the figure legends - Figure Legends (main + EV):
 - o Please note that the figure legend of EV3 I is mislabeled as figure EV3 F in the manuscript. This needs to be rectified.
 - o Please note that the exact p values are not provided in the legends of figures 1C, F; 2H, J; 3C, G; 4C, 5B, D; 6B, D; 7B, D, F, H, J; 8B, D, F; EV1 D, G, I; EV2 E, EV3 F, G; EV4 D; EV5 A, E, F.
 - o Please note that information related to n is missing in the legends of figures 1C, F; 5D, 6B, D; EV1 D, EV1 G, EV3 G, EV5 B, E.
 - o Please note that the error bars are not defined in the legends of figures 2C, D, F; 5D, 6B, D; 8B, EV2 A, C; EV3 C, G, I; EV4 C, D, E; EV5 B.
 - o Please note that the measure of center for the error bars needs to be defined in the legends of figures 2H, J; 3G, 4C, 5B, 6B, D; 7D, F, H, J; 8D, F; EV1 A, I; EV2 C, EV3 F, EV5 A, D, E, F, G.
 - o Please note that the scale bar is missing for figures 3D, F; 8E.
 - o Please note that the white arrows are not defined in the legend of figure 2E, EV3 B. This needs to be rectified.
- Papers published in EMBO Reports include a 'synopsis' and 'bullet points' to further enhance discoverability. Both are displayed on the html version of the paper and are freely accessible to all readers. The synopsis includes a short standfirst summarizing the study in 1 or 2 sentences (max 35 words) that summarize the paper and are provided by the authors and streamlined by the handling editor. I would therefore ask you to include your synopsis blurb and 3-5 bullet points listing the key experimental findings.
- In addition, please provide an image for the synopsis. This image should provide a rapid overview of the question addressed in the study but still needs to be kept fairly modest since the image size cannot exceed 550 (width) x 300-600 (height) pixels.

Thank you again for giving us to consider your manuscript for EMBO Reports, I look forward to your minor revision.

Kind regards,

Deniz Senyilmaz Tiebe

Referee #1:

I would like to thank the authors for their extensive revision. They have effectively addressed most of my previous points experimentally and provided clear responses to the remaining questions. Overall, I believe the study is both interesting and relevant for the fields of quality control and membrane contact sites. In my opinion, the study is suitable for publication in EMBO Reports.

Referee #2:

Comments to the authors

In the revised version of the manuscript, the authors now included several critical data that strengthen their argument. The CK666 treatment-induced upregulation of mitophagy under the chronic mitochondrial damage, as monitored by a reporter construct, is striking. The mechanistic difference between VDAC1 and MFN2 in PINK1/Parkin-mediated mitophagy is intriguing.

Minor points

1. On page 5, MEF cells should be included in the following sentence; Taken together, these results suggest that ADA significantly delays Parkin recruitment onto damaged mitochondria in both U2OS and HeLa cells.
2. On page 6, CCCP treatment in the following sentence should be A/O treatment; HeLa cells treated with CK666 show significantly higher mitochondrially localized Parkin upon 20 and 40 minutes of CCCP treatment.
3. On page 8, about the requirement of VDAC for Parkin recruitment, it might be better to cite following paper as well; Geisler et al., Nat Cell Biol., 2010, PMID: 20098416.
4. PINK1 paper cited on page 15 in Discussion (Guardia-Laguarta et al., 2019) seems to focus on ER-mitochondria contact sites as a location for the degradation of PINK1 cleaved-form. Therefore, the discussion point based on this paper is unclear.

Referee #3:

Concerning the issue of whether Arp2/3 inhibition results in faster CCCP-induced Parkin recruitment because there is synergistic stress, the authors write in the Discussion:

"One likely explanation might be that these intervention through eradication of ADA induces an increased mitochondrial stress when coupled with CCCP treatment. However, the fact that modulation of ERMC either by increasing them (VAPB OE, ER-RFP-mito OE) or by depleting them (Mfn2 KD, VDAC1 KD and PM- RFP-Mito OE) significantly modifies Parkin recruitment in the presence of ADA suggests that most likely ADA regulates Parkin recruitment through acutely disrupting ERMC."

This explanation is not very satisfying. The observation that increasing ERMC increases Parkin recruitment really does not address the issue of the stress level. More fundamentally, the authors do not provide any satisfying explanation for why ADA would exist for the purpose of delaying mitophagy in the presence of mitochondrial stress.

One of the original concerns is that the main experimental approach is very blunt. This concern remains a prominent feature in the revision. The manipulation of ERMC by expression of tethers fundamentally changes cell physiology, and it is hard to be sure of the mechanistic basis of the effects seen.

Detailed comments

1. Figure 1B is missing in the PDF.
2. Figure 2A, B, E: Since the colocalization of Parkin-GFP and mitochondria or ER is difficult to see, presenting a 3-channel merge image and single-channel images would be better.
3. There are several figure labeling errors in the manuscript. For example, (Fig 2E, F) on page 7, line 2, should be corrected to be (Fig 2G, H).
4. Page 8, lines 3 and 4: Fig EV4E, F is mentioned, but Fig EV4F does not exist.
5. Figure 6A, C: Are both bands Pink1, or just one? What is the evidence that this is really Pink1? In other publications, Pink1 stabilization by CCCP is much more dramatic than in these blots.
6. Figure 6B, D: Is the band intensity of Pink1 normalized by the band intensity of loading control (actin)?

Referee #1:

I would like to thank the authors for their extensive revision. They have effectively addressed most of my previous points experimentally and provided clear responses to the remaining questions.

Overall, I believe the study is both interesting and relevant for the fields of quality control and membrane contact sites. In my opinion, the study is suitable for publication in EMBO Reports.

We would like to thank and acknowledge the reviewer for the insightful suggestions which has certainly enhanced the quality of the manuscript.

Referee #2:

Comments to the authors

In the revised version of the manuscript, the authors now included several critical data that strengthen their argument. The CK666 treatment-induced upregulation of mitophagy under the chronic mitochondrial damage, as monitored by a reporter construct, is striking. The mechanistic difference between VDAC1 and MFN2 in PINK1/Parkin-mediated mitophagy is intriguing.

We thank the reviewer for pointing out the mistakes in the manuscript that have now been corrected in the final version. We also thank the reviewer for the suggestions which certainly increased the robustness of the study.

Minor points

1. On page 5, MEF cells should be included in the following sentence; Taken together, these results suggest that ADA significantly delays Parkin recruitment onto damaged mitochondria in both U2OS and HeLa cells. We have rectified this and added “MEF cells” to the relevant area in the text.
2. On page 6, CCCP treatment in the following sentence should be A/O treatment; HeLa cells treated with CK666 show significantly higher mitochondrially localized Parkin upon 20 and 40 minutes of CCCP treatment. We apologize for this mistake and have now rectified this in the manuscript
3. On page 8, about the requirement of VDAC for Parkin recruitment, it might be better to cite following paper as well; Geisler et al., Nat Cell Biol., 2010, PMID: 20098416. We have added this citation as suggested by the reviewer.
4. PINK1 paper cited on page 15 in Discussion (Guardia-Laguarta et al., 2019) seems to focus on ER-mitochondria contact sites as a location for the degradation of PINK1 cleaved-form. Therefore, the discussion point based on this paper is unclear. We acknowledge this point and have decided to leave out this part from the discussion.

Referee #3:

Concerning the issue of whether Arp2/3 inhibition results in faster CCCP-induced Parkin recruitment because there is synergistic stress, the authors write in the Discussion:

"One likely explanation might be that these intervention through eradication of ADA induces an increased mitochondrial stress when coupled with CCCP treatment. However, the fact that modulation of ERMC either by increasing them (VAPB OE, ER-RFP-mito OE) or by depleting them (Mfn2 KD, VDAC1 KD and PM- RFP-Mito OE) significantly modifies Parkin recruitment in the presence of ADA suggests that most likely ADA regulates Parkin recruitment through acutely disrupting ERMC."

This explanation is not very satisfying. The observation that increasing ERMC increases Parkin recruitment really does not address the issue of the stress level. More fundamentally, the authors do not provide any satisfying explanation for why ADA would exist for the purpose of delaying mitophagy in the presence of mitochondrial stress.

One of the original concerns is that the main experimental approach is very blunt. This concern remains a prominent feature in the revision. The manipulation of ERMC by expression of tethers fundamentally changes cell physiology, and it is hard to be sure of the mechanistic basis of the effects seen.

We acknowledge the specific concerns of the reviewer and agree that further studies are required to verify whether modulation of ERMC imposes additional mitochondrial stress. From our studies we reason that a healthy proximity of ER and mitochondria is required for proper Parkin recruitment as de-tethering ER and mitochondria significantly inhibits Parkin recruitment. We totally agree that further studies are required to see whether enhancing ERMC through synthetic tethers increases mitochondrial stress and conversely de-tethering ERMC through the PM-Mito synthetic tethers decreases mitochondrial stress leading to altered Parkin recruitment. This would be an intriguing line of investigation and we thank the reviewer for bringing in this insightful though that needs to be investigated in future. We have now added the following lines to the relevant section in the discussion:

"This being said, modulation of ERMC either through synthetic tethers or depletion of specific molecules could induce additional mitochondrial stress that might have its own effect on the dynamics of Parkin recruitment to dysfunctional mitochondria and additional studies are specifically needed to further address this issue."

Additionally we are equally intrigued and puzzled to see that during chronic mitochondrial damage, removal of ADA speeds up formation of mitolysosome. In fact mitochondrial mass is preserved in mtDNA depleted Rho 0 cells where mitochondria do not have any membrane potential and the cells completely rely on glycolysis due to non-existent oxphos. However these cells do not have loss of mitochondria suggesting that either they are preserved for some "unknown" function or they cant be removed through mitophagy due to some roadblock. We now at least provide some evidence that the "roadblock" might be the persistent ADA filaments, and on these lines this might be more of a pathological adaptation rather than a physiological

consequence. Further studies are required to evaluate the overall health of these cells upon removal of ADA and promotion of mitophagy.

Detailed comments

1. Figure 1B is missing in the PDF. We have now corrected this which might have been due to an error during compression.
2. Figure 2A, B, E: Since the colocalization of Parkin-GFP and mitochondria or ER is difficult to see, presenting a 3-channel merge image and single-channel images would be better. The three channel merge is presented on the left (main figure) and we also show merge of Parkin/ER and Parkin Mito to clarify that Parkin is in the vicinity of both ER and Mito.
3. There are several figure labeling errors in the manuscript. For example, (Fig 2E, F) on page 7, line 2, should be corrected to be (Fig 2G, H). We thank the reviewer for pointing this out and apologize for this mistake and have corrected this.
4. Page 8, lines 3 and 4: Fig EV4E, F is mentioned, but Fig EV4F does not exist. We thank the reviewer for pointing this out and apologize for this mistake and have corrected this.
5. Figure 6A, C: Are both bands Pink1, or just one? What is the evidence that this is really Pink1? In other publications, Pink1 stabilization by CCCP is much more dramatic than in these blots. We evaluated Pink1 stabilization by transfecting GFP-Pink1 which would give immunoreactivity at 75 Kd (as observed in our Western blots). In our hands the endogenous protein was not detected using the antibody used. While quantifying we have used the total intensity from both bands to quantify relative levels of Pink1.
6. Figure 6B, D: Is the band intensity of Pink1 normalized by the band intensity of loading control (actin)? Yes the band intensity was normalized to the actin loading control

Dr. Rajarshi Chakrabarti
Thomas Jefferson University
Pathology and Genomic Medicine
1020 Locust Street
Jefferson Alumni Hall 527G
Philadelphia, Pennsylvania 19107
United States

Dear Raj,

Thank you for submitting your revised manuscript. I have now looked at everything and all is fine. Therefore, I am very pleased to accept your manuscript for publication in EMBO Reports.

Congratulations on a nice work!

Kind regards,

Deniz

--

Deniz Senyilmaz Tiebe, PhD
Senior Scientific Editor
EMBO Reports
